# DATA VALUE ESTIMATION ON PRIVATE GRADIENTS

## ABSTRACT

For gradient-based machine learning (ML) methods commonly adopted in practice such as stochastic gradient descent, the *de facto* differential privacy (DP) technique is perturbing the gradients with random Gaussian noise. Data valuation attributes the ML performance to the training data and is widely used in privacy-aware applications that require enforcing DP such as data pricing, collaborative ML, and federated learning (FL). *Can existing data valuation methods still be used when DP is enforced via gradient perturbations?* We show that the answer is no with the default approach of injecting i.i.d. random noise to the gradients because the *estimation uncertainty* of the data value estimation paradoxically *linearly scales* with more estimation budget, producing estimates almost like random guesses. To address this issue, we propose to instead inject carefully correlated noise to provably remove the linear scaling of estimation uncertainty w.r.t. the budget with some assumptions on the gradient distribution. We also empirically demonstrate that our method gives better data value estimates on various ML tasks and is applicable to use cases including dataset valuation and FL.

## 1 INTRODUCTION

With growing data privacy regulations (Bukaty, 2019; Council of European Union, 2014) and machine learning (ML) model attacks (Shokri et al., 2017), privacy has become a primary concern in many scenarios such as collaborative ML (Sim et al., 2020; Xu et al., 2021) and federated learning (FL) (McMahan et al., 2017; Yang et al., 2019). Differential privacy (DP) (Dwork & Roth, 2014) is commonly adopted as the *de facto* framework to provide privacy protection for training data with theoretical guarantees. In deep learning, DP is typically achieved by perturbing the gradients w.r.t. the model's loss on the training data (Abadi et al., 2016).

Data valuation (Ghorbani & Zou, 2019) has received increased interest from providing attribution for parties in collaborative ML (Sim et al., 2022) and identifying data quality for data curation (Ghorbani & Zou, 2019) and data marketplace (Agarwal et al., 2019). Data values are often estimated with sampling-based methods such as Monte Carlo (Castro et al., 2009) on user statistics, e.g. gradients of user data (Ghorbani & Zou, 2019). However, the sensitive nature of the user statistics requires privacy during data valuation to protect the data of participants in collaboration (Sim et al., 2023) or to facilitate interaction between data buyers and sellers in a marketplace (Chen et al., 2023). Specifically, enforcing DP on the data is desirable in these scenarios for two reasons: 1) DP enjoys the post-processing immunity (Dwork & Roth, 2014), allowing further access to data without risking privacy leakage; 2) DP offers a natural trade-off between privacy protection and data quality: Data contributors can determine at their discretion the level of information protection at the expense of degraded data quality. While some previous works considered DP in data valuation (Sim et al., 2023; Wang et al., 2023; Watson et al., 2022), they either focused on a limited class of ML models or required a trusted central server, both of which are difficult to satisfy in real-world scenarios (Sim et al., 2022). A natural question arises: *Can we circumvent these two challenges in data valuation while enforcing DP?*

One might think of perturbing gradients on user data before using them for updating a gradient-trained parametric model to overcome the challenges. Unfortunately, we demonstrate that the naive approach of adding i.i.d. Gaussian noise (Dwork & Roth, 2014) to user gradients leads to practically useless data value estimates: The perturbation on the gradients erodes the information they carry, leading to a lowered data value and increased data value estimation uncertainty because the noise introduced by such perturbation accumulates with repeated evaluations (i.e., sampling methods

methods (Castro et al., 2009; Maleki et al., 2014)). While a lowered (mean of) data value estimate is intuitive and expected as a result of information loss (Dwork et al., 2015; Kairouz et al., 2015), *increased estimation uncertainty* can render existing data valuation methods useless because the magnitude of the noise scales with the evaluation budget for a fixed DP guarantee. As an illustration, the 3rd figure of Fig. 2 shows that as the number of evaluations $k$ increases, removing data with high data value estimates produces a curve closer to that with random removal, implying that more evaluation samples, paradoxically, have *worsened* the quality of data value estimates. We theoretically account for this counter-intuitive phenomenon by showing that perturbation introduced by DP can cause the estimation uncertainty of data value estimates, measured in terms of the variance brought by the perturbation, to *scale linearly in the number of evaluations*.

Fortunately, through our developed theoretical analysis, we derive an insight that leads to a method for controlling the estimation uncertainty. We focus the analysis on the family of semivalues widely adopted as data valuation metrics such as data Shapley (Ghorbani & Zou, 2019), Beta Shapley (Kwon & Zou, 2022), and data Banzhaf (Wang & Jia, 2023). We revisit the paradigm of data valuation in the context of gradient-based DP ML and thus design a technique that perturbs the gradients with *correlated noise* in repeated evaluations to mitigate the above issue. Contrary to the linear scaling of estimation uncertainty of the data value estimates with the evaluation budget in the naive approach, our proposed method is shown to control the estimation uncertainty to a *constant*. Additionally, we empirically demonstrate that, on various ML tasks, our proposed method produces (i) greater model degradation from removing high-value data; and (ii) higher AUC scores in identifying label-corrupted data. In comparison, the naive approach performs similarly to random selection. We also apply our approach to other scenarios including dataset valuation (Wu et al., 2022) and FL (McMahan et al., 2017). Our specific contributions are summarized as follows:

- We formalize a notion of *estimation uncertainty* (Eq. (3)) to specifically target the uncertainty due to DP. We then theoretically identify that under the naive approach of injecting i.i.d. noise for DP, the estimation uncertainty grows in $\Omega(k)$ with the evaluation budget $k$ (Prop. 5.1), resulting in low-quality data value estimates.
- As mitigation, we propose a simple yet effective approach (in Algorithm 1) via injecting correlated noise to control the estimation uncertainty to $\mathcal{O}(1)$ (Prop. 5.4) as opposed to $\Omega(k)$.
- We empirically demonstrate the implications of the escalating estimation uncertainty shown by the near-random performance on the data removal task (Sec. 6.1) and noisy label detection task (Sec. 6.2). Our approach outperforms the baseline approach on these tasks (Sec. 6.2) and on noisy label detection task in dataset valuation and FL (Sec. 6.3).

## 2 RELATED WORK

Prior works (Sim et al., 2023; Wang et al., 2023; Watson et al., 2022; Usynin et al., 2024) that consider privacy-aware data valuation have limited applicability due to limited settings. (Sim et al., 2023) considered perturbing user statistics to ensure DP but is restricted on the class of Bayesian models whereas we consider a wider family of models trained with gradient-based methods including neural networks. (Wang et al., 2023) proposed a private variant of KNN-Shapley but did not generalize to other semivalues, whereas our method applies to all semivalues. Watson et al. (2022) directly perturbed the semivalue estimates. However, both (Wang et al., 2023; Watson et al., 2022) require a trusted server to centralize the original gradients which may not reflect real-world scenarios where untrusted central servers pose added privacy risks, whereas our approach does not require a trusted central server. (Usynin et al., 2024) assessed using variance of gradients and privacy loss-input susceptibility score to select useful data points for DP training. The authors further propose methods to compute the DP version of these scores. (Bani-Harouni et al., 2023) considered improving the performance of DP-SGD by utilizing cosine similarity between privatized per-sample gradient and original gradient to decide whether to include the gradient in averaged gradient.

Li & Yu (2023); Wang & Jia (2023) identified that stochastic utility functions can lead to noisy data values which deteriorated the rank preservation and proposed to use (weighted) Banzhaf values as the semivalue metric. Although the noise introduced by DP also renders the utility functions stochastic, their methods do not consider the DP setting where noise scales with the number of evaluation budgets. We specifically consider mitigating the issue of the scaling noise.

Dwork et al. (2010); Li et al. (2015); Nasr et al. (2020) introduced correlated noise to mitigate the effect of noise brought by DP, which has since been widely adopted in online learning with DP requirement to improve learning performance (Choquette-Choo et al., 2023a;b; Denisov et al., 2023; Kairouz et al., 2021). In contrast, we apply correlated noise to the setting of data valuation with privacy needs, to improve the quality of data value estimates. (Koloskova et al., 2024) analyzed the use of correlated noise under a DP follow-the-regularized-leader setting.

## 3 PRELIMINARIES

We recall the definition of semivalue for data valuation (Ghorbani & Zou, 2019) and the necessary preliminaries on DP.

### 3.1 DATA VALUATION

**Semivalues.** Denote $[n] := \{1, 2, \ldots, n\}$. The *semivalue* of $i$ in a set $[n]$ of parties w.r.t. a utility function $V : 2^{[n]} \to \mathbb{R}$ and a weight function $w : [n] \to \mathbb{R}$ s.t. $\sum_{r=1}^{n} \binom{n-1}{r-1} w(r) = n$ is (Dubey et al., 1981)

$$\phi_i := \sum_{r=1}^{n} n^{-1} w(r) \sum_{S \subseteq N \setminus \{i\}, |S|=r-1} [V(S \cup \{i\}) - V(S)] . \tag{1}$$

Leave-one-out (Cook, 1977), Shapley value (Shapley, 1953), and Banzhaf value (Wang & Jia, 2023) are examples of semivalues. In data valuation, a party can be represented by a data point, a dataset, or (the data of) an agent in FL setting. In ML, semivalues are often treated as a random variable and estimated using Monte Carlo methods (Castro et al., 2009; Maleki et al., 2014) since $[n]$ is usually large and $V$ is stochastic (more details in App. B.1). Denote $P_j^{\pi}$ as the set of predecessors of party $j$ in a permutation $\pi$ uniformly randomly drawn from the set of all permutations $\Pi$, and let $p_j(\pi) := 2^{n-1} n^{-1} w(|P_j^{\pi} \cup \{j\}|)$, then $\phi_j = \mathbb{E}[\psi_j]$ with $\psi_j$ an average over $k$ random draws:

$$\psi_j = (1/k) \sum_{t=1}^{k} p_j(\pi^t) [V(P_j^{\pi^t} \cup \{j\}) - V(P_j^{\pi^t})] . \tag{2}$$

### 3.2 DIFFERENTIALLY PRIVATE MACHINE LEARNING (DP ML)

**Definition 3.1** (($\epsilon, \delta$)-Differential Privacy (Dwork & Roth, 2014, Def. 2.4)). A randomized algorithm $\mathcal{M}$ with domain $\mathcal{D}$ and range $\mathcal{R}$ is said to be ($\epsilon, \delta$)-differentially private if for any two neighboring[1] datasets $d, d' \in \mathcal{D}$, and for all event $S \subseteq \mathcal{R}$, $\Pr(\mathcal{M}(d) \in S) \leq \exp(\epsilon) \Pr(\mathcal{M}(d') \in S) + \delta$.

Importantly, the DP guarantee of $\mathcal{M}$ is immune against post-processing (Dwork & Roth, 2014, Proposition 2.1): The composition $f \circ \mathcal{M}$ with an arbitrary randomized mapping $f$ have the same DP guarantee as $\mathcal{M}$. We adopt this definition of DP to show the linearly scaling effect of perturbation (Sec. 5.1). We elaborate in App. B.2 that our analysis can be extended to other DP frameworks.

## 4 SETTINGS AND PROBLEM STATEMENT

**Settings.** Our analysis is based on the G-Shapley framework (Ghorbani & Zou, 2019) for gradient-based ML methods where a parametric ML model learns from the data of each party via the perturbed gradients of the data against a deterministic loss function $\mathcal{L} : [n] \times \mathbb{R}^d \to \mathbb{R}$ which maps the (data of a) party and model parameters ($\in \mathbb{R}^d$) to a score ($\in \mathbb{R}$). The utility improvement reflects the data value after the model updates its parameters with the gradient. For an evaluation budget $k$, $k$ uniformly random permutations $\pi^1, \ldots, \pi^k \in \Pi$ are sampled, and for each sampled permutation $\pi$ (superscript omitted), a model is randomly initialized with $\boldsymbol{\theta}_{\pi}$ and updated by the parties *in sequence* according to $\pi$. Then, denote $\boldsymbol{\theta}_{\pi,j}^p$ the model parameters immediately before party $j$ updates the model in permutation $\pi$. For each subsequent party $j$ in $\pi$, the Gaussian mechanism is adopted to obtain a perturbed gradient $\tilde{g}_{\pi,j}$ to update the model $\boldsymbol{\theta}_{\pi,j} := \boldsymbol{\theta}_{\pi,j}^p - \alpha \tilde{g}_{\pi,j}$ where $\tilde{g}_{\pi,j} := \hat{g}_{\pi,j} + z$ from a Gaussian noise $z \sim \mathcal{N}(\mathbf{0}, (C\sigma)^2 \boldsymbol{I})$ (with $C, \sigma > 0$) and the norm clipped gradient $\hat{g}_{\pi,j} := g_{\pi,j} / \max(1, \|g_{\pi,j}\|_2 / C)$ based on the gradient $g_{\pi,j} := \nabla_{\boldsymbol{\theta}} \mathcal{L}(j, \boldsymbol{\theta}_{\pi,j}^p)$ derived from $j$'s data. For a fixed test dataset, a utility $V(P_j^{\pi} \cup \{j\})$ representing the test performance depends on

---

[1]Two inputs $\boldsymbol{x}, \boldsymbol{x}'$ are neighboring if they differ by one training example (Abadi et al., 2016).

the model parameter $\boldsymbol{\theta}_{\pi,j}$ (e.g. $V$ is the negated test loss), so we replace $V(P_j^\pi \cup \{j\})$ with $V(\boldsymbol{\theta}_{\pi,j})$ hereafter to highlight the interaction between model parameters and utility. While it is possible $P_j^\pi \cup \{j\}$ results in different $\boldsymbol{\theta}_{\pi,j}$ due to the random model initialization and varying orders of parties in $\pi$, our subsequent definition of estimation uncertainty (Eq. (3)) carefully excludes their effects and focuses on the uncertainty due to DP if $V$ is *deterministic* w.r.t. model parameters $\boldsymbol{\theta}$ as we fix $\boldsymbol{\theta}_{\pi,j}^p$.

Algorithm 1 outlines this i.i.d. noise approach , our method and a variant , explained in Sec. 5.

---

**Algorithm 1** i.i.d. Corr. Noise ($\boldsymbol{X}$) Corr. Noise ($\boldsymbol{Y}$)

---

1: **Input:** number of parties $n$, utility function $V$, clipping norm $C$, loss function $\mathcal{L}$, noise multiplier $\sigma$, number of evaluations $k$, weight coefficient $p_j$, burn-in ratio $q$ .
2: **for** $t \leftarrow 1$ to $k$ **do**
3:      Draw $\pi^t \overset{\text{unif.}}{\sim} \Pi$ and initialize the model with $\boldsymbol{\theta}_{\pi^t}$;
4:      $i \leftarrow \pi^t[0]$ ; $\boldsymbol{\theta}_{\pi^t,i} \leftarrow \boldsymbol{\theta}_{\pi^t}$
5:      **for** $j \in \{\pi^t[1], \pi^t[2], \ldots, \pi^t[n]\}$ **do**
6:         $g_{\pi^t,j} \leftarrow \nabla_{\boldsymbol{\theta}} \mathcal{L}(j, \boldsymbol{\theta}_{\pi^t,j}^p)$
7:         $\tilde{g}_{\pi^t,j} \leftarrow g_{\pi^t,j} / \max(1.0, \|g_{\pi^t,j}\|_2 / C) + \mathcal{N}(\boldsymbol{0}, k(C\sigma)^2 \boldsymbol{I})$
8:         **if** Correlated Noise ($\boldsymbol{X}$) **or** Correlated Noise ($\boldsymbol{Y}$) **then**
9:             $\tilde{g}_{\pi^t,j} \leftarrow (1 - \boldsymbol{X}_{t,t}) \times \tilde{g}_j^{\text{roll}} + \boldsymbol{X}_{t,t} \times \tilde{g}_{\pi^t,j}$
10:           $\tilde{g}_j^{\text{roll}} \leftarrow (t-1)/t \times \tilde{g}_j^{\text{roll}} + 1/t \times \tilde{g}_{\pi^t,j}$
11:         **end if**
12:         $\boldsymbol{\theta}_{\pi^t,j} \leftarrow \boldsymbol{\theta}_{\pi^t,j}^p - \alpha \tilde{g}_{\pi^t,j}$
13:         **if** Correlated Noise ($\boldsymbol{Y}$) **then**
14:            **if** $t > kq$ **then**
15:                $\psi_j \leftarrow \frac{t-kq-1}{t-kq} \psi_j + \frac{p_j(\pi^t)}{t-kq} (V(\boldsymbol{\theta}_{\pi^t,j}) - V(\boldsymbol{\theta}_{\pi^t,i}))$
16:            **end if**
17:            $i \leftarrow j$ and **Continue**
18:         **end if**
19:         $\psi_j \leftarrow \frac{t-1}{t} \psi_j + p_j(\pi^t)(V(\boldsymbol{\theta}_{\pi^t,j}) - V(\boldsymbol{\theta}_{\pi^t,i}))/t$ ; $i \leftarrow j$
20:      **end for**
21: **end for**
22: **Return** $\psi$

---

**Problem Statement.** We study how noise introduced by DP impacts the estimation uncertainty of semivalue estimates. Formally, the estimation uncertainty for each party $j$ is defined as

$$\text{Var}[\psi_j | \boldsymbol{\theta}_{\pi^1,j}^p, \boldsymbol{\theta}_{\pi^2,j}^p, \ldots, \boldsymbol{\theta}_{\pi^k,j}^p] \,, \tag{3}$$

and we aim to precisely understand its (asymptotic) relationship with $k$ under i.i.d. noise and using our proposed method (in Sec. 5) respectively. While Eq. (3) may seem more complicated than $\text{Var}[\psi_j]$, the conditioning effectively removes the stochasticity in permutation sampling, gradient descent, and model initialization, thus focusing on the impact of noise introduced by DP. Indeed, when there is no perturbation due to DP and $V$ is deterministic (w.r.t. $\boldsymbol{\theta}$), $\text{Var}[\psi_j | \boldsymbol{\theta}_{\pi^1,j}^p, \boldsymbol{\theta}_{\pi^2,j}^p, \ldots, \boldsymbol{\theta}_{\pi^k,j}^p] = 0$, meaning Eq. (3) solely depends on the randomness from the perturbations of DP. Importantly, the definition of estimation uncertainty in Eq. (3) admits precise and intuitive results about the randomness due to DP, enabling us to pinpoint an issue with the i.i.d. noise approach; then, we propose the correlated noise approach to asymptotically reduce this uncertainty and mitigate the issue.

**Remark.** We note that minimizing the estimation uncertainty *does not* necessarily recover the original (i.e., without DP) data value estimate since DP exerts an upper bound on the marginal contribution (more discussed in App. B.4). Nevertheless, we highlight that this limitation does not diminish the merit of our analysis as reducing estimation uncertainty helps improve the preservation of the *ranking* of data value estimates of different parties compared to the naive approach of injecting

i.i.d. noise, which is an important axiomatic characterization and a common application of such data value metrics (Ghorbani & Zou, 2019; Zhou et al., 2023).

Our subsequent analysis is w.r.t. a particular party $j$ and for notational brevity, omits the subscript $j$ (e.g., $\boldsymbol{\theta}_{\pi,j} = \boldsymbol{\theta}_\pi, \tilde{g}_{\pi,j} = \tilde{g}_\pi$) where the context is clear. Table 4 in App. A consolidates the important notations used throughout this work.

## 5 Approach and Theoretical Results

We summarize the results on estimation uncertainty for the naive approach (i.i.d. noise), our method, and a variant in Table 1. All proofs and derivations are deferred to App. C.

Table 1: Summary of theoretical results. $\sigma_g^2$ refers to the average variance of unperturbed gradients. $q \in (0, 1)$ is a hyperparameter.

| Result | Asymptotic Estimation Uncertainty | Approach |
|---|---|---|
| Prop. 5.1 | $\Omega(k)$ | i.i.d. noise |
| Prop. 5.3 | $\mathcal{O}(\log^2 k + \sigma_g^4)$ | Our method |
| Prop. 5.4 | $\mathcal{O}\left(\log^2 (1/q)/(1-q)^2 + \sigma_g^4\right)$ | Our method (variant) |

Figure 1: Left: Box plots of $\Delta_{\cos}$ vs. budget $k$, where higher is better. Right: Box plots of $\Delta_{\ell_2}$ vs. budget $k$, where lower is better. $\epsilon = 1$ for both plots.

**Choice of $V$.** For the purpose of mathematical analysis, we consider, in our theoretical analysis, the case where $V$ is deterministic, non-positive, and Lipschitz smooth. An example of such $V$ is the (average) negated loss on a fixed test dataset (Ghorbani & Zou, 2019), elaborated in Sec. 5.1. Nevertheless, Sec. 6.1 additionally empirically investigates test accuracy as $V$, which is discrete, to show our method works even if these properties are relaxed.

### 5.1 I.I.D. Noise Causes Scaling Estimation Uncertainty

We first reveal that the propagation of noise from each $z_t$ to $V$ can be catastrophic to the estimator $\psi$: a *higher* evaluation budget $k$, surprisingly, leads to a *higher* estimation uncertainty.

The estimator $\psi$ in Eq. (2) aggregates various marginal contributions $m(\pi) = V(\boldsymbol{\theta}_\pi) - V(\boldsymbol{\theta}_\pi^p)$ over different $\pi \in \Pi$. Thus, if $\psi$ requires $k$ evaluations of $m$, party $j$ needs to reveal its gradients $k$ times. The repeated release of gradients increases the privacy risk, requiring greater perturbation to maintain the same DP guarantee. In the Gaussian mechanism, the best known variance of the Gaussian noise grows in $\Theta(k)$ (Abadi et al., 2016, Theorem 1) (more discussed in App. B.3), so each Gaussian noise is expressed as $z_t \sim \mathcal{N}(\boldsymbol{0}, k(C\sigma)^2 \boldsymbol{I})$ where both $C$ and $\sigma^2$ are constants satisfying a fixed $(\epsilon, \delta)$-DP guarantee hereafter.

The linear scaling of $\mathrm{Var}[z_t]$ creates a *vicious cycle*: More marginal contributions are needed to obtain a more certain semivalue estimation, incurring a larger $k$, which in turn necessitates greater perturbations for DP and increasing the estimation uncertainty. Formally, we show that the perturbations used to guarantee DP are amplified by a strongly concave utility function $V$ (achievable with a strongly convex loss w.r.t. model parameters, such as K-Means, Lasso, and logistic regression with weight decay), and causes the estimation uncertainty of semivalues to grow in the order of $\Omega(k)$, via two specific globally strongly convex loss functions:

**Proposition 5.1.** (**I.I.D. Noise**) $\forall t \in [k]$, denote $\boldsymbol{\theta}_{\pi^t} := \boldsymbol{\theta}_{\pi^t}^p - \alpha \tilde{g}_{\pi^t} = \boldsymbol{\theta}_{\pi^t}^p - \alpha(\hat{g}_{\pi^t} + z_t)$ where $\forall t \in [k], z_t \overset{\text{i.i.d.}}{\sim} \mathcal{N}(\boldsymbol{0}, k(C\sigma)^2 \boldsymbol{I})$ under the Gaussian mechanism. Denote a test dataset $\mathcal{D}_{\text{test}} = \{(\boldsymbol{x}_1, y_1), \ldots, (\boldsymbol{x}_l, y_l)\}$ of $l$ data points. If $V$ is the negated mean-squared error loss on a linear regression model

$$V(\boldsymbol{\theta}) := -l^{-1} \sum_{i=1}^l (\boldsymbol{\theta}^\top \boldsymbol{x}_i - y_i)^2 ,$$

or the negated $\ell_2$-regularized cross-entropy loss on a logistic regression model

$$V(\boldsymbol{\theta}) := l^{-1} \sum_{i=1}^l (1 - y_i) \log(1 - \mathrm{Sig}(\boldsymbol{\theta}, \boldsymbol{x}_i)) + l^{-1} \sum_{i=1}^l y_i \log(\mathrm{Sig}(\boldsymbol{\theta}, \boldsymbol{x}_i)) - \lambda \|\boldsymbol{\theta}\|_2^2$$

where $\text{Sig}(\boldsymbol{\theta}, \boldsymbol{x}_i) = (1 + e^{-\boldsymbol{\theta}^\top \boldsymbol{x}_i})^{-1}$ is the sigmoid function and $\lambda > 0$ is the regularization hyperparameter. Denote $m(\pi^t) := V(\boldsymbol{\theta}_{\pi^t}) - V(\boldsymbol{\theta}_{\pi^t}^p)$. Further denote a regular semivalue estimator $\psi := k^{-1} \sum_{t=1}^{k} p(\pi^t) m(\pi^t)$.[2] Then, $\text{Var}[\psi | \boldsymbol{\theta}_{\pi^1}^p, \boldsymbol{\theta}_{\pi^2}^p, \ldots, \boldsymbol{\theta}_{\pi^k}^p] = \Omega(k)$.

As an intuition, consider that in each iteration, a Gaussian noise of variance $k(C\sigma)^2$ is injected to each gradient and amplified when propagated to the loss (i.e., $-V$ by definition) due to strong convexity to become $\Omega(k^2)$, and the averaging over multiple evaluations shaves off a factor of $k$, yielding the final $\Omega(k)$ estimation uncertainty. Worse, the estimation uncertainty grows asymptotically with $k$: increasing the evaluation budget increases the estimation uncertainty due to the noise of DP.

One might ask whether this issue persists for deep neural networks that generally have a non-convex loss. As the loss surface of neural networks can exhibit strong convexity around local minima (Kleinberg et al., 2018; Milne, 2019), we derive, with a few additional assumptions including a notion of local strong convexity, a counter-part result to Prop. 5.1 w.r.t. deep neural networks where the estimation uncertainty is also $\Omega(k)$ via Prop. C.4 in App. C. In short, using i.i.d. noise can lead to scaling estimation uncertainty in many cases, raising a significant issue for data valuation, which we mitigate next.

## 5.2 Correlated Noise to Reduce Estimation Uncertainty

The key to mitigating this "linearly scaling" estimation uncertainty lies in reducing the variance of the noise $z_t$ in each iteration to render it less significant after amplification when propagated to the loss (equivalently $V$). Taking advantage of the way data values are estimated where private gradients are continuously released in each iteration, we can add a carefully correlated noise $z_t^*$ to the gradients $\hat{g}_{\pi^t}$ instead of independent noise $z_t \sim \mathcal{N}(\mathbf{0}, k(C\sigma)^2 \boldsymbol{I})$ while achieving the same DP guarantee.

Constructing $z_t^*$ exploits the post-processing property of DP: reusing previously released private statistics does not affect the DP guarantee level. At each iteration $t$, instead of directly injecting i.i.d. noise to $\hat{g}_{\pi^t}$ to become $\tilde{g}_{\pi^t} := \hat{g}_{\pi^t} + z_t$, estimation uncertainty can be reduced by reusing the i.i.d.-perturbed gradients $\tilde{g}_{\pi^1}, \ldots, \tilde{g}_{\pi^t}$. To ease the understanding of the core idea, we begin by assuming that the original gradients at each iteration are identical, i.e. $\hat{g}_{\pi^1} = \hat{g}_{\pi^2} = \ldots = \hat{g}_{\pi^k}$, and relax it later. Instead of $\tilde{g}_{\pi^t}$, a *weighted sum* (e.g., $\tilde{g}_{\pi^t}^* := t^{-1} \sum_{l=1}^{t} \tilde{g}_{\pi^l}$) of previously released private gradients can be utilized. This implicitly constructs the correlated noise $z_t^* := t^{-1} \sum_{l=1}^{t} z_l \sim \mathcal{N}(\mathbf{0}, (k/t)(C\sigma)^2 \boldsymbol{I})$, resulting a reduction in variance by a factor of $t$. To see how this variance reduction is propagated to $V$, compare the variance on the following strongly convex function (to mimic $V$): $\text{Var}[\|\tilde{g}_{\pi^t}^*\|_2^2] \to t^{-2} \text{Var}[\|\tilde{g}_{\pi^t}\|_2^2]$ as $k \to \infty$ (see Obs. C.1). Notice that the variance reduction is amplified from $t^{-1}$ to $t^{-2}$. Indeed, such variance reduction can lead to an $\mathcal{O}(\log^2 k)$ bound for the estimation uncertainty (see Prop. C.9). However, two questions remain: (i) Is the "identical gradient" assumption satisfied for semivalue estimation, and what if it is not? (ii) How to cleverly reuse previously privatized gradients to obtain a lower estimation uncertainty?

For question (i), unfortunately, the assumption is *not* satisfied: The gradients at each permutation $\hat{g}_{\pi^t}$, though obtained from the same underlying data, are not identical in general since $\boldsymbol{\theta}_{\pi^t}^p$ are different for different $\pi^t$. Specifically, consider a party $j$ whose unperturbed gradients in $k$ evaluations are $\hat{g}_{\pi^1}, \hat{g}_{\pi^2}, \ldots, \hat{g}_{\pi^k}$. The unperturbed gradients are first injected with a Gaussian noise to produce the perturbed gradients $\tilde{g}_{\pi^t} = \hat{g}_{\pi^t} + z_t$ for $t \in [k]$. Then, party $j$ will release the gradient

$$\tilde{g}_{\pi^t}^* := t^{-1} \sum_{l=1}^{t} \tilde{g}_{\pi^l} = t^{-1} \sum_{l=1}^{t} \hat{g}_{\pi^l} + t^{-1} \sum_{l=1}^{t} z_l$$

which, however, is not an unbiased estimator for $\hat{g}_{\pi^t}$ since $\mathbb{E}[\tilde{g}_{\pi^t}^*] = t^{-1} \sum_{l=1}^{t} \hat{g}_{\pi^l} \neq \hat{g}_{\pi^t}$ unless $\hat{g}_{\pi^1} = \hat{g}_{\pi^2} = \ldots = \hat{g}_{\pi^t}$ (i.e., identical gradients). Nevertheless, empirical observations suggest that as compared to $\tilde{g}_{\pi^t}$, $\hat{g}_{\pi^t}$ is much "closer" to $\tilde{g}_{\pi^t}^*$ in terms of the mean difference in cosine similarity

$$\Delta_{\cos} := n^{-1} \sum_{j \in [n]} k^{-1} \sum_{t=1}^{k} [\cos(\hat{g}_{\pi^t, j}, \tilde{g}_{\pi^t, j}^*) - \cos(\hat{g}_{\pi^t, j}, \tilde{g}_{\pi^t, j})]$$

where $\cos(a, b) := |a \cdot b| / (\|a\|_2 \|b\|_2)$, and the difference in $\ell_2$ distance

$$\Delta_{\ell_2} := n^{-1} \sum_{j \in [n]} k^{-1} \sum_{t=1}^{k} \|\hat{g}_{\pi^t, j} - \tilde{g}_{\pi^t, j}^*\|_2 - \|\hat{g}_{\pi^t, j} - \tilde{g}_{\pi^t, j}\|_2 \,,$$

---

[2] A regular semivalue has $p(\pi) > 0$ for all $\pi \in \Pi$ (Carreras & Freixas, 2002). Examples include Shapley value and Banzhaf value.

as shown in Fig. 1 where we empirically investigate the similarity with $400$ randomly selected data points from the diabetes dataset (Efron et al., 2004) trained with logistic regression. The difference is even more pronounced with a higher budget $k$, suggesting that $\tilde{g}^*_{\pi^t}$ is better than $\tilde{g}_{\pi^t}$ in approximating $\hat{g}_{\pi^t}$. Therefore, relaxing the assumption of identical gradients, we assume that they are i.i.d. samples of a distribution (also assumed in (Faghri et al., 2020; Zhang et al., 2013)). Additionally, we further assume an diagonal sub-Gaussian distribution, defined as follows.

**Definition 5.2** (**Diagonal Multivariate Sub-Gaussian Distribution**). Let $X \in \mathbb{R}^n$ be a random vector with $\mathbb{E}[X] = \mathbf{0}$. $X$ is said to follow a diagonal multivariate sub-Gaussian distribution if $\forall i \in [n], X_i$ follows a sub-Gaussian distribution and $\forall i, j \in [n]$ s.t. $i \neq j, \mathrm{Cov}[X_i, X_j] = 0$.

For question (ii), we specifically consider linear combinations of the private gradients to use linearity of expectation to ensure unbiasedness (see P2). Formally, we express the finally released gradients in all iterations as a matrix product $\boldsymbol{W} = \boldsymbol{X}\boldsymbol{A}$, where $\boldsymbol{A} = (\tilde{g}_{\pi^1}, \ldots, \tilde{g}_{\pi^k})^\top$ and $\boldsymbol{X}$ is a square matrix (which will be relaxed in Sec. 5.3) mapping $\boldsymbol{A}$ to the matrix of released gradients $\boldsymbol{W} = (\tilde{g}^*_{\pi^1}, \ldots, \tilde{g}^*_{\pi^k})^\top$. Since there are many possible matrices $\boldsymbol{W}$, we describe two key principles (specific to data valuation) in selecting $\boldsymbol{W}$, for a square matrix $\boldsymbol{X}$.

P1. **Lower traingularity**: $\boldsymbol{X}$ should be lower triangular.

P2. **Unbiasedness**: the combined gradient $\tilde{g}^*_{\pi^t}$ should be a *weighted sum* of the previous gradients, i.e., $\tilde{g}^*_{\pi^t} := \sum_{l=1}^t \boldsymbol{X}_{t,l}\tilde{g}_{\pi^l}$ where $\sum_{l=1}^t \boldsymbol{X}_{t,l} = 1$.

P1 ensures that each revealed gradient is calculated as a weighted prefix sum of the preceding perturbed gradients as future gradients are unknown. P2 ensures unbiased gradient estimate $\mathbb{E}[\tilde{g}^*_{\pi^t}] = \mathbb{E}[\hat{g}_{\pi^t}]$ (note that we treat $\hat{g}_{\pi^t}$ as a random variable here), and thus an unbiased estimate for the utility. The following square matrix $\boldsymbol{X}^*$ satisfies P1 and P2:

$$\boldsymbol{X}^*_{i,j} := i^{-1} \cdot \mathbf{1}_{j \leq i} . \tag{4}$$

Despite its simplicity, $\boldsymbol{X}^*$ is surprisingly a "one-size-fits-all" matrix as it delivers a lower asymptotic bound than $\Omega(k)$ (with i.i.d. noise) when $k$ is large, even if the unperturbed gradients are not identical:

**Proposition 5.3** (**Correlated Noise with $\boldsymbol{X}$**, *informal*). Let $\tilde{g}_{\pi^l}, l \in [t]$ be perturbed gradients using the Gaussian mechanism that satisfies $(\epsilon, \delta)$-DP. $\forall t \in [k]$, denote $\boldsymbol{\theta}^*_{\pi^t} := \boldsymbol{\theta}^p_{\pi^t} - \alpha \sum_{l=1}^t \boldsymbol{X}_{t,l}\tilde{g}_{\pi^l}$, $m^*(\pi^t) := V(\boldsymbol{\theta}^*_{\pi^t}) - V(\boldsymbol{\theta}^p_{\pi^t})$ and $\psi^* := k^{-1}\sum_{t=1}^k p(\pi^t)m^*(\pi^t)$. Assume that $\forall t \in [k], \hat{g}_{\pi^t} - \mathbb{E}[\hat{g}_{\pi^t}]$ i.i.d. follow an diagonal multivariate sub-Gaussian distribution with covariance $\Sigma \in \mathbb{R}^{(d \times d)}$ and let $\sigma_g^2 := d^{-1}\sum_{r=1}^d \Sigma_{r,r}$. Then, using a suitable matrix $\boldsymbol{X}$ can produce $\mathrm{Var}[\psi^* | \boldsymbol{\theta}^p_{\pi^1}, \boldsymbol{\theta}^p_{\pi^2}, \ldots, \boldsymbol{\theta}^p_{\pi^k}] = \mathcal{O}(\log^2 k + \sigma_g^4)$ and $\mathbb{E}[\psi^* - \psi] = \mathcal{O}(\log k + \sigma_g^2)$ while satisfying $(\epsilon, \delta)$-DP. Moreover, as $k \to \infty$, $\boldsymbol{X} \to \boldsymbol{X}^*$.

A formal statement of Prop. 5.3 and its proof is found in Appendix (Prop. C.10). Prop. 5.3 offers a better bound than $\Omega(k)$ in Prop. 5.1 as $\sigma_g^2$ is a constant. Notice that as $k \to \infty$, $\boldsymbol{X} \to \boldsymbol{X}^*$. An exciting implication of Prop. 5.3 is that one can use $\boldsymbol{X}^*$ to approximate $\boldsymbol{X}$ by setting a large $k$ regardless of $\sigma_g^2$ and achieve the given asymptotic bound. Another implication is that implementation of Prop. 5.3 is simple since all $\boldsymbol{X}_{t,l}$ are identical except for $\boldsymbol{X}_{t,t}$. Thus $\boldsymbol{X}$ is fully specified by just $k$ parameters: $\boldsymbol{X}_{t,t}$ for $t \in [k]$. As shown in Algorithm 1 in red , on top of i.i.d. , the model is updated using a weighted sum between $\tilde{g}^*_{\pi^{t-1}}$ and $\tilde{g}_{\pi^t}$ at each iteration $t$.

## 5.3 More Variance Reduction with Non-square Matrix

The bound in Prop. 5.3, though asymptotically better than $\Omega(k)$, still grows in terms of $k$: estimates still become worse even with more samples. Inspired by the "burn-in" technique (Neiswanger et al., 2014) widely employed in MCMC, we show that it is possible to achieve a constant bound via a *non-square* matrix (i.e., relaxing P1). For an arbitrary square matrix $\boldsymbol{X}$, consider its counterpart $\boldsymbol{Y}$ defined with a hyperparameter $q \in (0,1)$, $\boldsymbol{Y}_{(k-kq) \times k} := \boldsymbol{X}(kq+1:k;1:k)$ where the bracket means taking a sub-matrix with the selected rows and columns. In other words, given a $\boldsymbol{X}$, $\boldsymbol{Y}$ is a sub-matrix of $\boldsymbol{X}$ with the $(kq+1)$th row to the last row, and thus a counterpart to $\boldsymbol{X}$ and relaxes P1. The intuition is that in the first few iterations, $t$ is small, causing $\tilde{g}^*_{\pi^t}$ to still incur relatively large variances even with correlated noise via $\boldsymbol{X}$. As a remedy, $\boldsymbol{Y}$ effectively discards these highly fluctuating marginal contributions to yield an asymptotically even lower total variance than $\boldsymbol{X}$.

**Proposition 5.4 (Correlated Noise with $Y$, *informal*).** Let $\tilde{g}_{\pi^l}, l \in [t]$ be perturbed gradients using the Gaussian mechanism that satisfies $(\epsilon, \delta)$-DP. $\forall t \in \{kq + 1, \ldots, k\}$, denote $\boldsymbol{\theta}_{\pi^t}^* := \boldsymbol{\theta}_{\pi^t}^p - \alpha \sum_{l=1}^t \boldsymbol{Y}_{t-kq,l}\tilde{g}_{\pi^l}, m^*(\pi^t) := V(\boldsymbol{\theta}_{\pi^t}^*) - V(\boldsymbol{\theta}_{\pi^t}^p)$, and $\psi^* := (k - kq)^{-1} \sum_{t=kq+1}^k p(\pi^t) m^*(\pi^t)$ for $q \in (0, 1)$. With the same assumption and the suitable matrix $\boldsymbol{X}$ discussed in Prop. 5.3, setting $\forall t \in [k], \forall l \in [t], \boldsymbol{Y}_{t,l} = \boldsymbol{X}_{t,l}$ produces $\mathrm{Var}[\psi^* | \boldsymbol{\theta}_{\pi^1}^p, \boldsymbol{\theta}_{\pi^2}^p, \ldots, \boldsymbol{\theta}_{\pi^k}^p] = \mathcal{O}\left((1 - q)^{-2} \log^2 (1/q) + \sigma_g^4\right)$ and $\mathbb{E}[\psi^* - \psi] = \mathcal{O}((1 - q) \log 1/q + \sigma_g^2)$ while satisfying $(\epsilon, \delta)$-DP.

A formal statement of Prop. 5.4 is in Appendix (Prop. C.11). Prop. 5.4 shows that we can control the estimation uncertainty by a *constant* (i.e., entirely removing the effect of $k$) with a combination of correlated noise and burn-in. Intuitively, the first term represents the injected noise controlled by $q$. In particular, since $kq \in \mathbb{N}^+$, we have $q \geq 1/k$ which implies $\log^2 (1/q) \leq \log^2 k$. As such, when $k \to \infty$ and $q \to 1/k$, the bound is reduced to that in Prop. 5.3. As $q \to 1$, too many evaluation samples are "burnt", leaving insufficient samples to average out the noise as reflected by the exploding first term where $(1 - q)^{-2} \log^2 (1/q) \to \infty$. We show in detail how $q$ affects the estimation uncertainty in Sec. 6.2. Pseudo-code is in Algorithm 1 (in green ). On top of red and blue , green lines exclude the first $kq$ marginal contributions from being included in $\psi$.

# 6 EXPERIMENTS

We fix $C = 1.0$ and $(\epsilon = 1, \delta = 5 \times 10^{-5})$-DP guarantee unless otherwise specified. All experiments are repeated over 5 independent trials. We focus on classification tasks as they are more susceptible to noise and defer regression task to App. D.3. We consider data selection and noisy label detection tasks as standard evaluations of the effectiveness of a data value estimate (Ghorbani & Zou, 2019; Kwon & Zou, 2022; Wang & Jia, 2023; Zhou et al., 2023). Exploiting Prop. 5.3, we set a large $k \geq 200$ (except FL) and use $\boldsymbol{X}^*$ and $\boldsymbol{Y}^*$ for adding correlated noise. Additional experimental settings and results are in App. D.

While our theoretical results have provided a $(\epsilon, \delta)$-DP guarantee level, in App. D.1, we verify the privacy protection of our method by constructing a membership inference attack (MIA) following the setting in (Wang et al., 2023), and demonstrate that our method can successfully defend against the constructed MIA. For experiments in the main text, we focus on how our method improves the data value estimation with privacy protection.

## 6.1 INCREASING $k$ UNDER I.I.D. NOISE *Does Not* REDUCE THE ESTIMATION UNCERTAINTY

We empirically show the scaling estimation uncertainty as the evaluation budget $k$ increases and its implication. As a setup, we randomly choose 400 training examples from the diabetes dataset (Efron et al., 2004) with the remaining data points as the test dataset. We train a logistic regression using the negated cross-entropy loss on the test dataset as the utility function $V$. To evaluate the estimation uncertainty, we first examine the quality of data value estimates through mean-adjusted variance of $\psi_j$ (Zhou et al., 2023), which is the ratio between the empirical variance $s_j^2 := k^{-1}(k-1)^{-1} \sum_{t=1}^k [m_j(\pi^t) - \mu_j]^2$ and the empirical mean $\mu_j := k^{-1} \sum_{t=1}^k m_j(\pi^t)$. We also examine how test accuracy changes when removing training examples with the highest $\psi$'s. The leftmost figure of Fig. 2 shows that the mean-adjusted variance $s_{\text{i.i.d.}}^2/|\mu_{\text{i.i.d.}}|$ with i.i.d. noise increases with $k$, indicating greater estimation uncertainty, whereas using correlated noise, $s_{\text{corr.}}^2/|\mu_{\text{corr.}}|$ not only decreases but is also smaller by several magnitudes (in $10^5$). Moreover, $\mu_{\text{i.i.d.}}$'s computed with i.i.d. noise are increasingly negative as $k$ increases, whereas $\mu_{\text{corr.}}$'s computed with correlated noise stay positive, suggesting that the estimated $\psi$'s are less affected by noise. The 3rd figure of Fig. 2 shows that $\psi$'s computed with greater $k$ produces higher test accuracy during removal (i.e., closer to random removal), verifying our identified paradox that $\psi$'s computed with more budget are poorer estimates of data values.

## 6.2 CORRELATED NOISE IMPROVES THE QUALITY OF THE ESTIMATES

Following the same setup as Sec. 6.1, we compute the $\psi$'s using correlated noise (with $q = 0.8$) and no injected noise due to DP respectively. The results are shown in the rightmost figure of Fig. 2. Removing data points with high $\psi$'s computed with no injected noise produces low test accuracy

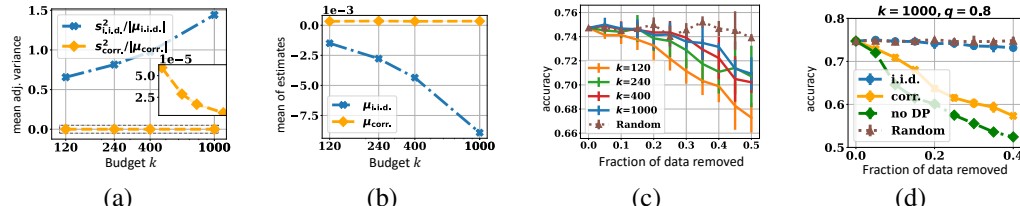

(a)        (b)        (c)        (d)

Figure 2: (a) $n^{-1} \sum_{j \in [n]} s_j^2 / |\mu_j|$ and (b) $\mu_j$ vs. $k$ using i.i.d. noise and correlated noise. (c) error bars of test accuracy vs. ratio of data removed with *the highest* $\psi$'s using different $k$ with i.i.d. noise and (d) also with correlated noise and no DP. "Random" means random removal.

close to that with correlated noise. In contrast, the curve of $\psi$'s computed with i.i.d. noise lies far above (i.e., close to random removal), suggesting that $\psi$'s computed with correlated noise are much more reflective of the true worth of data as compared to $\psi$'s computed with i.i.d. noise.

**Ablation study on the influence of** $q$**.** We study how much $q$ affects the data value estimation in a noisy label detection setting on two datasets. We randomly perturb $30\%$ of labels on a selection of 800 training examples from Covertype dataset (Blackard, 1998) (and MNIST dataset (LeCun et al., 1990) in App. D.4) respectively. The datasets are trained with logistic regression (LR) and a more complex convolutional network (CNN). Ideally, a good data value estimate should assign the lowest $\psi$'s to the perturbed training examples. To measure this, we plot the AUC-ROC curve (AUC) in Fig. 3 with $q = 0$ equivalent to $\boldsymbol{X}^*$. With increased $k$, the AUC with our method *increases* especially in the large-$q$ region while the AUC with i.i.d. noise *decreases*. We also observe that AUC generally increases with $q$ when $q \lessapprox 0.9$.

**Adopting other utility functions.** While our theoretical analysis is w.r.t. negated loss as the utility function $V$, we demonstrate similar results with test accuracy as $V$, shown in the right column of Fig. 3. Our method still outperforms i.i.d. noise, although the AUC is lower than when $V$ is negated loss. We think this is because accuracy is a less fine-grained metric. Hence subtle changes in the model performance (often due to perturbation) in this task cannot be well reflected. Moreover, we observe that as $q \to 1$, AUC degrades, consistent with the result in Prop. 5.4.

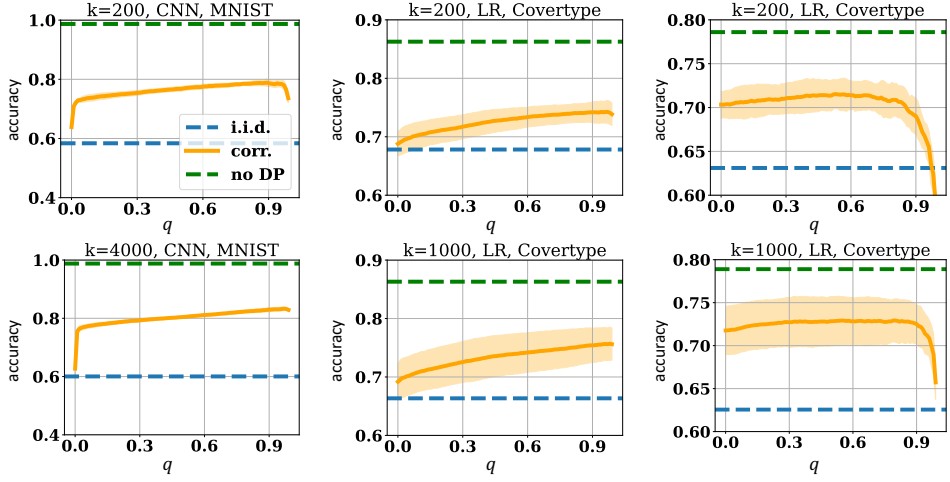

Figure 3: Plots of AUC v.s. burn-in ratio $q \in [0, 1)$ (with $q = 0$ equivalent to $\boldsymbol{X}^*$). $V$ is (left and middle) negated test loss and (right) test accuracy. Lines represent mean and shades represent 1 standard deviation. Higher is better.

**Experiments on other semivalues.** We compare the effect of utilizing correlated noise with data Banzhaf (Wang & Jia, 2023) and Beta Shapley (Kwon & Zou, 2022) in Table 2. We consider the same setup as Sec. 6.2 with $k = 1000$. We compare the performance using $\boldsymbol{X}^*$ and $\boldsymbol{Y}^*$ with

Table 2: Mean (std. errors) of AUC on Covertype trained with LR (top) and MNIST trained with CNN (bottom). The best score is highlighted. Higher is better.

| semivalues | no DP | i.i.d. noise | $X^*$ | $Y^*$ ($q = 0.5$) | $Y^*$ ($q = 0.9$) |
|---|---|---|---|---|---|
| Shapley | 0.905 (1.00e-03) | 0.675 (6.00e-03) | 0.735 (2.00e-02) | 0.774 (8.00e-03) | **0.788 (4.00e-03)** |
| Banzhaf | 0.896 (2.00e-03) | 0.533 (1.70e-02) | 0.725 (2.00e-02) | 0.770 (8.00e-03) | **0.777 (4.00e-03)** |
| Beta(4, 1) | 0.882 (1.00e-03) | 0.612 (1.40e-02) | 0.721 (2.10e-02) | 0.766 (8.00e-03) | **0.777 (4.00e-03)** |
| Beta(16, 1) | 0.875 (0.00e+00) | 0.557 (2.40e-02) | 0.707 (2.00e-02) | 0.757 (8.00e-03) | **0.767 (5.00e-03)** |
| Shapley | 0.988 (0.00) | 0.600 (1.50e-02) | 0.627 (2.30e-02) | 0.803 (4.00e-03) | **0.828 (6.00e-03)** |
| Banzhaf | 0.985 (1.00e-03) | 0.532 (2.00e-02) | 0.616 (2.30e-02) | 0.808 (4.00e-03) | **0.827 (7.00e-03)** |
| Beta(4, 1) | 0.991 (0.00) | 0.569 (1.50e-02) | 0.618 (2.10e-02) | 0.789 (4.00e-03) | **0.810 (7.00e-03)** |
| Beta(16, 1) | 0.992 (0.00) | 0.539 (1.00e-02) | 0.615 (1.10e-02) | 0.773 (5.00e-03) | **0.793 (9.00e-03)** |

Table 3: Mean (std. errors) of $\Delta_{\text{i.i.d.}}$ and $\Delta_{\text{corr.}}$ for dataset valuation (top) and FL (bottom). Best scores are highlighted (lower is better). $B$ is the dataset size.

| ML Task | $\{k, n, B\}$ | $\Delta_{\text{i.i.d.}}$ | $\Delta_{\text{corr.}}$ (Ours) |
|---|---|---|---|
| MNIST + CNN | $\{200, 800, 8\}$ | 0.290 (1.63e-02) | **0.170 (3.73e-02)** |
| CIFAR10 + CNN | $\{1000, 100, 32\}$ | 0.178 (5.85e-02) | **0.0303 (1.53e-02)** |
| CIFAR10 + ResNet18 | $\{1000, 100, 32\}$ | 0.204 (5.33e-02) | **0.119 (1.73e-02)** |
| CIFAR10 + ResNet34 | $\{1000, 100, 32\}$ | 0.141 (8.51e-03) | **0.0433 (7.10e-03)** |
| MNIST + CNN | $\{50, 50, 32\}$ | 0.195 (9.40e-03) | **0.0810 (8.06e-03)** |
| CIFAR10 + CNN | $\{50, 50, 32\}$ | 0.167 (3.96e-02) | **0.0415 (9.63e-03)** |

$q \in \{0.5, 0.9\}$. For all variants, the AUC is higher than that with i.i.d. noise. Particularly, $q = 0.9$ works the best for all 4 semivalues tested. In contrast, the AUCs with i.i.d. noise using data Banzhaf are $\approx 0.53$ on both datasets, *close to randomness (*0.5*)*. The results show that our method generalizes to various data valuation metrics. We also note a similar improvement for LOO (Cook, 1977) in App. D.4 despite it not being a regular semivalue.

### 6.3 APPLICATION TO OTHER USE CASES

We verify the effectiveness of using correlated noise for dataset valuation (Wu et al., 2022) and collaborator attribution in federated learning (Wang et al., 2020) (refer to App. D.2 for setup). We consider MNIST dataset and CIFAR10 (Krizhevsky et al., 2012) dataset, trained on CNN and fine-tuned on pretrained ResNet18/34 (He et al., 2015). We tabulate the difference in AUC between using correlated noise with $q = 0.9$, denoted as $\Delta_{\text{corr.}} \coloneqq \text{AUC}_{\text{no DP}} - \text{AUC}_{\text{corr.}}$, and using i.i.d. noise, denoted as $\Delta_{\text{i.i.d.}} \coloneqq \text{AUC}_{\text{no DP}} - \text{AUC}_{\text{i.i.d.}}$. For ResNet34, we use $\epsilon = 10$ as the model is more complex, causing both i.i.d. and our method to have degraded performance with strict privacy. Our methods outperform i.i.d. noise as shown in Table 3 top. For FL, we notice that the continually updating characteristic of the global model poses two challenges: (i) the overall scale of the loss values decreases in each round as the model gradually converges such that marginal contributions computed in later rounds are less significant, and (ii) the variance of gradients $\sigma_g^2$ is larger than in common data valuation scenarios. To tackle these challenges, we adopt test accuracy as $V$ so that $V$ are in the same scale in each round, use a matrix with $X_{t,t} > 1/t$ to control $\sigma_g^2$, and choose a small burn-in ratio $q = 0.2$ to keep more evaluations in the first few rounds (detailed in App. D.2). The results with these modifications are shown in Table 3 bottom. Our method outperforms i.i.d. noise by a large margin, showing the applicability of our approach beyond the default data valuation setting under which our theoretical results are developed.

## 7 CONCLUSION, LIMITATIONS, AND FUTURE WORKS

In this work, we identify a problem in data valuation where DP is enforced via perturbing gradients with i.i.d. noise: The estimation uncertainty scales linearly (i.e., $\Omega(k)$) with more budget $k$ and renders the data value estimates almost useless (i.e., close to random guesses in some investigated cases). As a solution, we propose to use correlated noise and theoretically show that using a weighted sum via matrix form using $X$ provably reduces the estimation uncertainty of semivalues from $\Omega(k)$ to $\mathcal{O}(1)$ and empirically demonstrate the implications of our method on various ML tasks and data valuation metrics. One limitation is the need to store the gradients. However, this limitation is alleviated when the number of parties is small (e.g. dataset valuation) or when the memory load can be distributed across parties (e.g. FL). Another limitation is that our theoretical result assumes an diagonal multivariate sub-Gaussian distribution of the gradients. Nevertheless, we empirically demonstrate that our method works for neural networks where the assumption is not explicitly satisfied. A future direction is to explore other possible $X$ to reduce estimation uncertainty further.

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

# APPENDIX

# A TABLE OF NOTATIONS

Table 4: Notations. The subscript $j$ is omitted where the context is clear.

| Notations | Definitions | Interpretation |
|---|---|---|
| $[n]$ | $[n] := \{1, 2, \ldots, n\}$ | the set of numbers indexed from 1 to $n$ |
| $\Pi$ | $\Pi := \mathrm{Perm}([n])$ | the set of all permutations of $[n]$ parties |
| $\pi^t$ | $\pi^t \sim \mathrm{Unif}(\Pi)$ | a sample of permutation drawn uniformly from $\Pi$ at the $t$th iteration |
| $\hat{g}_{\pi^t,j}, \hat{g}_{\pi^t}$ | $\hat{g}_{\pi^t,j} \approx g_{\pi^t,j}$ | the (clipped) gradient (of party $j$) computed at the $t$th permutation ($\pi^t \in \Pi$) |
| $\tilde{g}_{\pi^t,j}, \tilde{g}_{\pi^t}$ | $\tilde{g}_{\pi^t,j} := \hat{g}_{\pi^t,j} + z_t$ | the perturbed gradient with Gaussian mechanism $z_t$ at the $t$th iteration |
| $\tilde{g}^*_{\pi^t,j}, \tilde{g}^*_{\pi^t}$ | $\tilde{g}^*_{\pi^t,j} := \hat{g}_{\pi^t,j} + z^*$ | the perturbed gradient computed with correlated noise $z^*$ (realized via $\boldsymbol{X}$ or $\boldsymbol{Y}$) |
| $P^\pi_j$ | N.A. | the set of all precedents of party $j$ in permutation $\pi$ |
| $\pi[[P^\pi_j]]$ | N.A. | the index of the party immediately before $j$ in permutation $\pi$ |
| $w(s)$ | N.A. | weight coefficient of marginal contribution with cardinality $s$ |
| $p_j(\pi), p(\pi)$ | $p_j(\pi) := nw(|P^\pi_j| + 1)$ | the weight coefficient of the marginal contribution made by party $j$ at permutation $\pi$ |
| $\boldsymbol{\theta}^p_{\pi^t,j}, \boldsymbol{\theta}^p_{\pi^t}$ | $\boldsymbol{\theta}^p_{\pi^t,j} := \boldsymbol{\theta}_{\pi^t, \pi^t[[P^\pi_j]]}$ | the model parameters immediately before updated with party $j$'s gradients at permutation $\pi^t$ |
| $\boldsymbol{\theta}_{\pi^t,j}, \boldsymbol{\theta}_{\pi^t}$ | $\boldsymbol{\theta}_{\pi^t,j} := \boldsymbol{\theta}^p_{\pi^t,j} - \alpha\hat{g}_{\pi^t,j}$ | model parameters computed after updated with party $j$'s gradients at iteration $t$ |
| $V(\boldsymbol{\theta}_{\pi^t,j})$ | $V(\boldsymbol{\theta}_{\pi^t,j}) := V(P^{\pi^t}_j \cup \{j\})$ | the utility of the model updated with gradients of party $j$'s gradients at permutation $\pi^t$ |
| $m_j(\pi^t), m(\pi^t)$ | $m_j(\pi^t) := V(\boldsymbol{\theta}_{\pi^t,j}) - V(\boldsymbol{\theta}^p_{\pi^t,j})$ | marginal contribution (of party $j$) at permutation $\pi^t$ |
| $m^*_j(\pi^t), m^*(\pi^t)$ | depends on implementation | marginal contribution (of party $j$) with correlated noise at permutation $\pi^t$ |
| $\phi_j, \phi$ | $\phi_j := \mathbb{E}_{\pi \in \Pi}[p_j(\pi) m_j(\pi)]$ | the exact semivalue |
| $\psi_j, \psi$ | $\psi_j := t^{-1} \sum_{t=1}^k m_j(\pi^t)$ | a semivalue estimate (of party $j$) with i.i.d. noise in $k$ iterations |
| $\psi^*_j, \psi^*$ | depends on implementation | a semivalue estimate (of party $j$) with correlated noise in $k$ iterations |

# B ADDITIONAL DISCUSSIONS

## B.1 SEMIVALUE AS A RANDOM VARIABLE.

The exact computation of semivalues is often intractable in practice due to the need to compute an exponential number of marginal contributions $V(S \cup \{i\}) - V(S)$. Moreover, in data valuation, the utility function $V$ is not deterministic w.r.t. $S$: $V$ is commonly defined as the test accuracy or test loss (Ghorbani & Zou, 2019), which is stochastic due to (stochastic) gradient descent and random model initialization. Such stochasticity becomes more pronounced when gradients are injected with artificial noise to ensure privacy such as in DP-SGD (Abadi et al., 2016) (recalled later). Hence, the semivalue estimator $\psi_i$ is treated as a *random variable* whose randomness comes from the marginal contributions.

## B.2 DIFFERENTIAL PRIVACY FRAMEWORK.

We emphasize that our analysis adopts this definition as we rely on the Gaussian mechanism and leverage its composition property (Dwork & Roth, 2014). Our analysis holds for other DP frameworks capturing privacy guarantee with Gaussian mechanism and possessing composition and post-processing properties, such as Rényi DP (Mironov, 2017) and $z$-CDP (Bun & Steinke, 2016). Typically, both Rényi DP and $z$-CDP satisfy post-processing immunity and have composition properties. Moreover, both provide a DP guarantee for the Gaussian mechanism. Notably, while both DP frameworks provide a modestly tighter privacy analysis than compared to (Dwork & Roth, 2014), to the best of our knowledge, they still require the variance of the Gaussian noise to linearly scale with $k$ to satisfy a given level of DP guarantee. In fact, some existing works have identified that the moment accountant method (Abadi et al., 2016) which we adopt is an instantiation of Rényi DP (Ouadrhiri & Abdelhadi, 2022) translated back to $(\epsilon, \delta)$-DP. Lastly, we note that our theoretical analysis is independent of the choice of DP framework except for requiring $z_t \sim \mathcal{N}(\mathbf{0}, k\sigma^2 \boldsymbol{I})$.

**Gaussian mechanism.** We note that other mechanisms exist for achieving privacy with DP guarantees. We choose the Gaussian mechanism for the convenience of mathematical analysis compared to other mechanisms such as the Laplace mechanism as well as a manifold of existing works discussing the theoretical properties of the Gaussian mechanism.

### B.3 COMPOSITION OF DP MECHANISMS.

While it is possible to find a theoretical privacy budget lower than the result in (Abadi et al., 2016, Theorem 1), the current asymptotic bound $\sigma^2 = \Omega(k)$ is the best we know. We also note that there have been efforts at improving the bound (Asoodeh et al., 2021). However, the results are not asymptotically better in terms of $k$, hence do not affect our theoretical results. Moreover, we emphasize that our theoretical analysis and proof strategy do not depend on the exact form of $\sigma^2$ other than assuming it depends on $k$. If a lower bound is found in the future, our analysis can be readily adapted.

**Note on per evaluation DP guarantee and final DP guarantee.** We are concerned about the final DP guarantee (i.e., $(\epsilon, \delta)$-DP guarantee after $k$ compositions) unless otherwise stated. Notably, by setting $z_t \sim \mathcal{N}(k\sigma^2 \boldsymbol{I})$, each per-evaluation DP-guarantee is stronger, or put differently, the privacy budget required at each evaluation to satisfy a given $(\epsilon, \delta)$-DP guarantee is lowered such that, after $k$ compositions, the desired final DP-guarantee can be achieved.

### B.4 DP GUARANTEE LEVEL AND DATA VALUE ESTIMATES

**Noise introduced by DP causes a lower mean of data value estimates.** Mathematically, DP imposes a limit on the leave-one-out property of the privatized mechanism $\mathcal{M}$. In particular, if $V(\cdot) \in [0, 1]$ (e.g., when test accuracy is used as $V$), then we have, for any party $i$ and subsets $S \subseteq [n] \setminus \{i\}$, the following (Dwork et al., 2015, Lemma 6):

$$|\psi_i| := |\mathbb{E}_S[V(S \cup \{i\}) - V(S)]| \leq e^\epsilon - 1 + \delta$$

where $(\epsilon, \delta)$ are the parameters satisfying $(\epsilon, \delta)$-DP. Assuming a fixed $\delta$, stronger DP implies a lower $\epsilon$, hence decreased right-hand side value, i.e., the upper bound for the marginal contribution. This inequality suggests that stronger DP results in lower *absolute value* of the data value estimates. We find this a reasonable behavior as the decreased value reflects the erosion of information carried by the data. Instead, having a lower absolute data value does not forbid us to still preserve the relative order of the estimates by minimizing the estimation uncertainty.

**Impact of different $(\epsilon, \delta)$-DP guarantee levels on our method.** Through our theoretical development, we have established that the estimation uncertainty can be controlled to be *independent* of the evaluation budget $k$, i.e., the noise due to DP is only affected by the final DP guarantee level. That said, an overly large final DP guarantee level shrinks the gap between the performance of i.i.d. noise and without DP noise since perturbation is small, and *vice versa*. Therefore, to highlight the effectiveness of our method in controlling the estimation uncertainty, we fix a moderate final DP guarantee level $(\epsilon, \delta) = (1.0, 5 \times 10^{-5})$ throughout the main text s.t. a clear contrast in performance on various ML tasks can be observed. We also provide additional experiments on different $\epsilon$ values in App. D.5.

**Computing $\text{Var}[\psi_j]$.** One can derive $\text{Var}[\psi_j]$ given $\text{Var}[\psi_j | \boldsymbol{\theta}^p_{\pi^1, j}, \ldots, \boldsymbol{\theta}^p_{\pi^k, j}]$ via the law of total variance and assumptions on the inter-dependence between $\boldsymbol{\theta}^p_{\pi^t, j}$'s and $k$, which is not the focus of this work – how DP impacts data valuation, particularly semivalue estimation – and thus left for future work.

### B.5 SOCIETAL IMPACT

As discussed in the introduction in the main text, we believe our contribution has a huge potential societal impact in improving privacy, especially with the rising awareness of protecting personal data (Bukaty, 2019; Council of European Union, 2014). We do not find a direct path to any negative societal impact with our contribution.

## C    PROOFS AND ADDITIONAL RESULTS

### C.1    PROOF OF THE EXAMPLE IN SEC. 5.2.

**Observation C.1.** For a particular party $j$, assume, for $t \in [k]$, the norm-clipped gradients $\hat{g}_{\pi^1,j} = \hat{g}_{\pi^2,j} = \ldots = \hat{g}_{\pi^k,j}$. Denote $\forall t \in [k], \tilde{g}_{\pi^t,j} := \hat{g}_{\pi^t,j} + z_t$ where $z_t \overset{\text{i.i.d.}}{\sim} \mathcal{N}(\mathbf{0}, k(C\sigma)^2\mathbf{I})$ are i.i.d. drawn. Let $\tilde{g}^*_{\pi^t,j} := t^{-1}\sum_{l=1}^{t}\tilde{g}_{\pi^l,j}$. Then, $\mathrm{Var}[\|\tilde{g}^*_{\pi^t,j}\|_2^2] \to t^{-2}\mathrm{Var}[\|\tilde{g}_{\pi^t,j}\|_2^2]$ as $k \to \infty$.

*Proof.* We make the following notations to help ease understanding. Denote $z_t^* := t^{-1}\sum_{l=1}^{t}z_l \sim \mathcal{N}(\mathbf{0}, ((C\sigma)^2/t)\mathbf{I})$. Denote $\hat{g} := \hat{g}_{\pi^1,j} = \hat{g}_{\pi^2,j} = \ldots = \hat{g}_{\pi^k,j}$. Further denote $(\cdot)_r$ the $r$th element of a vector. Then, we have

$$\mathrm{Var}[\|\tilde{g}^*_{\pi^t,j}\|_2^2] = \mathrm{Var}[\|\hat{g} + z_t^*\|_2^2]$$

$$= \mathrm{Var}[\sum_{r=1}^{d}(\hat{g} + z_t^*)_r^2]$$

$$= \sum_{r=1}^{d}\mathrm{Var}[(\hat{g} + z_t^*)_r^2]$$

$$= \sum_{r=1}^{d}(2\hat{g})_r^2\mathrm{Var}[(z_t^*)_r] + \sum_{r=1}^{d}\mathrm{Var}[(z_t^*)_r^2] + 2\sum_{r=1}^{d}\mathrm{Cov}[2(\hat{g})_r(z_t^*)_r, (z_t^*)_r^2]$$

$$= 4\|\hat{g}\|_2^2\mathrm{Var}[(z_t^*)_1] + d\mathrm{Var}[(z_t^*)_1^2] + 2\sum_{r=1}^{d}\left(\mathbb{E}[2(\hat{g})_r(z_t^*)_r^3] - \mathbb{E}[2(\hat{g})_r(z_t^*)_r]\mathbb{E}[(z_t^*)_r^2]\right)$$

$$= 4\|\hat{g}\|_2^2k(C\sigma)^2/t + 2dk^2(C\sigma)^4/t^2$$

where in the last step we use the fact $\mathbb{E}[z] = \mathbb{E}[z^3] = 0$ if $z$ follows a Normal distribution with mean 0 as well as the fact that $(t/\sigma^2)(z_t^*)_1^2 \sim \chi_1^2$ where $\chi_1^2$ is a chi-squared distribution with degree of freedom 1. On the other hand,

$$\mathrm{Var}[\|\tilde{g}_{\pi^t,j}\|_2^2] = \mathrm{Var}[\|\hat{g} + z_t\|_2^2]$$

$$= \sum_{r=1}^{d}\mathrm{Var}[(\hat{g} + z_t)_r^2]$$

$$= 2\sum_{r=1}^{d}(2\hat{g})_r\mathrm{Var}[(z_t)_r] + \sum_{r=1}^{d}\mathrm{Var}[(z_t)_r^2] + 2\sum_{r=1}^{d}\mathrm{Cov}[2(\hat{g})_r(z_t)_r, (z_t)_r^2]$$

$$= 4\|\hat{g}\|_2^2\mathrm{Var}[(z_t)_1] + d\mathrm{Var}[(z_t)_1^2] + 2\sum_{r=1}^{d}\left(\mathbb{E}[2(\hat{g})_r(z_t)_r^3] - \mathbb{E}[2(\hat{g})_r(z_t)_r]\mathbb{E}[(z_t)_r^2]\right)$$

$$= 4\|\hat{g}\|_2^2k(C\sigma)^2 + 2dk^2(C\sigma)^4$$

where the last step follows the same logic as the last equation. We can easily verify that

$$\lim_{k\to\infty}\frac{\mathrm{Var}[\|\tilde{g}_{\pi^t,j}\|_2^2]}{\mathrm{Var}[\|\tilde{g}^*_{\pi^t,j}\|_2^2]} = \frac{2d(C\sigma)^4}{2d(C\sigma)^4/t^2} = t^2 \ .$$

$\square$

### C.2    PROOF OF PROP. 5.1.

We first establish the following lemma to facilitate the proof.

**Lemma C.2.** If random variables $X$ and $Y$ are independent, then

$$\mathrm{Var}[XY] \geq \mathbb{E}[Y]^2\mathrm{Var}[X] \ .$$

*Proof.* Since $X$ and $Y$ are independent, $X|Y = X$ and $Y|X = Y$. Then, by the law of total variance, we have

$$
\begin{aligned}
\text{Var}[XY] &= \mathbb{E}[\text{Var}[XY|X]] + \text{Var}[\mathbb{E}[XY|X]] \\
&= \mathbb{E}[X^2\text{Var}[Y|X]] + \text{Var}[X\mathbb{E}[Y|X]] \\
&= \mathbb{E}[X^2\text{Var}[Y]] + \text{Var}[X\mathbb{E}[Y]] \\
&= \mathbb{E}[Y]^2\text{Var}[X] + \text{Var}[Y]\mathbb{E}[X^2] \\
&\geq \mathbb{E}[Y]^2\text{Var}[X] \ .
\end{aligned}
$$

$\square$

**Proposition C.3** (Reproduced from Prop. 5.1.). $\forall t \in [k]$, denote $\boldsymbol{\theta}_{\pi^t} := \boldsymbol{\theta}_{\pi^t}^p - \alpha\tilde{g}_{\pi^t} = \boldsymbol{\theta}_{\pi^t}^p - \alpha(\hat{g}_{\pi^t} + z_t)$ where $\forall t \in [k]$, $z_t \overset{\text{i.i.d.}}{\sim} \mathcal{N}(\mathbf{0}, k(C\sigma)^2\boldsymbol{I})$ with $\sigma, C > 0$. Given a test dataset consisting of $l$ data points $\mathcal{D}_{\text{test}} = \{(\boldsymbol{x}_1, y_1), \ldots, (\boldsymbol{x}_l, y_l)\}$. If $V$ is the negated mean-squared error loss on a linear regression model

$$V(\boldsymbol{\theta}) := -l^{-1}\sum_{i=1}^{l}(\boldsymbol{\theta}^\top\boldsymbol{x}_i - y_i)^2$$

or the negated $\ell_2$-regularized cross-entropy loss on a logistic regression model

$$
\begin{aligned}
V(\boldsymbol{\theta}) := &l^{-1}\sum_{i=1}^{l}(1 - y_i)\log(1 - \text{Sig}(\boldsymbol{\theta}, \boldsymbol{x}_i)) \\
&+ l^{-1}\sum_{i=1}^{l}y_i\log(\text{Sig}(\boldsymbol{\theta}, \boldsymbol{x}_i)) - \lambda\|\boldsymbol{\theta}\|_2^2
\end{aligned}
$$

where $\text{Sig}(\boldsymbol{\theta}, \boldsymbol{x}_i) = (1 + e^{-\boldsymbol{\theta}^\top\boldsymbol{x}_i})^{-1}$ is the sigmoid function and $\lambda > 0$ is the regularization hyperparameter. Denote $m(\pi^t) := V(\boldsymbol{\theta}_{\pi^t}) - V(\boldsymbol{\theta}_{\pi^t}^p)$. Further denote a regular semivalue estimator $\psi := k^{-1}\sum_{t=1}^{k}p(\pi^t)m(\pi^t)$. [3] The estimation uncertainty satisfies $\text{Var}[\psi|\boldsymbol{\theta}_{\pi^1}^p, \boldsymbol{\theta}_{\pi^2}^p, \ldots, \boldsymbol{\theta}_{\pi^k}^p] = \Omega(k)$.

*Proof of Prop. 5.1.* We first analyze the variance of $z_t$ on each data point in $\mathcal{D}_{\text{test}}$. Note that $\boldsymbol{\theta}_{\pi^t,j} = \boldsymbol{\theta}_{\pi^t,j}^p - \alpha(\hat{g}_{\pi^t,j} + z_t) = (\boldsymbol{\theta}_{\pi^t,j}^p - \alpha\hat{g}_{\pi^t,j}) - \alpha z_t$. Denote $\bar{\boldsymbol{\theta}}_{\pi^t,j} := \boldsymbol{\theta}_{\pi^t,j}^p - \alpha\hat{g}_{\pi^t,j}$. Notice that, conditional on $\boldsymbol{\theta}_{\pi^t,j}^p$, $\bar{\boldsymbol{\theta}}_{\pi^t,j}$ can be *deterministically calculated* via gradient computation since the underlying data is fixed. Then we have $\boldsymbol{\theta}_{\pi^t,j} = \bar{\boldsymbol{\theta}}_{\pi^t,j} - \alpha z_t$ where the randomness of $\boldsymbol{\theta}_{\pi^t,j}$ arises from the randomness of $z_t$. Denote Denote $\underline{p} := \min_{\pi \in \Pi} p_j(\pi) > 0$ the minimal weight, which is positive since $\psi_j$ is a regular semivalue. Due to the independence of all $z_t$'s, we first consider the variance of utility

$$
\begin{aligned}
\text{Var}\left[\frac{\sum_{t=1}^{k}p_j(\pi^t)V(\boldsymbol{\theta}_{\pi^t,j})}{k}\middle|\boldsymbol{\theta}_{\pi^1,j}^p, \ldots, \boldsymbol{\theta}_{\pi^k,j}^p\right] &= \frac{\sum_{t=1}^{k}\text{Var}[p_j(\pi^t)V(\boldsymbol{\theta}_{\pi^t,j})|\boldsymbol{\theta}_{\pi^t,j}^p]}{k^2} \\
&\geq \frac{\sum_{t=1}^{k}\mathbb{E}[p_j(\pi^t)|\boldsymbol{\theta}_{\pi^t,j}^p]^2\text{Var}[V(\boldsymbol{\theta}_{\pi^t,j})|\boldsymbol{\theta}_{\pi^t,j}^p]}{k^2} \\
&\geq \frac{\sum_{t=1}^{k}\underline{p}^2\text{Var}[V(\bar{\boldsymbol{\theta}}_{\pi^t,j} - \alpha z_t)|\boldsymbol{\theta}_{\pi^t,j}^p]}{k^2} \\
&= \frac{\sum_{t=1}^{k}\underline{p}^2\text{Var}[V(\bar{\boldsymbol{\theta}}_{\pi^t,j} - \alpha z_t)|\bar{\boldsymbol{\theta}}_{\pi^t,j}]}{k^2}
\end{aligned}
$$

where the 2nd step is derived from Lemma C.2 since $p_j$ and $V$ are independent. We analyze the conditional variance w.r.t. the two loss functions respectively in the following.

**Linear regression.** $V$ here is the negated MSE on $\mathcal{D}_{\text{test}}$. The above variance is further analyzed w.r.t. each single test data point $(\boldsymbol{x}_i, y_i) \in \mathcal{D}_{\text{test}}$ as

$$
\begin{aligned}
\text{Var}[V(\bar{\boldsymbol{\theta}}_{\pi^t,j} - \alpha z_t)|\bar{\boldsymbol{\theta}}_{\pi^t,j}] &= \text{Var}[-\frac{1}{l}\sum_{i=1}^{l}((\bar{\boldsymbol{\theta}}_{\pi^t,j} - \alpha z_t)^\top\boldsymbol{x}_i - y_i)^2|\bar{\boldsymbol{\theta}}_{\pi^t,j}] \\
&= \frac{1}{l^2}\sum_{i=1}^{l}\text{Var}[(\bar{\boldsymbol{\theta}}_{\pi^t,j} - \alpha z_t)^\top\boldsymbol{x}_i - y_i)^2|\bar{\boldsymbol{\theta}}_{\pi^t,j}] \ .
\end{aligned}
$$

---

[3] A regular semivalue has $w(s) > 0$ for all $s \in [n]$ (i.e., $p(\pi) > 0$ for all $\pi \in \Pi$) (Carreras & Freixas, 2002). Examples of regular semivalues include Shapley value, Banzhaf value, and Beta Shapley.

Note that $\forall t \in [k]$ and $\forall i \in [l]$, the parameters $\bar{\boldsymbol{\theta}}_{\pi^t,j}$, test data point feature $\boldsymbol{x}_i$, and test data point label $y_i$ are fixed conditional on $\bar{\boldsymbol{\theta}}_{\pi^t,j}$. Hence, $(\bar{\boldsymbol{\theta}}_{\pi^t,j} - \alpha z_t)^\top \boldsymbol{x}_i - y_i$ is an affine transformation of a normally distributed random variable $z_t$:

$$(\bar{\boldsymbol{\theta}}_{\pi^t,j} - \alpha z_t)^\top \boldsymbol{x}_i - y_i | \bar{\boldsymbol{\theta}}_{\pi^t,j} \sim \mathcal{N}(\bar{\boldsymbol{\theta}}_{\pi^t,j}^\top \boldsymbol{x}_i - y_i, \alpha^2 k \sigma^2 \boldsymbol{x}_i^\top \boldsymbol{x}_i) ,$$

the square of which produces a noncentral chi-squared random variable with 1 degree of freedom

$$((\bar{\boldsymbol{\theta}}_{\pi^t,j} - \alpha z_t)^\top \boldsymbol{x}_i - y_i)^2 | \bar{\boldsymbol{\theta}}_{\pi^t,j} = \lambda_i \chi_1^2(\mu_{i,t}^2/\lambda_i)$$

where $\lambda_i = \alpha^2 k \sigma^2 \boldsymbol{x}_i^\top \boldsymbol{x}_i$ and $\mu_{i,t} = \bar{\boldsymbol{\theta}}_{\pi^t,j}^\top \boldsymbol{x}_i - y_i$.

Then, use the closed-form expression for the variance of a noncentral chi-squared random variable,

$$\text{Var}[(\bar{\boldsymbol{\theta}}_{\pi^t,j} - \alpha z_t)^\top \boldsymbol{x}_i - y_i)^2 | \bar{\boldsymbol{\theta}}_{\pi^t,j}] = \text{Var}[\lambda_i \chi_1^2(\mu_{i,t}^2) | \bar{\boldsymbol{\theta}}_{\pi^t,j}] = 2\lambda_i^2(1 + 2\mu_{i,t}^2/\lambda_i) = 2\lambda_i^2 + 4\mu_{i,t}^2\lambda_i .$$

Denote $\beta := 2\sum_{i=1}^l \alpha^4 \sigma^4 (\boldsymbol{x}_i^\top \boldsymbol{x}_i)^2$ and $\gamma_t := 4\alpha^2 \sigma^2 \sum_{i=1}^l (\boldsymbol{x}_i^\top \boldsymbol{x}_i)(\bar{\boldsymbol{\theta}}_{\pi^t,j}^\top \boldsymbol{x}_i - y_i)^2$. Note that $\beta$ and $\gamma_t$ are constants conditional on $\bar{\boldsymbol{\theta}}_{\pi^t,j}$. We may rewrite them as $\beta = \frac{1}{k^2}\sum_{i=1}^l 2\lambda_i^2$ and $\gamma_t = \frac{1}{k}\sum_{i=1}^l 4\mu_{i,t}^2\lambda_i$. With these, the variance of $V$ is

$$\text{Var}[V(\bar{\boldsymbol{\theta}}_{\pi^t,j} - \alpha z_t) | \bar{\boldsymbol{\theta}}_{\pi^t,j}] = \text{Var}[-\frac{1}{l}\sum_{i=1}^l ((\bar{\boldsymbol{\theta}}_{\pi^t,j} - \alpha z_t)^\top \boldsymbol{x}_i - y_i)^2 | \bar{\boldsymbol{\theta}}_{\pi^t,j}]$$

$$= \frac{1}{l^2}\sum_{i=1}^l \text{Var}[(\bar{\boldsymbol{\theta}}_{\pi^t,j} - \alpha z_t)^\top \boldsymbol{x}_i - y_i)^2 | \bar{\boldsymbol{\theta}}_{\pi^t,j}]$$

$$= \frac{\beta}{l^2}k^2 + \frac{\gamma_t}{l^2}k .$$

With this, we have the lower bound of the total variance of $V$ as

$$\text{Var}\left[\frac{\sum_{t=1}^k p_j(\pi^t)V(\boldsymbol{\theta}_{\pi^t,j})}{k} \middle| \boldsymbol{\theta}_{\pi^1,j}^p, \ldots, \boldsymbol{\theta}_{\pi^k,j}^p\right] \geq \frac{\sum_{t=1}^k \underline{p}^2 \text{Var}[V(\boldsymbol{\theta}_{\pi^t,j}) | \boldsymbol{\theta}_{\pi^t,j}^p]}{k^2}$$

$$= \frac{\sum_{t=1}^k \underline{p}^2 \text{Var}[V(\bar{\boldsymbol{\theta}}_{\pi^t,j} - \alpha z_t) | \bar{\boldsymbol{\theta}}_{\pi^t,j}]}{k^2}$$

$$\geq \frac{\underline{p}^2 \sum_{t=1}^k \frac{\beta}{l^2}k^2}{k^2}$$

$$= \frac{\underline{p}^2 k \beta}{l^2}$$

$$= \Omega(k) .$$

**Logistic regression.** $V$ here is the negated $\ell_2$-regularized logistic loss. Similarly, we can break the variance of the utility into the variance of the loss on each test data point

$$\text{Var}[V(\bar{\boldsymbol{\theta}}_{\pi^t,j} - \alpha z_t) | \bar{\boldsymbol{\theta}}_{\pi^t,j}] = \text{Var}[\frac{1}{l}\sum_{i=1}^l y_i \log(1/(1 + e^{-(\bar{\boldsymbol{\theta}}_{\pi^t,j} - \alpha z_t)^\top \boldsymbol{x}_i})) + (1 - y)\log(1 - 1/(1 + e^{-(\bar{\boldsymbol{\theta}}_{\pi^t,j} - \alpha z_t)^\top \boldsymbol{x}_i}))$$

$$- \|\bar{\boldsymbol{\theta}}_{\pi^t,j} - \alpha z_t\|_2^2 | \bar{\boldsymbol{\theta}}_{\pi^t,j}]$$

$$= \frac{1}{l^2}\sum_{i=1}^l \text{Var}[y_i \log(1/(1 + e^{-(\bar{\boldsymbol{\theta}}_{\pi^t,j} - \alpha z_t)^\top \boldsymbol{x}_i})) + (1 - y_i)\log(1 - 1/(1 + e^{-(\bar{\boldsymbol{\theta}}_{\pi^t,j} - \alpha z_t)^\top \boldsymbol{x}_i}))$$

$$- \|\bar{\boldsymbol{\theta}}_{\pi^t,j} - \alpha z_t\|_2^2 | \bar{\boldsymbol{\theta}}_{\pi^t,j}] .$$

Denote the logistic loss $p_i(\boldsymbol{\theta}) := y_i \log(1/(1 + e^{-\boldsymbol{\theta}^\top \boldsymbol{x}_i})) + (1 - y_i)\log(1 - 1/(1 + e^{-\boldsymbol{\theta}^\top \boldsymbol{x}_i})) \geq 0$. Further denote the $\ell_2$ regularizer $g(\boldsymbol{\theta}) = \lambda\|\boldsymbol{\theta}\|_2^2 \geq 0$. To ease notation, we let

$$\xi_t := (\bar{\boldsymbol{\theta}}_{\pi^t,j} - \alpha z_t)^\top \boldsymbol{x}_i | \bar{\boldsymbol{\theta}}_{\pi^t,j} \sim \mathcal{N}(\bar{\boldsymbol{\theta}}_{\pi^t,j}^\top \boldsymbol{x}_i, \alpha^2 k \sigma^2 \boldsymbol{x}_i^\top \boldsymbol{x}_i) .$$

Let $u := \bar{\boldsymbol{\theta}}_{\pi^t,j}^\top \boldsymbol{x}_i$ and $s^2 := \alpha^2 k \sigma^2 \boldsymbol{x}_i^\top \boldsymbol{x}_i$. Now we show that $\mathbb{E}[-p_i(\bar{\boldsymbol{\theta}}_{\pi^t,j} - \alpha z_t) | \bar{\boldsymbol{\theta}}_{\pi^t,j}] = \mathcal{O}(\sqrt{k})$. First, consider when the label $y_i = 1$,

$$
\begin{aligned}
0 \le \mathbb{E}[-p_i(\bar{\boldsymbol{\theta}}_{\pi^t,j} - \alpha z_t) | \bar{\boldsymbol{\theta}}_{\pi^t,j}] &= -\mathbb{E}[\log(1/(1 + e^{-\xi_t})] \\
&= \mathbb{E}[\log(1 + e^{-\xi_t})] \\
&= \int_{-\infty}^{\infty} p(\xi_t) \log(1 + e^{-\xi_t}) \mathrm{d}\xi_t \\
&= \int_{-\infty}^{0} p(\xi_t) \log(1 + e^{-\xi_t}) \mathrm{d}\xi_t + \int_{0}^{\infty} p(\xi_t) \log(1 + e^{-\xi_t}) \mathrm{d}\xi_t \\
&\le \int_{-\infty}^{0} p(\xi_t)(1 + \log(e^{-\xi_t})) \mathrm{d}\xi_t + \int_{0}^{\infty} p(\xi_t) \log 2 \, \mathrm{d}\xi_t \\
&\le \int_{-\infty}^{0} p(\xi_t)(1 - \xi_t) \mathrm{d}\xi_t + \int_{-\infty}^{\infty} p(\xi_t) \log 2 \, \mathrm{d}\xi_t \\
&= \int_{-\infty}^{0} p(\xi_t)(1 - \xi_t) \mathrm{d}\xi_t + \log 2 \\
&\le \log 2 + 1 + \int_{-\infty}^{0} p(\xi_t)(-\xi_t) \mathrm{d}\xi_t \\
&= \log 2 + 1 + \frac{1}{s\sqrt{2\pi}} \int_{-\infty}^{0} (-\xi_t) \exp\left(-\frac{(\xi_t - u)^2}{2s^2}\right) \mathrm{d}\xi_t \\
&= \log 2 + 1 + \left(-\frac{1}{2}\mu \operatorname{erfc}\left(\frac{\sqrt{2}\mu}{2s}\right) + \sqrt{\frac{1}{2\pi}} s \exp\left(-\frac{u^2}{2s^2}\right)\right) \\
&\le \log 2 + 1 + |u| + s\sqrt{\frac{1}{2\pi}} \exp\left(-\frac{u^2}{2s^2}\right) \\
&\le \log 2 + 1 + |u| + s \\
&= \mathcal{O}(s) \\
&= \mathcal{O}(\sqrt{k})
\end{aligned}
$$

where erfc represents the complementary error function and the integral $\frac{1}{s\sqrt{2\pi}} \int_{-\infty}^{0} (-\xi_t) \exp\left(-\frac{(\xi_t - u)^2}{2s^2}\right) \mathrm{d}\xi_t = \left(-\frac{1}{2}\mu \operatorname{erfc}\left(\frac{\sqrt{2}\mu}{2s}\right) + \sqrt{\frac{1}{2\pi}} s \exp\left(-\frac{u^2}{2s^2}\right)\right)$ can be derived with a math solver (in our case, the "sympy" package of Python, and a notebook involving the code snippet for reproducing the result is included in the supplementary materials). Similarly, when the label $y_i = 0$,

$$
\begin{aligned}
\mathbb{E}[-p_i(\bar{\boldsymbol{\theta}}_{\pi^t,j} - \alpha z_t) | \bar{\boldsymbol{\theta}}_{\pi^t,j}] &= \mathbb{E}[-\log(e^{-\xi_t}/(1 + e^{-\xi_t}))] \\
&= -\mathbb{E}[-\xi_t - \log(1 + e^{-\xi_t})] \\
&= \mathbb{E}[\xi_t] + \mathbb{E}[\log(1 + e^{-\xi_t}] \\
&\le \mathbb{E}[\xi_t] + \log 2 + 1 + \int_{-\infty}^{0} p(\xi_t)(-\xi_t) \mathrm{d}\xi_t \\
&= \log 2 + 1 + \int_{0}^{\infty} p(\xi_t) \xi_t \mathrm{d}\xi_t \\
&= \log 2 + 1 + \frac{1}{s\sqrt{2\pi}} \int_{0}^{\infty} \xi_t \exp\left(-\frac{(\xi_t - u)^2}{2s^2}\right) \mathrm{d}\xi_t \\
&\le \log 2 + 1 + \frac{1}{2s}\left(2|us| + \sqrt{\frac{2}{\pi}} s^2 \exp\left(-\frac{u^2}{2s^2}\right)\right) \\
&= \mathcal{O}(\sqrt{k}) \, .
\end{aligned}
$$

In conclusion, $\mathbb{E}[-p(\bar{\boldsymbol{\theta}}_{\pi^t,j} - \alpha z_t) | \bar{\boldsymbol{\theta}}_{\pi^t,j}] = \mathcal{O}(\sqrt{k})$. Next, we work out the expectation of $\|\bar{\boldsymbol{\theta}}_{\pi^t,j} - \alpha z_t\|_2^2 | \bar{\boldsymbol{\theta}}_{\pi^t,j}$. Note that $\bar{\boldsymbol{\theta}}_{\pi^t,j} - \alpha z_t \in \mathbb{R}^d$ where $d$ is the dimension of the model parameters. By

linearity of expectation, we have

$$\mathbb{E}[\|\bar{\boldsymbol{\theta}}_{\pi^t,j} - \alpha z_t\|_2^2 | \bar{\boldsymbol{\theta}}_{\pi^t,j}] = \sum_{r=1}^d \mathbb{E}[(\bar{\boldsymbol{\theta}}_{\pi^t,j} - \alpha z_t)_r^2 | \bar{\boldsymbol{\theta}}_{\pi^t,j}]$$

$$= \sum_{r=1}^d \left( \mathbb{E}[\bar{\boldsymbol{\theta}}_{\pi^t,j}^2 | \bar{\boldsymbol{\theta}}_{\pi^t,j}] - 2\alpha \bar{\boldsymbol{\theta}}_{\pi^t,j} \mathbb{E}[z_t] + \alpha^2 \mathbb{E}[z_t^2] \right)$$

$$= \bar{\boldsymbol{\theta}}_{\pi^t,j}^2 + \alpha^2 dk\sigma^2$$

$$= \mathcal{O}(k) .$$

With this, we can expand the following square of expectation as

$$\mathbb{E}[-p(\bar{\boldsymbol{\theta}}_{\pi^t,j} - \alpha z_t) + \|\bar{\boldsymbol{\theta}}_{\pi^t,j} - \alpha z_t\|_2^2 | \bar{\boldsymbol{\theta}}_{\pi^t,j}]^2$$

$$= \mathbb{E}[-p(\bar{\boldsymbol{\theta}}_{\pi^t,j} - \alpha z_t) | \bar{\boldsymbol{\theta}}_{\pi^t,j}]^2 - 2\mathbb{E}[-p(\bar{\boldsymbol{\theta}}_{\pi^t,j} - \alpha z_t) | \bar{\boldsymbol{\theta}}_{\pi^t,j}] \mathbb{E}[\|\bar{\boldsymbol{\theta}}_{\pi^t,j} - \alpha z_t\|_2^2 | \bar{\boldsymbol{\theta}}_{\pi^t,j}] + \mathbb{E}[\|\bar{\boldsymbol{\theta}}_{\pi^t,j} - \alpha z_t\|_2^2 | \bar{\boldsymbol{\theta}}_{\pi^t,j}]^2$$

$$= \mathcal{O}(k) + \mathcal{O}(k\sqrt{k}) + \mathbb{E}[\|\bar{\boldsymbol{\theta}}_{\pi^t,j} - \alpha z_t\|_2^2 | \bar{\boldsymbol{\theta}}_{\pi^t,j}]^2$$

$$= \mathcal{O}(k\sqrt{k}) + \mathbb{E}[\|\bar{\boldsymbol{\theta}}_{\pi^t,j} - \alpha z_t\|_2^2 | \bar{\boldsymbol{\theta}}_{\pi^t,j}]^2 .$$

Next, consider that

$$\mathbb{E}[(-p(\bar{\boldsymbol{\theta}}_{\pi^t,j} - \alpha z_t) + \|\bar{\boldsymbol{\theta}}_{\pi^t,j} - \alpha z_t\|_2^2)^2 | \bar{\boldsymbol{\theta}}_{\pi^t,j}] \geq \mathbb{E}[\|\bar{\boldsymbol{\theta}}_{\pi^t,j} - \alpha z_t\|_2^4 | \bar{\boldsymbol{\theta}}_{\pi^t,j}] .$$

So, we can derive a bound of the variance from below by

$$\mathrm{Var}[-p(\bar{\boldsymbol{\theta}}_{\pi^t,j} - \alpha z_t) + \|\bar{\boldsymbol{\theta}}_{\pi^t,j} - \alpha z_t\|_2^2 | \bar{\boldsymbol{\theta}}_{\pi^t,j}] \geq \mathbb{E}[\|\bar{\boldsymbol{\theta}}_{\pi^t,j} - \alpha z_t\|_2^4 | \bar{\boldsymbol{\theta}}_{\pi^t,j}] - (\mathcal{O}(k\sqrt{k}) + \mathbb{E}[\|\bar{\boldsymbol{\theta}}_{\pi^t,j} - \alpha z_t\|_2^2 | \bar{\boldsymbol{\theta}}_{\pi^t,j}]^2)$$

$$= \mathrm{Var}[\|\bar{\boldsymbol{\theta}}_{\pi^t,j} - \alpha z_t\|_2^2 | \bar{\boldsymbol{\theta}}_{\pi^t,j}] - \mathcal{O}(k\sqrt{k})$$

$$= \sum_{r=1}^d \mathrm{Var}[(\bar{\boldsymbol{\theta}}_{\pi^t,j} - \alpha z_t)_r^2 | \bar{\boldsymbol{\theta}}_{\pi^t,j}] - \mathcal{O}(k\sqrt{k}) .$$

Consider that for each $r \in [d]$ where $d$ is the dimension of the model parameters,

$$(\bar{\boldsymbol{\theta}}_{\pi^t,j} - \alpha z_t)_r | \bar{\boldsymbol{\theta}}_{\pi^t,j} \sim \mathcal{N}((\bar{\boldsymbol{\theta}}_{\pi^t,j})_r, \alpha^2 k\sigma^2)$$

and squaring it produces a noncentral chi-squared random variable with 1 degree of freedom

$$(\bar{\boldsymbol{\theta}}_{\pi^t,j} - \alpha z_t)_r^2 | \bar{\boldsymbol{\theta}}_{\pi^t,j} \sim \lambda \chi_1^2(\mu^2/\lambda)$$

where $\lambda = \alpha^2 k\sigma^2$ and $\mu = (\boldsymbol{\theta}_{\pi^t,j})_r$. By the closed-form expression for the variance of a noncentral chi-squared random variable,

$$\mathrm{Var}[(\bar{\boldsymbol{\theta}}_{\pi^t,j} - \alpha z_t)_r^2 | \bar{\boldsymbol{\theta}}_{\pi^t,j}] = \mathrm{Var}[\lambda \chi_1^2(\mu^2/\lambda)]$$

$$= 2\lambda^2(1 + 2\mu^2/\lambda)$$

$$= 2\alpha^4 k^2 \sigma^2 + 4(\bar{\boldsymbol{\theta}}_{\pi^t,j})_r^2 \alpha^2 k\sigma^2 .$$

Plug this result back into the previous inequality,

$$\mathrm{Var}[-p(\bar{\boldsymbol{\theta}}_{\pi^t,j} - \alpha z_t) + \|\bar{\boldsymbol{\theta}}_{\pi^t,j} - \alpha z_t\|_2^2 | \bar{\boldsymbol{\theta}}_{\pi^t,j}] \geq 2d\alpha^4 k^2 \sigma^2 + 4d(\bar{\boldsymbol{\theta}}_{\pi^t,j})_r^2 \alpha^2 k\sigma^2 - \mathcal{O}(k\sqrt{k}) .$$

Hence,

$$\mathrm{Var}\left[ \frac{\sum_{t=1}^k p_j(\pi^t) V(\boldsymbol{\theta}_{\pi^t,j})}{k} \bigg| \boldsymbol{\theta}_{\pi^1,j}^p, \dots, \boldsymbol{\theta}_{\pi^k,j}^p \right] \geq \frac{\sum_{t=1}^k \underline{p}^2 \mathrm{Var}[V(\boldsymbol{\theta}_{\pi^t,j}) | \boldsymbol{\theta}_{\pi^t,j}^p]}{k^2}$$

$$= \frac{\underline{p}^2 \sum_{t=1}^k \mathrm{Var}[V(\bar{\boldsymbol{\theta}}_{\pi^t,j} - \alpha z_t) | \bar{\boldsymbol{\theta}}_{\pi^t,j}]}{k^2}$$

$$\geq \frac{\underline{p}^2 \sum_{t=1}^k 2d\alpha^4 k^2 \sigma^2 - \mathcal{O}(k\sqrt{k})}{k^2}$$

$$= 2\underline{p}^2 d\alpha^4 \sigma^2 k - \mathcal{O}(\sqrt{k})$$

$$= \Omega(k) .$$

Lastly, we derive the conditional variance of a semivalue estimator with independent noise as

$$
\begin{aligned}
\mathrm{Var}[\psi_j | \boldsymbol{\theta}^p_{\pi^1,j}, \boldsymbol{\theta}^p_{\pi^2,j}, \ldots, \boldsymbol{\theta}^p_{\pi^k,j}] &= \mathrm{Var}\left[\frac{\sum_{t=1}^k p_j(\pi^t)[V(\boldsymbol{\theta}_{\pi^t,j}) - V(\boldsymbol{\theta}^p_{\pi^t,j})]}{k} \middle| \boldsymbol{\theta}^p_{\pi^1,j}, \boldsymbol{\theta}^p_{\pi^2,j}, \ldots, \boldsymbol{\theta}^p_{\pi^k,j}\right] \\
&\geq \mathrm{Var}\left[\frac{\sum_{t=1}^k p_j(\pi^t)V(\boldsymbol{\theta}_{\pi^t,j})}{k} \middle| \bar{\boldsymbol{\theta}}_{\pi^1,j}, \ldots, \bar{\boldsymbol{\theta}}_{\pi^k,j}\right] \\
&\geq \frac{\sum_{t=1}^k \underline{p}^2 \mathrm{Var}[V(\boldsymbol{\theta}_{\pi^t,j}) | \bar{\boldsymbol{\theta}}_{\pi^t,j}]}{k^2} \\
&= \Omega(k) \ .
\end{aligned}
$$

$\square$

## C.3   MORE GENERAL PROPOSITION FOR ESTIMATION UNCERTAINTY WITH I.I.D. NOISE

**Proposition C.4. (I.I.D. Noise More General)** Denote $\bar{\boldsymbol{\theta}}_{\pi^t,j} := \boldsymbol{\theta}^p_{\pi^t,j} - \alpha \hat{g}_{\pi^t,j}$ and $\boldsymbol{\theta}_{\pi^t,j} := \boldsymbol{\theta}^p_{\pi^t,j} - \alpha \tilde{g}_{\pi^t,j}$. Suppose $V$ is "locally" $m$-strongly concave in a domain $\mathcal{D}$ which contains the model parameters in all $k$ evaluations up to a margin $\alpha K$, i.e. the union of $\ell_\infty(\alpha K)$ balls $\cup_{t \in [k]}\{\boldsymbol{\theta} \in \mathbb{R}^d : \|\boldsymbol{\theta} - \bar{\boldsymbol{\theta}}_{\pi^t,j}\|_\infty \leq \alpha K\} \subseteq \mathcal{D}$. Let $\tilde{g}_{\pi^t,j} := \hat{g}_{\pi^t,j} + z_t$ where each $z_t$ follows a truncated Gaussian distribution bounded in the range $[-K\mathbf{1}, K\mathbf{1}]$, i.e., $z_t \overset{\text{i.i.d.}}{\sim} \mathcal{N}_{\text{trunc}}(\mathbf{0}, k(C\sigma)^2 \boldsymbol{I}, -K\mathbf{1}, K\mathbf{1})$.[4] Assume that $\forall t \in [k]$, $\bar{\boldsymbol{\theta}}_{\pi^t,j}$ is close to a local optimum in the sense that $\|\nabla_{\boldsymbol{\theta}} V(\bar{\boldsymbol{\theta}}_{\pi^t,j})\|_2 < \min(\frac{m}{2}, -V(\bar{\boldsymbol{\theta}}_{\pi^t,j}))$. Let $\psi_j := k^{-1}\sum_{t=1}^k p_j(\pi^t)[V(\boldsymbol{\theta}_{\pi^t,j}) - V(\boldsymbol{\theta}^p_{\pi^t,j})]$. Then, for $k \leq K^2/(C\sigma)^2$, $\mathrm{Var}[\psi_j | \boldsymbol{\theta}^p_{\pi^1,j}, \ldots, \boldsymbol{\theta}^p_{\pi^k,j}] = \Omega(k)$.

*Proof.* First, denote $\varphi$ and $\Phi$ the pdf and cdf of a standard normal distribution. Let a truncated normal distribution $X \sim \mathcal{N}_{\text{trunc}}(\mu, \sigma^2, a, b)$. Further denote $\alpha := \frac{a-\mu}{\sigma}, \beta := \frac{b-\mu}{\sigma}$. Then, the probability density function (pdf) of $X$ is

$$
f(x; \mu, \sigma, a, b) := \frac{1}{\sigma} \frac{\varphi(\xi)}{\Phi(\beta) - \Phi(\alpha)} \ .
$$

$X$ admits a closed-form formula for the variance as

$$
\mathrm{Var}[X] = \sigma^2 \left[1 - \frac{\beta\varphi(\beta) - \alpha\varphi(\alpha)}{Z} - \left(\frac{\varphi(\alpha) - \varphi(\beta)}{Z}\right)^2\right] \leq \mathbb{E}[X^2]
$$

where $Z := \Phi(\beta) - \Phi(\alpha)$. Let $z_t := (\xi_1, \xi_2, \ldots, \xi_d)$. Note that element $\xi_r$ follows $\xi_r \sim \mathcal{N}_{\text{trunc}}(0, k(C\sigma)^2, -K, K)$. Since $K \geq \sqrt{k}\sigma$, each element of $z_t$ is truncated at least 1 standard deviation away from the mean. This suggests that $\beta \geq 1, \alpha \leq -1$. Hence, $|\varphi(\alpha) - \varphi(\beta)| \leq \max(\varphi(\alpha), \varphi(\beta)) \leq \varphi(1) = \frac{1}{\sqrt{2\pi}}e^{-1/2}$ and $Z \geq \Phi(1) - \Phi(-1) > 0.6$. Moreover, note that, for $|x| \geq 1$,

$$
(x\varphi(x))' = \frac{1}{\sqrt{2\pi}}(xe^{-x^2/2})' = \frac{1}{\sqrt{2\pi}}(1 - 2x^2)e^{-x^2/2} < 0 \ .
$$

Therefore, $\beta\varphi(\beta) - \alpha\varphi(\alpha) \leq 2\varphi(1) = \frac{2}{\sqrt{2\pi}}e^{-1/2}$. With these, we have that

$$
\begin{aligned}
1 - \frac{\beta\varphi(\beta) - \alpha\varphi(\alpha)}{Z} - \left(\frac{\varphi(\alpha) - \varphi(\beta)}{Z}\right)^2 &\geq 1 - \frac{2}{0.6 \times \sqrt{2\pi}} - \left(\frac{e^{-1/2}}{0.6 \times \sqrt{2\pi}}\right)^2 \\
&\geq 1 - 0.81 - 0.17 \\
&\geq 0.02 \ .
\end{aligned}
$$

Consider that

$$
\mathbb{E}[\|\alpha z_t\|_2^2] = \alpha^2 \sum_{r=1}^d \mathbb{E}[\xi_r^2] \geq \alpha^2 \sum_{r=1}^d \mathrm{Var}[\xi] \geq 0.02\alpha^2 dk(C\sigma)^2 \ .
$$

---

[4] Let $X \sim \mathcal{N}(\mu, \sigma^2)$. Then, conditional on $a < X < b$, $X$ follows a truncated normal distribution $\mathcal{N}_{\text{trunc}}(\mu, \sigma^2, a, b)$.

Next, notice that $\boldsymbol{\theta}_{\pi^t,j} - \bar{\boldsymbol{\theta}}_{\pi^t,j} = \alpha(\hat{g}_{\pi^t,j} - \tilde{g}_{\pi^t,j}) = -\alpha z_t$, i.e., $\boldsymbol{\theta}_{\pi^t,j} = \bar{\boldsymbol{\theta}}_{\pi^t,j} - \alpha z_t$. Since $z_t$ is clipped by $K\mathbf{1}$, we have that $\forall t \in k, |\boldsymbol{\theta}_{\pi^t,j} - \bar{\boldsymbol{\theta}}_{\pi^t,j}| = \alpha|z_t| \leq \alpha K\mathbf{1}$. This suggests that, $\forall t \in [k]$, $\boldsymbol{\theta}_{\pi^t,j}$ lies in the union of $\ell_\infty(\alpha K)$ balls, and therefore, $\forall t \in [k]$, $\boldsymbol{\theta}_{\pi^t,j} \in \mathcal{D}$. Let $M := \max_{t \in [k]} \|\nabla_{\boldsymbol{\theta}} V(\bar{\boldsymbol{\theta}}_{\pi^t,j})\|_2 < \frac{m}{2}$. Recall that if a function $f$ is $m$-strongly convex, there is

$$f(y) \geq f(x) + \nabla f(x)^\top (y - x) + \frac{m}{2}\|y - x\|^2 \,.$$

Since $V$ is $m$-strongly concave, $-V$ is $m$-strongly convex. Therefore, we have for $-V$

$$-V(\boldsymbol{\theta}_{\pi^t,j}) = -V(\bar{\boldsymbol{\theta}}_{\pi^t,j} - \alpha z_t) \geq -V(\bar{\boldsymbol{\theta}}_{\pi^t,j}) + \nabla_{\boldsymbol{\theta}} V(\bar{\boldsymbol{\theta}}_{\pi^t,j})^\top \alpha z_t + \frac{m}{2}\|\alpha z_t\|_2^2 \geq 0 \,.$$

By strong convexity, we also have

$$-V(\bar{\boldsymbol{\theta}}_{\pi^t,j}) \geq -V(\bar{\boldsymbol{\theta}}_{\pi^t,j} - \alpha z_t) - \nabla_{\boldsymbol{\theta}} V(\bar{\boldsymbol{\theta}}_{\pi^t,j} - \alpha z_t)^\top \alpha z_t + \frac{m}{2}\|\alpha z_t\|_2^2$$

$$-V(\bar{\boldsymbol{\theta}}_{\pi^t,j} - \alpha z_t) \leq -V(\bar{\boldsymbol{\theta}}_{\pi^t,j}) + \nabla_{\boldsymbol{\theta}} V(\bar{\boldsymbol{\theta}}_{\pi^t,j} - \alpha z_t)^\top \alpha z_t - \frac{m}{2}\|\alpha z_t\|_2^2 \,.$$

Consider that

$$\mathbb{E}[(-V(\boldsymbol{\theta}_{\pi^t,j}))^2 | \bar{\boldsymbol{\theta}}_{\pi^t,j}]$$

$$= \mathbb{E}[(-V(\bar{\boldsymbol{\theta}}_{\pi^t,j} - \alpha z_t))^2 | \bar{\boldsymbol{\theta}}_{\pi^t,j}]$$

$$\geq \mathbb{E}[(-V(\bar{\boldsymbol{\theta}}_{\pi^t,j}) + \nabla_{\boldsymbol{\theta}} V(\bar{\boldsymbol{\theta}}_{\pi^t,j})^\top \alpha z_t + \frac{m}{2}\|\alpha z_t\|_2^2)^2 | \bar{\boldsymbol{\theta}}_{\pi^t,j}]$$

$$\geq V(\bar{\boldsymbol{\theta}}_{\pi^t,j})^2 + \frac{m^2}{4}\mathbb{E}[\|\alpha z_t\|_2^4] - mV(\bar{\boldsymbol{\theta}}_{\pi^t,j})\mathbb{E}[\|\alpha z_t\|_2^2]$$

$$- 2\alpha V(\bar{\boldsymbol{\theta}}_{\pi^t,j})\mathbb{E}[\nabla_{\boldsymbol{\theta}} V(\bar{\boldsymbol{\theta}}_{\pi^t,j})^\top z_t | \bar{\boldsymbol{\theta}}_{\pi^t,j}] + m\alpha\mathbb{E}[(\nabla_{\boldsymbol{\theta}} V(\bar{\boldsymbol{\theta}}_{\pi^t,j})^\top z_t)\|z_t\|_2^2 | \bar{\boldsymbol{\theta}}_{\pi^t,j}]$$

$$\geq V(\bar{\boldsymbol{\theta}}_{\pi^t,j})^2 + \frac{m^2}{4}\mathbb{E}[\|\alpha z_t\|_2^4] - 2\alpha V(\bar{\boldsymbol{\theta}}_{\pi^t,j})\mathbb{E}[\nabla_{\boldsymbol{\theta}} V(\bar{\boldsymbol{\theta}}_{\pi^t,j})^\top z_t | \bar{\boldsymbol{\theta}}_{\pi^t,j}] + m\alpha\mathbb{E}[(\nabla_{\boldsymbol{\theta}} V(\bar{\boldsymbol{\theta}}_{\pi^t,j})^\top z_t)\|z_t\|_2^2 | \bar{\boldsymbol{\theta}}_{\pi^t,j}] \,.$$

Consider that, similar to a (non-truncated) normal distribution, $\mathbb{E}[\xi_r] = \mathbb{E}[\xi_r^3] = 0$ due to symmetry of $\xi_r$, we have

$$\mathbb{E}[\nabla_{\boldsymbol{\theta}} V(\bar{\boldsymbol{\theta}}_{\pi^t,j})^\top z_t | \bar{\boldsymbol{\theta}}_{\pi^t,j}] = \sum_{r=1}^d \mathbb{E}[(\nabla_{\boldsymbol{\theta}} V(\bar{\boldsymbol{\theta}}_{\pi^t,j}))_r \xi_r | \bar{\boldsymbol{\theta}}_{\pi^t,j}] = \sum_{r=1}^d (\nabla_{\boldsymbol{\theta}} V(\bar{\boldsymbol{\theta}}_{\pi^t,j}))_r \mathbb{E}[\xi_r] = 0$$

and

$$\mathbb{E}[(\nabla_{\boldsymbol{\theta}} V(\bar{\boldsymbol{\theta}}_{\pi^t,j})^\top z_t)\|z_t\|_2^2 | \bar{\boldsymbol{\theta}}_{\pi^t,j}] = \sum_{r=1}^d \mathbb{E}[(\nabla_{\boldsymbol{\theta}} V(\bar{\boldsymbol{\theta}}_{\pi^t,j}))_r \xi_r^3 | \bar{\boldsymbol{\theta}}_{\pi^t,j}] + \sum_{r \neq r'} \mathbb{E}[(\nabla_{\boldsymbol{\theta}} V(\bar{\boldsymbol{\theta}}_{\pi^t,j}))_r \xi_r \xi_{r'}^2 | \bar{\boldsymbol{\theta}}_{\pi^t,j}]$$

$$= \sum_{r=1}^d (\nabla_{\boldsymbol{\theta}} V(\bar{\boldsymbol{\theta}}_{\pi^t,j}))_r \mathbb{E}[\xi_r^3] + \sum_{r \neq r'} \mathbb{E}[(\nabla_{\boldsymbol{\theta}} V(\bar{\boldsymbol{\theta}}_{\pi^t,j}))_r \xi_r | \bar{\boldsymbol{\theta}}_{\pi^t,j}]\mathbb{E}[\xi_{r'}^2]$$

$$= 0 + 0$$

$$= 0 \,.$$

Therefore,

$$\mathbb{E}[(-V(\boldsymbol{\theta}_{\pi^t,j}))^2 | \bar{\boldsymbol{\theta}}_{\pi^t,j}] \geq V(\bar{\boldsymbol{\theta}}_{\pi^t,j})^2 + \frac{m^2}{4}\mathbb{E}[\|\alpha z_t\|_2^4] \,.$$

Since $|z_t| \leq K\mathbf{1}$, $(\bar{\boldsymbol{\theta}}_{\pi^t,j} - \alpha z_t)$ is closed and bounded. By the extreme value theorem, there exists a maximum of $\|\nabla_{\boldsymbol{\theta}} V(\bar{\boldsymbol{\theta}}_{\pi^t,j} - \alpha z_t)\|_2^2$. We may let $\|\nabla_{\boldsymbol{\theta}} V(\bar{\boldsymbol{\theta}}_{\pi^t,j} - \alpha z_t)\|_2^2 \leq D$. Then, $\|\nabla_{\boldsymbol{\theta}} V(\bar{\boldsymbol{\theta}}_{\pi^t,j} - \alpha z_t)\|_2 \leq \sqrt{D}$. With this, we have

$$\mathbb{E}[\nabla_{\boldsymbol{\theta}} V(\bar{\boldsymbol{\theta}}_{\pi^t,j} - \alpha z_t)^\top \alpha z_t | \bar{\boldsymbol{\theta}}_{\pi^t,j}] \leq \mathbb{E}[\|\nabla_{\boldsymbol{\theta}} V(\bar{\boldsymbol{\theta}}_{\pi^t,j} - \alpha z_t)\|_2 \|\alpha z_t\|_2 | \bar{\boldsymbol{\theta}}_{\pi^t,j}]$$

$$= \|\nabla_{\boldsymbol{\theta}} V(\bar{\boldsymbol{\theta}}_{\pi^t,j} - \alpha z_t)\|_2 \mathbb{E}[\|\alpha z_t\|_2]$$

$$\leq \sqrt{D}\mathbb{E}[\|\alpha z_t\|_2]$$

and

$$\mathbb{E}[\nabla_{\boldsymbol{\theta}}V(\bar{\boldsymbol{\theta}}_{\pi^t,j}-\alpha z_t)^{\top}\alpha z_t|\bar{\boldsymbol{\theta}}_{\pi^t,j}] \geq \mathbb{E}[-\|\nabla_{\boldsymbol{\theta}}V(\bar{\boldsymbol{\theta}}_{\pi^t,j}-\alpha z_t)\|_2\|\alpha z_t\|_2|\bar{\boldsymbol{\theta}}_{\pi^t,j}]$$

$$= -\|\nabla_{\boldsymbol{\theta}}V(\bar{\boldsymbol{\theta}}_{\pi^t,j}-\alpha z_t)\|_2\mathbb{E}[\|\alpha z_t\|_2]$$

$$\geq -\sqrt{D}\mathbb{E}[\|\alpha z_t\|_2] \, ,$$

where the 1st step in both inequalities above is by Cauchy-Scharwz inequality. We can further bound $\mathbb{E}[\|\alpha z_t\|_2]$ by

$$\mathbb{E}[\|\alpha z_t\|_2] = \sqrt{\mathbb{E}[\|\alpha z_t\|_2]^2}$$

$$\leq \sqrt{\mathbb{E}[\|\alpha z_t\|_2^2]}$$

$$= \sqrt{\alpha^2 \sum_{r=1}^{d}\mathbb{E}[\xi_r^2]}$$

$$= \alpha\sqrt{\sum_{r=1}^{d}\mathrm{Var}[\xi]}$$

$$= \alpha\sqrt{\sum_{r=1}^{d}k(C\sigma)^2}$$

$$= \alpha C\sigma\sqrt{dk}$$

where in the 2nd and 4th steps we use the formula $\mathrm{Var}[X]=\mathbb{E}[X^2]-\mathbb{E}[X]^2$. With this, we have

$$\mathbb{E}[-V(\bar{\boldsymbol{\theta}}_{\pi^t,j}-\alpha z_t)|\bar{\boldsymbol{\theta}}_{\pi^t,j}]^2 \leq \mathbb{E}[-V(\bar{\boldsymbol{\theta}}_{\pi^t,j})+\nabla_{\boldsymbol{\theta}}V(\bar{\boldsymbol{\theta}}_{\pi^t,j}-\alpha z_t)^{\top}\alpha z_t - \frac{m}{2}\|\alpha z_t\|_2^2|\bar{\boldsymbol{\theta}}_{\pi^t,j}]^2$$

$$\leq V(\bar{\boldsymbol{\theta}}_{\pi^t,j})^2 + D\mathbb{E}[\|\alpha z_t\|_2]^2 + \frac{m^2}{4}\mathbb{E}[\|\alpha z_t\|_2^2]^2$$

$$- 2\alpha V(\bar{\boldsymbol{\theta}}_{\pi^t,j})\mathbb{E}[\nabla_{\boldsymbol{\theta}}V(\bar{\boldsymbol{\theta}}_{\pi^t,j}-\alpha z_t)^{\top}\alpha z_t|\bar{\boldsymbol{\theta}}_{\pi^t,j}] + mV(\bar{\boldsymbol{\theta}}_{\pi^t,j})\mathbb{E}[\|\alpha z_t\|_2^2]$$

$$- m\mathbb{E}[\nabla_{\boldsymbol{\theta}}V(\bar{\boldsymbol{\theta}}_{\pi^t,j}-\alpha z_t)^{\top}\alpha z_t|\bar{\boldsymbol{\theta}}_{\pi^t,j}]\mathbb{E}[\|\alpha z_t\|_2^2]$$

$$\leq V(\bar{\boldsymbol{\theta}}_{\pi^t,j})^2 + D\mathbb{E}[\|\alpha z_t\|_2^2] + \frac{m^2}{4}\mathbb{E}[\|\alpha z_t\|_2^2]^2 - 2\alpha V(\bar{\boldsymbol{\theta}}_{\pi^t,j})\sqrt{D}\mathbb{E}[\|\alpha z_t\|_2]$$

$$+ m\sqrt{D}\mathbb{E}[\|\alpha z_t\|_2]\mathbb{E}[\|\alpha z_t\|_2^2]$$

$$\leq V(\bar{\boldsymbol{\theta}}_{\pi^t,j})^2 + \frac{m^2}{4}\mathbb{E}[\|\alpha z_t\|_2^2]^2 + (D+m\sqrt{D}\alpha\sigma\sqrt{dk})\mathbb{E}[\|\alpha z_t\|_2^2] - 2\alpha^2 V(\bar{\boldsymbol{\theta}}_{\pi^t,j})\sqrt{D}\sigma\sqrt{dk}$$

$$= V(\bar{\boldsymbol{\theta}}_{\pi^t,j})^2 + \frac{m^2}{4}\mathbb{E}[\|\alpha z_t\|_2^2]^2 + (D+m\sqrt{D}\alpha\sigma\sqrt{dk})\alpha^2 dk\sigma^2 - 2\alpha^2 V(\bar{\boldsymbol{\theta}}_{\pi^t,j})\sqrt{D}\sigma\sqrt{dk}$$

$$= \frac{m^2}{4}\mathbb{E}[\|\alpha z_t\|_2^2]^2 + \mathcal{O}(k\sqrt{k})$$

where in the last step we make use of the fact $\mathbb{E}[\|\alpha z_t\|_2^2] = \alpha^2\sum_{r=1}^{d}\mathbb{E}[\xi_r^2] = \alpha^2 dk(C\sigma)^2$. Therefore,

$$\mathrm{Var}[V(\boldsymbol{\theta}_{\pi^t,j})|\bar{\boldsymbol{\theta}}_{\pi^t,j}] = \mathbb{E}[V(\boldsymbol{\theta}_{\pi^t,j})^2|\bar{\boldsymbol{\theta}}_{\pi^t,j}] - \mathbb{E}[V(\boldsymbol{\theta}_{\pi^t,j})|\bar{\boldsymbol{\theta}}_{\pi^t,j}]^2$$

$$\geq \frac{m^2}{4}\mathbb{E}[\|\alpha z_t\|_2^4] - \frac{m^2}{4}\mathbb{E}[\|\alpha z_t\|_2^2]^2 + \Omega(k\sqrt{k})$$

$$= \mathrm{Var}[\|\alpha z_t\|_2^2] + \Omega(k\sqrt{k}) \, .$$

Note that

$$\mathrm{Var}[\|\alpha z_t\|_2^2] = \mathrm{Var}[\sum_{r=1}^{d}\xi_r^2]$$

$$= \sum_{r=1}^{d}\mathrm{Var}[\xi_r^2] \, .$$

Intuitively, $\mathrm{Var}[\xi_r^2]$ should be asymptotically the same as the variance of a chi-squared random variable, i.e., $\Omega(k^2)$. However, because of truncation, the expression of $\mathrm{Var}[\xi_r^2]$ is much more complicated

than that of a chi-squared random variable. We use a program built with Python Sympy to analyze the exact form of $\mathrm{Var}[\xi_r^2]$. The relevant code (written in jupyter notebook) has been included in the supplementary materials. Particularly, its variance has the following expression:

$$-\frac{k^2\sigma^4\left(-\frac{\sqrt{2}Ke^{-\frac{K^2}{2k\sigma^2}}}{\sqrt{\pi}\sqrt{k}\sigma}+\mathrm{erf}\left(\frac{\sqrt{2}K}{2\sqrt{k}\sigma}\right)\right)^2}{\mathrm{erf}^2\left(\frac{\sqrt{2}K}{2\sqrt{k}\sigma}\right)}+\frac{k^2\sigma^4\left(-\frac{6\sqrt{2}Ke^{-\frac{K^2}{2k\sigma^2}}}{\sqrt{\pi}\sqrt{k}\sigma}-\frac{\sqrt{2}\left(\frac{K^3e^{-\frac{K^2}{2k\sigma^2}}}{k^{3/2}\sigma^3}-\frac{3Ke^{-\frac{K^2}{2k\sigma^2}}}{\sqrt{k}\sigma}\right)}{\sqrt{\pi}}+3\,\mathrm{erf}\left(\frac{\sqrt{2}K}{2\sqrt{k}\sigma}\right)\right)}{\mathrm{erf}\left(\frac{\sqrt{2}K}{2\sqrt{k}\sigma}\right)}\,.$$

Note that we have $K^2 \geq k\sigma^2$. As such, we may denote $v := K^2/(k\sigma^2) \geq 1$. Then, the above expression can be reduced to

$$k^2\sigma^4\left(-\frac{\left(-\frac{\sqrt{2}v^{0.5}e^{-\frac{v}{2}}}{\sqrt{\pi}}+\mathrm{erf}\left(\frac{\sqrt{2}v^{0.5}}{2}\right)\right)^2}{\mathrm{erf}^2\left(\frac{\sqrt{2}v^{0.5}}{2}\right)}+\frac{-\frac{6\sqrt{2}v^{0.5}e^{-\frac{v}{2}}}{\sqrt{\pi}}-\frac{\sqrt{2}\left(-3v^{0.5}e^{-\frac{v}{2}}+v^{1.5}e^{-\frac{v}{2}}\right)}{\sqrt{\pi}}+3\,\mathrm{erf}\left(\frac{\sqrt{2}v^{0.5}}{2}\right)}{\mathrm{erf}\left(\frac{\sqrt{2}v^{0.5}}{2}\right)}\right)\,.$$

Let the part in the large bracket be $f(v)$. A plot of $f$ is in Fig. 4. Note that as $v \to \infty$, $\mathrm{erf}\left(\frac{\sqrt{2}v^{0.5}}{2}\right)$ becomes the dominant term, thus $f \to 2$ as $v \to 1$, $f$ decreases, but stays positive. With these, we conclude that $\mathrm{Var}[\xi_r^2] \propto k^2$. As such, $\mathrm{Var}[V(\boldsymbol{\theta}_{\pi^t,j})] = \Omega(k^2)$.

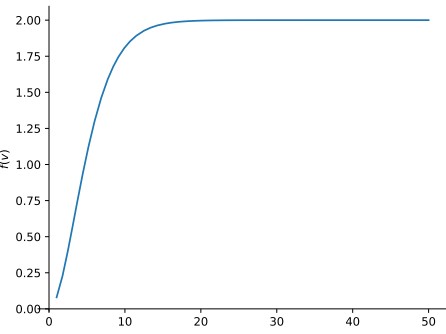

Figure 4: Plot of $f$ vs. $v \in [1, 50]$.

Note that, when $\boldsymbol{\theta}^p_{\pi^t,j}$ is determined, so is $\bar{\boldsymbol{\theta}}_{\pi^t,j}$. Denote $\underline{p} := \min_{\pi \in \Pi} p_j(\pi)$ and apply Lemma C.2. Since each $z_t$ is i.i.d., we have

$$\mathrm{Var}[\psi_j|\boldsymbol{\theta}^p_{\pi^1,j},\ldots,\boldsymbol{\theta}^p_{\pi^k,j}] = \mathrm{Var}\left[\frac{\sum_{t=1}^k p_j(\pi^t)[V(\boldsymbol{\theta}_{\pi^t,j})-V(\boldsymbol{\theta}^p_{\pi^t,j})]}{k}\Bigg|\boldsymbol{\theta}^p_{\pi^1,j},\ldots,\boldsymbol{\theta}^p_{\pi^k,j}\right]$$

$$= \mathrm{Var}\left[\frac{\sum_{t=1}^k p_j(\pi^t)V(\boldsymbol{\theta}_{\pi^t,j})}{k}\Bigg|\bar{\boldsymbol{\theta}}_{\pi^1,j},\ldots,\bar{\boldsymbol{\theta}}_{\pi^k,j}\right]$$

$$\geq \frac{\underline{p}^2}{k^2}\sum_{t=1}^k \mathrm{Var}[V(\boldsymbol{\theta}_{\pi^t,j})|\bar{\boldsymbol{\theta}}_{\pi^t,j}]$$

$$= \frac{\sum_{t=1}^k \Omega(k^2)}{k^2}$$

$$= \Omega(k)\,.$$

$\square$

## C.4 Useful Lemmas and Corollaries

**Lemma C.5.** Let $z_t^* \coloneqq \sum_{l=1}^{t} \boldsymbol{X}_{t,l} z_l$ where $z_l \overset{\text{i.i.d.}}{\sim} \mathcal{N}(\boldsymbol{0}, k(C\sigma)^2 \boldsymbol{I}) \in \mathbb{R}^d$ are i.i.d. drawn. Among all matrices satisfying P1 and P2, $\boldsymbol{X}^*$ as defined in Eq. (4) produces the lowest values of $N \coloneqq \sum_{t=1}^{k} \mathbb{E}[\|z_t^*\|_2^2]$, $P \coloneqq \sum_{t=1}^{k} \mathbb{E}[\|z_t^*\|_2^4]$, and $Q \coloneqq \sum_{t=1}^{k} \sqrt{\mathbb{E}[\|z_t^*\|_2^4]}$. In particular, we have

$$N = \mathcal{O}(k \log k),$$
$$P = \mathcal{O}(k^2),$$
$$Q = \mathcal{O}(k \log k).$$

*Proof.* Since $z_t^* \in \mathbb{R}^d$ and is diagonal, we may write $z_t^* = (\xi_{t,1}, \xi_{t,2}, \dots, \xi_{t,d})$, where $\forall t \in [k], r \in [d], \xi_{t,r} \sim \mathcal{N}(0, k(C\sigma)^2 \sum_{l=1}^{t} \boldsymbol{X}_{t,l}^2)$ since all $z_l$'s are independent. Then, for each $z_t^*$, there is

$$\mathbb{E}[\|z_t^*\|_2^2] = \mathbb{E}[\sum_{r=1}^{d} \xi_{t,r}^2]$$
$$= \mathbb{E}[\sum_{r=1}^{d} (k(C\sigma)^2 \sum_{l=1}^{t} \boldsymbol{X}_{t,l}^2) \chi_1^2]$$
$$= \mathbb{E}[(k(C\sigma)^2 \sum_{l=1}^{t} \boldsymbol{X}_{t,l}^2) \chi_d^2]$$
$$= dk(C\sigma)^2 \sum_{l=1}^{t} \boldsymbol{X}_{t,l}^2$$

where $\chi_d^2$ refers to a chi-squared random variable with a degree of freedom $d$ which satisfies $\mathbb{E}[\chi_d^2] = d$. Since $N$ is the sum of $\mathbb{E}[\|z_t^*\|_2^2]$ over all $t \in [k]$, we can minimize $N$ if $\exists \boldsymbol{X}^*$, s.t. $\forall t \in [k], \boldsymbol{X}^* = \operatorname{argmin}_{\boldsymbol{X}} \mathbb{E}[\|z_t^*\|_2^2]$. Fortunately, the optimization problem is a convex quadratic program. Specifically, let $\boldsymbol{w}_t = (\boldsymbol{X}_{t,1}, \boldsymbol{X}_{t,2}, ..., \boldsymbol{X}_{t,t})^\top$, we have the following optimisation problem for each $t \in [k]$:

$$\begin{aligned} \min_{\boldsymbol{w}_t} \quad & \boldsymbol{w}_t^\top \boldsymbol{w}_t \\ \text{s.t.} \quad & \mathbf{1}^\top \boldsymbol{w}_t = 1 \\ & \boldsymbol{w}_t \geq \boldsymbol{0} . \end{aligned} \tag{5}$$

To solve this optimization problem, we adopt the standard Lagrange multiplier approach. The Lagrangian is

$$L(\boldsymbol{w}_t, \lambda) = \boldsymbol{w}_t^\top \boldsymbol{w}_t - \lambda(1 - \mathbf{1}^\top \boldsymbol{w}_t) .$$

Then, we wish to have

$$\nabla_{w_i} L(\boldsymbol{w}_t, \lambda) = 0$$
$$\forall l \in [t], 2\boldsymbol{X}_{t,l} + \lambda = 0 .$$

This suggests that the optimal is obtained when $\boldsymbol{X}_{t,1} = \boldsymbol{X}_{t,2} = ... = \boldsymbol{X}_{t,t} = -\lambda/2 = 1/t$ for each $t \in [k]$, which matches the form of $\boldsymbol{X}^*$ in Eq. 4. With $\boldsymbol{X}^*$, $\xi_{t,r} \sim \mathcal{N}(0, k(C\sigma)^2/t)$, thus we can

derive the lowest value of $N$ as

$$\min N = \sum_{t=1}^{k} \mathbb{E}[\|z_t^*\|_2^2]$$

$$= \sum_{t=1}^{k} \mathbb{E}[\sum_{r=1}^{d} \xi_{t,r}^2]$$

$$= \sum_{t=1}^{k} \frac{k(C\sigma)^2}{t} \mathbb{E}[\chi_d^2]$$

$$= \sum_{t=1}^{k} \frac{dk(C\sigma)^2}{t}$$

$$\leq dk(C\sigma)^2 (1 + \log k)$$

$$= \mathcal{O}(k \log k)$$

where the second last step is by the inequality $\sum_{t=1}^{k} 1/t < 1 + \log k$. Next, we show that $\boldsymbol{X}^*$ is also a minimizer of $P$. Consider each summand of $P$ can be expressed as

$$\mathbb{E}[\|z_t^*\|_2^4] = \mathbb{E}[(\sum_{r=1}^{d} \xi_{t,r}^2)^2]$$

$$= \mathbb{E}[\sum_{r=1}^{d} (\xi_{t,r}^2)^2] + \mathbb{E}[\sum_{r \neq r'} \xi_{t,r}^2 \xi_{t,r'}^2]$$

$$= \sum_{r=1}^{d} k^2 (C\sigma)^4 (\sum_{l=1}^{t} \boldsymbol{X}_{t,l}^2)^2 \mathbb{E}[(\chi_1^2)^2] + d(d-1)\mathbb{E}[\xi_{t,1}^2]^2$$

$$= \sum_{r=1}^{d} k^2 (C\sigma)^4 (\sum_{l=1}^{t} \boldsymbol{X}_{t,l}^2)^2 \mathbb{E}[(\chi_1^2)^2] + d(d-1)(k(C\sigma)^2 (\sum_{l=1}^{t} \boldsymbol{X}_{t,l}^2))^2 \mathbb{E}[\chi_1^2]^2$$

$$= (3d + d(d-1)) k^2 (C\sigma)^4 (\sum_{l=1}^{t} \boldsymbol{X}_{t,l}^2)^2$$

$$= d(d+2) k^2 (C\sigma)^4 (\sum_{l=1}^{t} \boldsymbol{X}_{t,l}^2)^2 \ .$$

where in the 3rd step we use the fact $\mathbb{E}[\xi_{t,r}^2 \xi_{t,r'}^2] = \mathbb{E}[\xi_{t,r}^2]\mathbb{E}[\xi_{t,r'}^2] = \mathbb{E}[\xi_{t,1}^2]^2$ since $\xi_{t,r}$'s are i.i.d. for all $r \in [d]$. In the 4th step we use the fact that $\mathbb{E}[(\xi_1^2)^2] = 3$. Notice that since $\sum_{l=1}^{t} \boldsymbol{X}_{t,l}^2 \geq 0$, minimizing $\mathbb{E}[\|z_t^*\|_2^4]$ is equivalent to minimizing $\sum_{l=1}^{t} \boldsymbol{X}_{t,l}^2$. The minimizer is discussed in the above convex quadratic program where the optimal solution is $\boldsymbol{X}^*$. As such, we can derive the minimum of $P$ as

$$\min P = \sum_{t=1}^{k} \mathbb{E}[\|z_t^*\|_2^4]$$

$$= \sum_{t=1}^{k} d(d+2) k^2 (C\sigma)^4 / t^2$$

$$\leq \sum_{t=1}^{\infty} d(d+2) k^2 (C\sigma)^4 / t^2$$

$$= d(d+2) k^2 (C\sigma)^4 \pi^2 / 6$$

$$= \mathcal{O}(k^2)$$

where the second last step is by the fact $\sum_{t=1}^{\infty} 1/t^2 = \pi^2/6$. Lastly, we show, in the same vein, that the minimizer of $Q$ is also $\boldsymbol{X}^*$. This is obvious since the minimizer of $\mathbb{E}[\|z_t^*\|_2^4]$ is obviously the

minimizer of $\sqrt{\mathbb{E}[\|z_t^*\|_2^4]}$. The minimum of $Q$ can hence be derived as

$$
\begin{aligned}
\min Q &= \sum_{t=1}^{k} \sqrt{\mathbb{E}[\|z_t^*\|_2^4]} \\
&= \sum_{t=1}^{k} \sqrt{d(d+2)} k (C\sigma)^2 / t \\
&\leq \sqrt{d(d+2)} k (C\sigma)^2 (1 + \log k) \\
&= \mathcal{O}(k \log k) \, .
\end{aligned}
$$

□

**Lemma C.6.** Let $z_t^* := -\zeta_t + \sum_{l=1}^{t} \boldsymbol{X}_{t,l}(z_l + \zeta_l)$ where $z_l \overset{\text{i.i.d.}}{\sim} \mathcal{N}(\boldsymbol{0}, k(C\sigma)^2 \boldsymbol{I}) \in \mathbb{R}^d$ and $\zeta_l \overset{\text{i.i.d.}}{\sim} \mathcal{N}(\boldsymbol{0}, \Sigma) \in \mathbb{R}^d$ are i.i.d. drawn and $\Sigma \in \mathbb{R}^{(d \times d)}$ is a diagonal matrix. Denote $\sigma_g^2 := 1/d \sum_{r=1}^{d} \Sigma_{r,r}$. The matrix defined as $\forall t \in [k], \forall l \in [t-1], \boldsymbol{X}_{t,l} = t^{-1} (k(C\sigma)^2 + \sigma_g^2)^{-1} k(C\sigma)^2$ and $\forall t \in [k], \boldsymbol{X}_{t,t} = t^{-1} (k(C\sigma)^2 + \sigma_g^2)^{-1} (k(C\sigma)^2 + t\sigma_g^2)$ satisfies P1 and P2. Denote $N := \sum_{t=1}^{k} \mathbb{E}[\|z_t^*\|_2^2]$, $P := \sum_{t=1}^{k} \mathbb{E}[\|z_t^*\|_2^4]$, and $Q := \sum_{t=1}^{k} \sqrt{\mathbb{E}[\|z_t^*\|_2^4]}$. With this defined $\boldsymbol{X}$, we have

$$
\begin{aligned}
N &= \mathcal{O}(k \log k + k \sigma_g^2) \, , \\
P &= \mathcal{O}(k^2 + k \sigma_g^4 + \sigma_g^2 k \log k) \, , \\
Q &= \mathcal{O}(k \log k + k \sigma_g^2) \, .
\end{aligned}
$$

*Proof.* We may write $z_t^* = (\xi_{t,1}, \xi_{t,2}, \dots, \xi_{t,d})$, where $\forall t \in [k], r \in [d], \xi_{t,r} \sim \mathcal{N}(0, k(C\sigma)^2 \sum_{l=1}^{t} \boldsymbol{X}_{t,l}^2 + \Sigma_{r,r} \sum_{l=1}^{t-1} \boldsymbol{X}_{t,l}^2 + (1 - \boldsymbol{X}_{t,t})^2 \Sigma_{r,r})$ since all the $z_l$'s and $\zeta_l$'s are independent. We may denote $\sigma_{t,r}^2 := k(C\sigma)^2 \sum_{l=1}^{t} \boldsymbol{X}_{t,l}^2 + \Sigma_{r,r} \sum_{l=1}^{t-1} \boldsymbol{X}_{t,l}^2 + (1 - \boldsymbol{X}_{t,t})^2 \Sigma_{r,r}$. Then, for each $z_t^*$, there is

$$
\begin{aligned}
\mathbb{E}[\|z_t^*\|_2^2] &= \mathbb{E}[\sum_{r=1}^{d} \xi_{t,r}^2] \\
&= \sum_{r=1}^{d} \mathbb{E}[\sigma_{t,r}^2 \chi_1^2] \\
&= \sum_{r=1}^{d} \sigma_{t,r}^2
\end{aligned}
$$

where in the second step, we use the property that if $\xi \sim \mathcal{N}(0, (C\sigma)^2)$, then $\xi^2/(C\sigma)^2 = \chi_1^2$ where $\chi_1^2$ refers to a chi-squared random variable with degree of freedom 1. Similar to Lemma C.5, let $\boldsymbol{w}_t = (\boldsymbol{X}_{t,1}, \boldsymbol{X}_{t,2}, \dots, \boldsymbol{X}_{t,t})^\top$, we have the following optimization problem for each $t \in [k]$:

$$
\begin{aligned}
\min \quad & \sum_{r=1}^{d} \sigma_{t,r}^2 \\
\text{s.t.} \quad & \boldsymbol{1}^\top \boldsymbol{w}_t = 1 \\
& \boldsymbol{w}_t \geq \boldsymbol{0} \, .
\end{aligned}
\tag{6}
$$

To solve this optimization problem, we adopt the standard Lagrange multiplier approach. The Lagrangian is

$$
L(\boldsymbol{w}_t, \lambda) = \sum_{r=1}^{d} \sigma_{t,r}^2 - \lambda(1 - \boldsymbol{1}^\top \boldsymbol{w}_t) \, .
$$

Then, we wish to have

$$
\forall l \in [t], \nabla_{\boldsymbol{X}_{t,l}} L(\boldsymbol{w}_t, \lambda) = 0 \, ,
$$

which leads to

$$
\forall l \in [t-1], 2(dk(C\sigma)^2 + \sum_{r=1}^{d} \Sigma_{r,r}) \boldsymbol{X}_{t,l} + \lambda = 0
$$

$$
2(dk(C\sigma)^2 + \sum_{r=1}^{d} \Sigma_{r,r}) \boldsymbol{X}_{t,t} - 2 \sum_{r=1}^{d} \Sigma_{r,r} + \lambda = 0 \, .
$$

Solving this, we get the optimal solution as

$$\forall l \in [t-1], \boldsymbol{X}_{t,l} = \frac{dk(C\sigma)^2}{t(dk(C\sigma)^2 + \sum_{r=1}^{d} \Sigma_{r,r})} = \frac{k(C\sigma)^2}{t(k(C\sigma)^2 + \sigma_g^2)}$$

and $\boldsymbol{X}_{t,t} = 1 - \sum_{l=1}^{t-1} \boldsymbol{X}_{t,l} = \frac{dk(C\sigma)^2 + t\sum_{r=1}^{d}\Sigma_{r,r}}{t(dk(C\sigma)^2 + \sum_{r=1}^{d}\Sigma_{r,r})} = \frac{k(C\sigma)^2 + t\sigma_g^2}{t(k(C\sigma)^2 + \sigma_g^2)}.$

As compared to Lemma C.5, it is difficult to find a minimizer for $P$ and $Q$ with the Lagrange Multiplier approach as it involves a complicated quadratic equation about each $\boldsymbol{X}_{t,l}$. Nevertheless, we show that with the current minimizer we have, we can derive a good enough upper bound for $N, P, Q$. First substitute the values of $\boldsymbol{X}_{t,l}$ and we can express $\sigma_{t,r}^2$ in terms of $(C\sigma)^2$ and $\sigma_g^2$ as

$$\sigma_{t,r}^2 = \frac{k(C\sigma)^2}{t}\left(\frac{k^2(C\sigma)^4}{(k(C\sigma)^2 + \sigma_g^2)^2} + \frac{t\sigma_g^4}{(k(C\sigma)^2 + \sigma_g^2)^2} + \frac{2(C\sigma)^2\sigma_g^2}{(k(C\sigma)^2 + \sigma_g^2)^2}\right) + \frac{(t-1)d\sigma_g^2}{t} \cdot \frac{k^2(C\sigma)^4}{(k(C\sigma)^2 + \sigma_g^2)^2}$$

$$\leq \frac{3k(C\sigma)^2}{t} + \frac{k(C\sigma)^2\sigma_g^4}{(k(C\sigma)^2 + \sigma_g^2)^2} + d\sigma_g^2 \; .$$

Now consider the values of $N, P, Q$ with $\boldsymbol{X}$:

$$N = \sum_{t=1}^{k} \mathbb{E}[\|z_t^*\|_2^2]$$

$$= \sum_{t=1}^{k} \mathbb{E}[\sum_{r=1}^{d} \xi_{t,r}^2]$$

$$= \sum_{t=1}^{k} \sum_{r=1}^{d} \sigma_{t,r}^2 \mathbb{E}[\chi_1^2]$$

$$\leq \sum_{t=1}^{k} \left(\frac{3dk(C\sigma)^2}{t} + \frac{kd(C\sigma)^2\sigma_g^4}{(k(C\sigma)^2 + \sigma_g^2)^2} + d^2\sigma_g^2\right)$$

$$\leq 3dk(C\sigma)^2(1 + \log k) + \frac{k^2d(C\sigma)^2\sigma_g^4}{(k(C\sigma)^2 + \sigma_g^2)^2} + kd^2\sigma_g^2$$

$$= \mathcal{O}(k\log k + k\sigma_g^2)$$

where in the second last step we use the fact $\sum_{t=1}^{k} 1/t \leq 1 + \log k$.

$$P = \sum_{t=1}^{k} \mathbb{E}[\|z_t^*\|_2^4]$$

$$= \sum_{t=1}^{k} d(d+2)\sigma_{t,r}^4$$

$$\leq d(d+2)\sum_{t=1}^{k}\left(\frac{3dk(C\sigma)^2}{t} + \frac{kd(C\sigma)^2\sigma_g^4}{(k(C\sigma)^2 + \sigma_g^2)^2} + d^2\sigma_g^2\right)^2$$

$$= d(d+2)\sum_{t=1}^{k}\left(\frac{9d^2k^2(C\sigma)^4}{t^2} + \left(\frac{kd(C\sigma)^2\sigma_g^4}{(k(C\sigma)^2 + \sigma_g^2)^2}\right)^2 + d^4\sigma_g^4 + \frac{6dk(C\sigma)^2}{t}\frac{kd(C\sigma)^2\sigma_g^4}{(k(C\sigma)^2 + \sigma_g^2)^2}\right.$$

$$\left. + \frac{6d^3k(C\sigma)^2\sigma_g^2}{t} + \frac{2kd(C\sigma)^2\sigma_g^4}{(k(C\sigma)^2 + \sigma_g^2)^2}d^2\sigma_g^2\right)$$

$$\leq d(d+2)\left(\frac{3d^2k^2(C\sigma)^4\pi^2}{2} + \frac{k^3d^2(C\sigma)^4\sigma_g^8}{(k(C\sigma)^2+\sigma_g^2)^4} + kd^4\sigma_g^4 + 6d^2(C\sigma)^2(1 + \log k)\frac{k^2d(C\sigma)^2\sigma_g^4}{(k(C\sigma)^2+\sigma_g^2)^2}\right.$$

$$\left. + 6d^3k(C\sigma)^2\sigma_g^2(1 + \log k) + \frac{2k^2d^3(C\sigma)^2\sigma_g^6}{(k(C\sigma)^2 + \sigma_g^2)^2}\right)$$

$$= \mathcal{O}(k^2 + k\sigma_g^4 + \sigma_g^2 k\log k)$$

where in the second last step we use the fact $\sum_{t=1}^{k} 1/t^2 \leq \sum_{t=1}^{\infty} 1/t^2 = \pi^2/6$.

$$
\begin{aligned}
Q &= \sum_{t=1}^{k} \sqrt{\mathbb{E}[\|z_t^*\|_2^4]} \\
&= \sqrt{d(d+2)} \sum_{t=1}^{k} \sigma_{t,r}^2 \\
&= \mathcal{O}(N) \\
&= \mathcal{O}(k \log k + k\sigma_g^2) \,.
\end{aligned}
$$

$\square$

**Corollary C.7.** Let $z_t^* := -\zeta_t + \sum_{l=1}^{t} \boldsymbol{X}_{t,l}(z_l + \zeta_l)$ where $z_l \overset{\text{i.i.d.}}{\sim} \mathcal{N}(\boldsymbol{0}, k(C\sigma)^2\boldsymbol{I}) \in \mathbb{R}^d$ and $\zeta_l \overset{\text{i.i.d.}}{\sim} \mathcal{N}(\boldsymbol{0}, \Sigma) \in \mathbb{R}^d$ are i.i.d. drawn *and* $\Sigma \in \mathbb{R}^{(d \times d)}$ *is a diagonal matrix.* Denote $\sigma_g^2 := 1/d \sum_{r=1}^{d} \Sigma_{r,r}$. *The matrix defined as* $\forall t \in [k], \forall l \in [t-1], \boldsymbol{X}_{t,l} = t^{-1}(k(C\sigma)^2 + \sigma_g^2)^{-1}k(C\sigma)^2$ and $\forall t \in [k], \boldsymbol{X}_{t,t} = t^{-1}(k(C\sigma)^2 + \sigma_g^2)^{-1}(k(C\sigma)^2 + t\sigma_g^2)$ *satisfies* P1 *and* P2. Denote $N := \sum_{t=kq+1}^{k} \mathbb{E}[\|z_t^*\|_2^2]$, $P := \sum_{t=kq+1}^{k} \mathbb{E}[\|z_t^*\|_2^4]$, and $Q := \sum_{t=kq+1}^{k} \sqrt{\mathbb{E}[\|z_t^*\|_2^4]}$. With $\boldsymbol{X}$, we have $N, P, Q$ are upper bounded by

$$
N = \mathcal{O}(k \log \frac{1}{q} + k(1-q)\sigma_g^2) \,,
$$

$$
P = \mathcal{O}(k^2 + (1-q)k\sigma_g^4 + \sigma_g^2 k \log \frac{1}{q}) \,,
$$

$$
Q = \mathcal{O}(k \log \frac{1}{q} + k(1-q)\sigma_g^2) \,.
$$

*Proof.* First, note that the optimal matrix $\boldsymbol{X}$ is not related to where the summation starts. Therefore, the minimizer for $N$ found in Lemma C.6 is also a minimizer for $N$ in this lemma. Next, we find the upper bounds for $N, P, Q$ using $\boldsymbol{X}$:

$$
\begin{aligned}
N &= \sum_{t=kq+1}^{k} \mathbb{E}[\|z_t^*\|_2^2] \\
&= \sum_{t=kq+1}^{k} \mathbb{E}[\sum_{r=1}^{d} \xi_{t,r}^2] \\
&= \sum_{t=kq+1}^{k} \sum_{r=1}^{d} \sigma_{t,r}^2 \mathbb{E}[\chi_1^2] \\
&\leq \sum_{t=kq+1}^{k} \frac{3dk(C\sigma)^2}{t} + \frac{kd(C\sigma)^2\sigma_g^4}{(k(C\sigma)^2 + \sigma_g^2)^2} + d^2\sigma_g^2 \\
&\leq 3dk(C\sigma)^2(1 + \log k - \log kq) + k(1-q)\frac{kd(C\sigma)^2\sigma_g^4}{(k(C\sigma)^2 + \sigma_g^2)^2} + k(1-q)d^2\sigma_g^2 \\
&= \mathcal{O}(k \log \frac{k}{kq} + k(1-q)\sigma_g^2) \\
&= \mathcal{O}(k \log \frac{1}{q} + k(1-q)\sigma_g^2) \,,
\end{aligned}
$$

where in the 3rd last step we make use of the fact $\sum_{t=kq+1}^{k} 1/t = \sum_{t=1}^{k} 1/t - \sum_{t=1}^{kq} \leq 1 + \log K - \log kq$.

$$
P = \sum_{t=kq+1}^{k} \mathbb{E}[\|z_t^*\|_2^4]
$$

$$
= \sum_{t=kq+1}^{k} d(d+2)\sigma_{t,r}^4
$$

$$
\leq d(d+2) \sum_{t=kq+1}^{k} \left( \frac{3dk(C\sigma)^2}{t} + \frac{kd(C\sigma)^2\sigma_g^4}{(k(C\sigma)^2 + \sigma_g^2)^2} + d^2\sigma_g^2 \right)^2
$$

$$
= d(d+2) \sum_{t=kq+1}^{k} \left( \frac{9d^2k^2(C\sigma)^4}{t^2} + \left( \frac{kd(C\sigma)^2\sigma_g^4}{(k(C\sigma)^2 + \sigma_g^2)^2} \right)^2 + d^4\sigma_g^4 \right.
$$

$$
+ \frac{6dk(C\sigma)^2}{t} \frac{kd(C\sigma)^2\sigma_g^4}{(k(C\sigma)^2 + \sigma_g^2)^2} + \frac{6d^3k(C\sigma)^2\sigma_g^2}{t} + \left. \frac{2kd(C\sigma)^2\sigma_g^4}{(k(C\sigma)^2 + \sigma_g^2)^2}d^2\sigma_g^2 \right)
$$

$$
\leq d(d+2) \left( \frac{3d^2k^2(C\sigma)^4\pi^2}{2} + \frac{k^3d^2(C\sigma)^4\sigma_g^8}{(k(C\sigma)^2 + \sigma_g^2)^4} + (1-q)kd^4\sigma_g^4 \right.
$$

$$
\left. + 6d^2(C\sigma)^2(1 + \log k - \log kq)\frac{k^2d(C\sigma)^2\sigma_g^4}{(k(C\sigma)^2 + \sigma_g^2)^2} + 6d^3k(C\sigma)^2\sigma_g^2(1 + \log k - \log kq) + \frac{2k^2d^3(C\sigma)^2\sigma_g^6}{(k(C\sigma)^2 + \sigma_g^2)^2} \right)
$$

$$
= \mathcal{O}(k^2 + (1-q)k\sigma_g^4 + \sigma_g^2 k \log \frac{1}{q})
$$

where in the second last step we use the fact $\sum_{t=kq+1}^{k} 1/t^2 \leq \sum_{t=1}^{\infty} 1/t^2 = \pi^2/6$.

$$
Q = \sum_{t=tq+1}^{k} \sqrt{\mathbb{E}[\|z_t^*\|_2^4]}
$$

$$
= \sqrt{d(d+2)} \sum_{t=tq+1}^{k} \sigma_{t,r}^2
$$

$$
= \mathcal{O}(N)
$$

$$
= \mathcal{O}(k \log \frac{1}{q} + k(1-q)\sigma_g^2) .
$$

$\square$

**Lemma C.8.** Let $z_t^* := \sum_{l=1}^{t} \boldsymbol{X}_{t,l} z_l$ where $z_l \stackrel{\text{i.i.d.}}{\sim} \mathcal{N}(\boldsymbol{0}, k(C\sigma)^2 \boldsymbol{I}) \in \mathbb{R}^d$ are i.i.d. drawn. Further let $N, P, Q$ be defined equivalently as Lemma C.5. Then, we have $\mathbb{E}[(\sum_{t=1}^{k} \|z_t^*\|_2^2)^2] \leq P + N^2 + Q^2$.

*Proof.* For this proof we make use of two facts, namely, for two random variables $X, Y$, there is $\mathbb{E}[XY] = \mathbb{E}[X]\mathbb{E}[Y] - \text{Cov}[X, Y]$ and $\text{Cov}[X, Y] \leq \sqrt{\text{Var}[X]\text{Var}[Y]}$. Also notice that for any

random variable $X$, $\text{Var}[X] \leq \mathbb{E}[X^2]$. With these, we have

$$
\begin{aligned}
\mathbb{E}[(\sum_{t=1}^{k} \|z_t^*\|_2^2)^2] &= \sum_{t=1}^{k} \mathbb{E}[\|z_t^*\|_2^4] + \sum_{t=1}^{k}\sum_{t'=1}^{k} \mathbb{E}[\|z_t^*\|_2^2 \|z_{t'}^*\|_2^2] \\
&= \sum_{t=1}^{k} \mathbb{E}[\|z_t^*\|_2^4] + \sum_{t=1}^{k}\sum_{t'=1}^{k} \left( \mathbb{E}[\|z_t^*\|_2^2]\mathbb{E}[\|z_{t'}^*\|_2^2] + \text{Cov}[\|z_t^*\|_2^2, \|z_{t'}^*\|_2^2] \right) \\
&= \sum_{t=1}^{k} \mathbb{E}[\|z_t^*\|_2^4] + \sum_{t=1}^{k}\sum_{t'=1}^{k} \mathbb{E}[\|z_t^*\|_2^2]\mathbb{E}[\|z_{t'}^*\|_2^2] + \sum_{t=1}^{k}\sum_{t'=1}^{k} \text{Cov}[\|z_t^*\|_2^2, \|z_{t'}^*\|_2^2] \\
&\leq \sum_{t=1}^{k} \mathbb{E}[\|z_t^*\|_2^4] + \sum_{t=1}^{k}\sum_{t'=1}^{k} \mathbb{E}[\|z_t^*\|_2^2]\mathbb{E}[\|z_{t'}^*\|_2^2] + \sum_{t=1}^{k} \sqrt{\text{Var}[\|z_t^*\|_2^2]} \sum_{t'=1}^{k} \sqrt{\text{Var}[\|z_{t'}^*\|_2^2]} \\
&\leq \sum_{t=1}^{k} \mathbb{E}[\|z_t^*\|_2^4] + \sum_{t=1}^{k} \mathbb{E}[\|z_t^*\|_2^2] \sum_{t'=1}^{k} \mathbb{E}[\|z_{t'}^*\|_2^2] + \sum_{t=1}^{k} \sqrt{\mathbb{E}[\|z_t^*\|_2^4]} \sum_{t'=1}^{k} \sqrt{\mathbb{E}[\|z_{t'}^*\|_2^4]} \\
&= P + N^2 + Q^2 \ .
\end{aligned}
$$

$\square$

## C.5 A Hypothetical Case

**Proposition C.9.** $\forall t \in [k]$ and, denote $\boldsymbol{\theta}_{\pi^t}^* := \boldsymbol{\theta}_{\pi^t}^p - \alpha \sum_{l=1}^{t} \boldsymbol{X}_{t,l}\tilde{g}_{\pi^l}$. Assume that $\hat{g}_{\pi^1} = \hat{g}_{\pi^2} = \ldots = \hat{g}_{\pi^k}$. Denote $m^*(\pi^t) := V(\boldsymbol{\theta}_{\pi^t}^*) - V(\boldsymbol{\theta}_{\pi^t}^p)$ and $\psi^* := k^{-1}\sum_{t=1}^{k} p(\pi^t)m^*(\pi^t)$. Then $\boldsymbol{X}^*$ defined in Eq. (4) achieves an estimation uncertainty $\text{Var}[\psi^*|\boldsymbol{\theta}_{\pi^1}^p, \boldsymbol{\theta}_{\pi^2}^p, \ldots, \boldsymbol{\theta}_{\pi^k}^p] = (\log^2 k)$.

*Proof.* Denote $\bar{\boldsymbol{\theta}}_{\pi^t,j} := \boldsymbol{\theta}_{\pi^t,j}^p - \alpha \hat{g}_{\pi^t,j}$. As explained in the proof for Prop. 5.1, notice that $\bar{\boldsymbol{\theta}}_{\pi^t,j}$ is deterministic conditional on $\boldsymbol{\theta}_{\pi^t,j}^p$. Then, with $\boldsymbol{X}^*$,

$$
\begin{aligned}
\boldsymbol{\theta}_{\pi^t,j}^* &= \boldsymbol{\theta}_{\pi^t,j}^p - \alpha \sum_{l=1}^{t} \boldsymbol{X}_{t,l}^* \tilde{g}_{\pi^l,j} \\
&= (\boldsymbol{\theta}_{\pi^t,j}^p - \alpha \sum_{l=1}^{t} \boldsymbol{X}_{t,l}^* \hat{g}_{\pi^l,j}) - \alpha \sum_{l=1}^{t} \boldsymbol{X}_{t,l}^* \alpha z_l \\
&= (\boldsymbol{\theta}_{\pi^t,j}^p - \alpha \hat{g}_{\pi^t,j}) - \alpha \sum_{l=1}^{t} \boldsymbol{X}_{t,l}^* \alpha z_l \\
&= \bar{\boldsymbol{\theta}}_{\pi^t,j} - \alpha \sum_{l=1}^{t} \boldsymbol{X}_{t,l}^* \alpha z_l \\
&= \bar{\boldsymbol{\theta}}_{\pi^t,j} - \alpha \frac{\sum_{l=1}^{t} \alpha z_l}{t} \ .
\end{aligned}
$$

Since $\forall t \in [k], z_t \overset{\text{i.i.d.}}{\sim} \mathcal{N}(\boldsymbol{0}, k(C\sigma)^2\boldsymbol{I})$, we can denote $z_t^* := \frac{\sum_{l=1}^{t} z_l}{t} \sim \mathcal{N}(\boldsymbol{0}, \frac{k}{t}(C\sigma)^2\boldsymbol{I})$. Then, $\boldsymbol{\theta}_{\pi^t,j}^* = \boldsymbol{\theta}_{\pi^t,j} - \alpha z_t^*$.

Denote $\bar{p} := \max_{\pi \in \Pi} p(\pi)$. Consider that

$$
\text{Var}\left[ \frac{\sum_{t=1}^{k} p(\pi^t)V(\boldsymbol{\theta}_{\pi^t,j}^*)}{k} \middle| \bar{\boldsymbol{\theta}}_{\pi^1,j}, \ldots, \bar{\boldsymbol{\theta}}_{\pi^k,j} \right] = \frac{1}{k^2}\text{Var}\left[ \sum_{t=1}^{k} p(\pi^t)V(\boldsymbol{\theta}_{\pi^t,j}^*) \middle| \bar{\boldsymbol{\theta}}_{\pi^1,j}, \ldots, \bar{\boldsymbol{\theta}}_{\pi^k,j} \right] \ .
$$

By definition of smoothness, if $f$ is $L$-smooth, then $\forall x, y$,

$$
f(y) \leq f(x) + \nabla f(x)^\top (y - x) + \frac{L}{2}\|y - x\|_2^2 \ .
$$

Since $V$ is smooth, $-V$ is $(L\text{-})$smooth as well, which gives

$$-V(\bar{\boldsymbol{\theta}}_{\pi^t,j} - \alpha z_t^*) \leq -V(\bar{\boldsymbol{\theta}}_{\pi^t,j}) + \nabla_{\boldsymbol{\theta}} V(\bar{\boldsymbol{\theta}}_{\pi^t,j})^\top \alpha z_t^* + \frac{L}{2}\|\alpha z_t^*\|_2^2 \, .$$

Thus, we can bound the variance as

$$\mathrm{Var}\left[\sum_{t=1}^k p(\pi^t) V(\boldsymbol{\theta}_{\pi^t,j}^*)\Big|\bar{\boldsymbol{\theta}}_{\pi^1,j}, \ldots, \bar{\boldsymbol{\theta}}_{\pi^k,j}\right]$$

$$= \mathrm{Var}\left[\sum_{t=1}^k p(\pi^t) V(\bar{\boldsymbol{\theta}}_{\pi^t,j} - \alpha z_t^*)\Big|\bar{\boldsymbol{\theta}}_{\pi^1,j}, \ldots, \bar{\boldsymbol{\theta}}_{\pi^k,j}\right]$$

$$= \mathbb{E}\left[\left(\sum_{t=1}^k p(\pi^t) V(\bar{\boldsymbol{\theta}}_{\pi^t,j} - \alpha z_t^*)\right)^2\Big|\bar{\boldsymbol{\theta}}_{\pi^1,j}, \ldots, \bar{\boldsymbol{\theta}}_{\pi^k,j}\right]$$

$$- \mathbb{E}\left[\sum_{t=1}^k p(\pi^t) V(\bar{\boldsymbol{\theta}}_{\pi^t,j} - \alpha z_t^*)\Big|\bar{\boldsymbol{\theta}}_{\pi^1,j}, \ldots, \bar{\boldsymbol{\theta}}_{\pi^k,j}\right]^2$$

$$\leq \mathbb{E}\left[\left(\sum_{t=1}^k p(\pi^t) V(\bar{\boldsymbol{\theta}}_{\pi^t,j} - \alpha z_t^*)\right)^2\Big|\bar{\boldsymbol{\theta}}_{\pi^1,j}, \ldots, \bar{\boldsymbol{\theta}}_{\pi^k,j}\right]$$

$$\leq \underbrace{\mathbb{E}\left[\left(p(\pi^t)\sum_{t=1}^k -V(\bar{\boldsymbol{\theta}}_{\pi^t,j}) + \nabla_{\boldsymbol{\theta}} V(\bar{\boldsymbol{\theta}}_{\pi^t,j})^\top \alpha z_t^* + \frac{L}{2}\|\alpha z_t^*\|_2^2\right)^2\Big|\bar{\boldsymbol{\theta}}_{\pi^1,j}, \ldots, \bar{\boldsymbol{\theta}}_{\pi^k,j}\right]}_{E_1} \, .$$

Denote $M := \max_{\pi \in \Pi} -V(\boldsymbol{\theta}_{\pi,j})$ and $D := \max_{\pi \in \Pi} \|\nabla_{\boldsymbol{\theta}} V(\boldsymbol{\theta}_{\pi,j})\|_2^2$. With these results, the first part can be expanded as

$$E_1 \leq \bar{p}^2(\sum_{t=1}^k V(\bar{\boldsymbol{\theta}}_{\pi^t,j}))^2 + \mathbb{E}\left[\left(\sum_{t=1}^k p(\pi^t) \nabla_{\boldsymbol{\theta}} V(\bar{\boldsymbol{\theta}}_{\pi^t,j})^\top \alpha z_t^*\right)^2\Big|\bar{\boldsymbol{\theta}}_{\pi^1,j}, \ldots, \bar{\boldsymbol{\theta}}_{\pi^k,j}\right]$$

$$+ \frac{\bar{p}^2 L^2}{4}\mathbb{E}[(\sum_{t=1}^k \|\alpha z_t^*\|_2^2)^2] - \bar{p}^2 kL\sum_{t=1}^k V(\bar{\boldsymbol{\theta}}_{\pi^t,j})\mathbb{E}[\|\alpha z_t^*\|_2^2]$$

$$\leq \bar{p}^2(\sum_{t=1}^k V(\bar{\boldsymbol{\theta}}_{\pi^t,j}))^2 + \bar{p}^2\sum_{t=1}^k \mathbb{E}[\|\nabla_{\boldsymbol{\theta}} V(\bar{\boldsymbol{\theta}}_{\pi^t,j})\|_2^2\|\alpha z_t^*\|_2^2|\bar{\boldsymbol{\theta}}_{\pi^1,j}, \ldots, \bar{\boldsymbol{\theta}}_{\pi^k,j}]$$

$$+ \frac{\bar{p}^2 L^2}{4}\mathbb{E}[(\sum_{t=1}^k \|\alpha z_t^*\|_2^2)^2] - \bar{p}^2 kL\sum_{t=1}^k V(\bar{\boldsymbol{\theta}}_{\pi^t,j})\mathbb{E}[\|\alpha z_t^*\|_2^2]$$

$$= \bar{p}^2(\sum_{t=1}^k V(\bar{\boldsymbol{\theta}}_{\pi^t,j}))^2 + \bar{p}^2\sum_{t=1}^k \mathbb{E}[\|\nabla_{\boldsymbol{\theta}} V(\bar{\boldsymbol{\theta}}_{\pi^t,j})\|_2^2\|\alpha z_t^*\|_2^2] + \frac{\bar{p}^2 L^2}{4}\mathbb{E}[(\sum_{t=1}^k \|\alpha z_t^*\|_2^2)^2] - \bar{p}^2 kL\sum_{t=1}^k V(\bar{\boldsymbol{\theta}}_{\pi^t,j})\mathbb{E}[\|\alpha z_t^*\|_2^2] \, .$$

Here, applying Lemma C.5 and let $N = \sum_{t=1}^k \mathbb{E}[\|z_t^*\|_2^2], P = \sum_{t=1}^k \mathbb{E}[\|z_t^*\|_2^4], Q = \sum_{t=1}^k \sqrt{\mathbb{E}[\|z_t^*\|_2^4]}$. Additionally, by Lemma C.8, we have $\mathbb{E}[(\sum_{t=1}^k \|\alpha z_t^*\|_2^2)^2] \leq P + N^2 + Q^2$. Therefore, the above inequality can be bounded by

$$E_1 \leq \bar{p}^2 k^2 M^2 + \bar{p}^2 DN + \frac{\bar{p}^2 L^2}{4}(P + N^2 + Q^2) + \bar{p}^2 kLMN$$

$$= \mathcal{O}(k^2 + N + P + N^2 + Q^2 + kN)$$

where the minimum is attained at $\boldsymbol{X}^*$, with $N = \mathcal{O}(k \log k), P = \mathcal{O}(k^2), Q = \mathcal{O}(k \log k)$ and thus $\mathcal{O}(k^2 + N + P + N^2 + Q^2 + kN) = \mathcal{O}(k^2 \log^2 k)$. So, the variance of the average utility is

$$\mathrm{Var}\left[\frac{\sum_{t=1}^k p(\pi^t) V(\boldsymbol{\theta}_{\pi^t,j}^*)}{k}\Big|\bar{\boldsymbol{\theta}}_{\pi^1,j}, \ldots, \bar{\boldsymbol{\theta}}_{\pi^k,j}\right] = \frac{E_1}{k^2} = \frac{1}{k^2}\mathcal{O}(k^2 \log^2 k) = \mathcal{O}(\log^2 k)$$

by the multiplicative property of big-$\mathcal{O}$ notations. As such, for the semivalue estimator, we have

$$
\begin{aligned}
\text{Var}[\psi_j^* | \boldsymbol{\theta}_{\pi^1,j}^p, \boldsymbol{\theta}_{\pi^2,j}^p, \ldots, \boldsymbol{\theta}_{\pi^k,j}^p] &= \text{Var}\left[\left.\frac{\sum_{t=1}^k p(\pi^t)[V(\boldsymbol{\theta}_{\pi^t,j}^*) - V(\boldsymbol{\theta}_{\pi^t,j}^p)]}{k}\right| \boldsymbol{\theta}_{\pi^1,j}^p, \boldsymbol{\theta}_{\pi^2,j}^p, \ldots, \boldsymbol{\theta}_{\pi^k,j}^p\right] \\
&= \text{Var}\left[\left.\frac{\sum_{t=1}^k p(\pi^t)V(\boldsymbol{\theta}_{\pi^t,j}^*)}{k}\right| \bar{\boldsymbol{\theta}}_{\pi^1,j}, \ldots, \bar{\boldsymbol{\theta}}_{\pi^k,j}\right] \\
&= \mathcal{O}(\log^2 k) \,.
\end{aligned}
$$

$\square$

## C.6 Proof of Prop. 5.3.

**Proposition C.10.** (Reproduced from Prop. 5.3, *formal*) Let $\tilde{g}_{\pi^l}, l \in [t]$ be perturbed gradients using the Gaussian mechanism that satisfies $(\epsilon, \delta)$-DP. $\forall t \in [k]$, denote $\boldsymbol{\theta}_{\pi^t}^* := \boldsymbol{\theta}_{\pi^t}^p - \alpha \sum_{l=1}^t \boldsymbol{X}_{t,l} \tilde{g}_{\pi^l}$, $m^*(\pi^t) := V(\boldsymbol{\theta}_{\pi^t}^*) - V(\boldsymbol{\theta}_{\pi^t}^p)$ and $\psi^* := k^{-1} \sum_{t=1}^k p(\pi^t) m^*(\pi^t)$. Assume that $\forall t \in [k]$, $\hat{g}_{\pi^t} - \mathbb{E}[\hat{g}_{\pi^t}]$ i.i.d. follow an diagonal multivariate sub-Gaussian distribution with covariance $\Sigma \in \mathbb{R}^{(d \times d)}$ and let $\sigma_g^2 := 1/d \sum_{r=1}^d \Sigma_{r,r}$. Then the matrix satisfying $\forall t \in [k], \forall l \in [t-1], \boldsymbol{X}_{t,l} = t^{-1}(k(C\sigma)^2 + \sigma_g^2)^{-1} k(C\sigma)^2$ and $\forall t \in [k], \boldsymbol{X}_{t,t} = t^{-1}(k(C\sigma)^2 + \sigma_g^2)^{-1}(k(C\sigma)^2 + t\sigma_g^2)$ produces an estimation uncertainty $\text{Var}[\psi^* | \boldsymbol{\theta}_{\pi^1}^p, \boldsymbol{\theta}_{\pi^2}^p, \ldots, \boldsymbol{\theta}_{\pi^k}^p] = \mathcal{O}(\log^2 k + \sigma_g^4)$ and $\mathbb{E}[\psi^* - \psi] = \mathcal{O}(\log k + \sigma_g^2)$ while satisfying $(\epsilon, \delta)$-DP. Moreover, as $k \to \infty$, $\boldsymbol{X} \to \boldsymbol{X}^*$.

*Proof.* Denote $\bar{\boldsymbol{\theta}}_{\pi^t,j} := \boldsymbol{\theta}_{\pi^t,j}^p - \alpha \hat{g}_{\pi^t,j}$. Note that as compared to the case of Prop. C.9, $\hat{g}_{\pi^t,j}$ is now a random variable (coming from an diagonal sub-Gaussian distribution). We may denote the mean of the distribution as $\mu_g$. Then $\hat{g}_{\pi^t,j} = \mu_g + \zeta_t'$ where $\zeta_t' \overset{\text{i.i.d.}}{\sim}$ sub-Gaussian$(\boldsymbol{0}, \Sigma)$ with $\Sigma$ a diagonal covariance matrix with $\sigma_g^2 = \max_{r \in [d]}(\Sigma)_{r,r}$. As such, $\forall t \in [k]$, denote

$$
z_t'^* := (\bar{\boldsymbol{\theta}}_{\pi^t,j} - \boldsymbol{\theta}_{\pi^t,j}^*)/\alpha = -\hat{g}_{\pi^t,j} + \sum_{l=1}^t \boldsymbol{X}_{t,l} \tilde{g}_{t,l} = -\zeta_t' + \sum_{l=1}^t \boldsymbol{X}_{t,l}(z_l + \zeta_l') \,.
$$

We use the Gaussian distribution to bound the moments of $z_t'^*$. To do this, first define a Gaussian counterpart of $\zeta_t'$, $\zeta_t \sim \mathcal{N}(\boldsymbol{0}, \Sigma)$. Then denote

$$
z_t^* := -\zeta_t + \sum_{l=1}^t \boldsymbol{X}_{t,l}(z_l + \zeta_l) \,.
$$

By properties of sub-Gaussian distribution, we have that $z_t'^*$ follows a sub-Gaussian with the same mean and variance as $z_t^*$. Hence, $\mathbb{E}[\|z_t'^*\|_2^2] = \mathbb{E}[\|z_t^*\|_2^2]$ and $\mathbb{E}[\|z_t'^*\|_2^4] \leq \frac{16}{3} \mathbb{E}[\|z_t^*\|_2^4]$. The second result is a direct consequence of (Rigollet, 2015, Lemma 1.4) and (Balakrishnan, 2016, Section 7.2). Specifically, let the variance of $z_t'^*$ and $z_t^*$ be $\sigma_z^2$. (Rigollet, 2015, Lemma 1.4) states that $\mathbb{E}[\|z_t'^*\|_2^4] \leq 16\sigma_z^4$ and (Balakrishnan, 2016, Section 7.2) states that $\mathbb{E}[\|z_t^*\|_2^4] = 3\sigma_z^4$.

Further denote $\bar{p} := \max_{\pi \in \Pi} p(\pi)$. Consider that

$$
\begin{aligned}
\text{Var}\left[\left.\frac{\sum_{t=1}^k p(\pi^t)V(\boldsymbol{\theta}_{\pi^t,j}^*)}{k}\right| \bar{\boldsymbol{\theta}}_{\pi^1,j}, \ldots, \bar{\boldsymbol{\theta}}_{\pi^k,j}\right] &= \frac{1}{k^2} \text{Var}\left[\left.\sum_{t=1}^k p(\pi^t)V(\boldsymbol{\theta}_{\pi^t,j}^*)\right| \bar{\boldsymbol{\theta}}_{\pi^1,j}, \ldots, \bar{\boldsymbol{\theta}}_{\pi^k,j}\right] \\
&= \frac{1}{k^2} \text{Var}\left[\left.\sum_{t=1}^k p(\pi^t)V(\boldsymbol{\theta}_{\pi^t,j}^*)\right| \bar{\boldsymbol{\theta}}_{\pi^1,j}, \ldots, \bar{\boldsymbol{\theta}}_{\pi^k,j}\right] \\
&= \frac{1}{k^2} \text{Var}\left[\left.\sum_{t=1}^k p(\pi^t)V(\bar{\boldsymbol{\theta}}_{\pi^t,j} - \alpha z_t^*)\right| \bar{\boldsymbol{\theta}}_{\pi^1,j}, \ldots, \bar{\boldsymbol{\theta}}_{\pi^k,j}\right] \,.
\end{aligned}
$$

By definition of smoothness, if $f$ is $L$-smooth, then $\forall x, y$,

$$
f(y) \leq f(x) + \nabla f(x)^\top (y - x) + \frac{L}{2} \|y - x\|_2^2 \,.
$$

Since $V$ is smooth, $-V$ is $(L\text{-})$smooth as well, which gives

$$0 \le -V(\bar{\boldsymbol{\theta}}_{\pi^t,j} - \alpha z_t'^*) \le -V(\bar{\boldsymbol{\theta}}_{\pi^t,j}) + \nabla_{\boldsymbol{\theta}} V(\bar{\boldsymbol{\theta}}_{\pi^t,j})^\top \alpha z_t'^* + \frac{L}{2}\|\alpha z_t'^*\|_2^2 .$$

As such, we have, for the bias, we have

$$\mathbb{E}[\psi^* - \psi] = \mathbb{E}\left[\mathbb{E}\left[\frac{\sum_{t=1}^k p(\pi^t)[V(\boldsymbol{\theta}_{\pi^t,j}^*) - V(\bar{\boldsymbol{\theta}}_{\pi^t,j})]}{k}\Big|\bar{\boldsymbol{\theta}}_{\pi^1,j},\ldots,\bar{\boldsymbol{\theta}}_{\pi^k,j}\right]\right]$$

$$\le \mathbb{E}\left[\mathbb{E}\left[\frac{\sum_{t=1}^k \nabla_{\boldsymbol{\theta}} V(\bar{\boldsymbol{\theta}}_{\pi^t,j})^\top \alpha z_t'^* + \frac{L}{2}\|\alpha z_t'^*\|_2^2}{k}\Big|\bar{\boldsymbol{\theta}}_{\pi^1,j},\ldots,\bar{\boldsymbol{\theta}}_{\pi^k,j}\right]\right]$$

$$= \mathbb{E}\left[\sum_{t=1}^k \frac{\frac{L}{2}\|\alpha z_t'^*\|_2^2}{k}\right] .$$

By Lemma C.6, we have

$$\mathbb{E}[\sum_{t=1}^k \|\alpha z_t'^*\|_2^2] = N = \mathcal{O}(k\log k + k\sigma_g^2) .$$

Therefore, we have the bound on the bias as $\mathbb{E}[\psi^* - \psi] = \mathbb{E}[\sum_{t=1}^k \|\alpha z_t'^*\|_2^2]/k = \mathcal{O}(\log k + \sigma_g^2)$.

Consider that the variance is upper-bounded as

$$\text{Var}\left[\sum_{t=1}^k p(\pi^t)V(\bar{\boldsymbol{\theta}}_{\pi^t,j} - \alpha z_t'^*)\Big|\bar{\boldsymbol{\theta}}_{\pi^1,j},\ldots,\bar{\boldsymbol{\theta}}_{\pi^k,j}\right]$$

$$= \mathbb{E}\left[\left(\sum_{t=1}^k p(\pi^t)V(\bar{\boldsymbol{\theta}}_{\pi^t,j} - \alpha z_t'^*)\right)^2\Big|\bar{\boldsymbol{\theta}}_{\pi^1,j},\ldots,\bar{\boldsymbol{\theta}}_{\pi^k,j}\right] - \mathbb{E}\left[\sum_{t=1}^k p(\pi^t)V(\bar{\boldsymbol{\theta}}_{\pi^t,j} - \alpha z_t'^*)\Big|\bar{\boldsymbol{\theta}}_{\pi^1,j},\ldots,\bar{\boldsymbol{\theta}}_{\pi^k,j}\right]^2$$

$$\le \underbrace{\mathbb{E}\left[\left(\sum_{t=1}^k p(\pi^t)V(\bar{\boldsymbol{\theta}}_{\pi^t,j} - \alpha z_t'^*)\right)^2\Big|\bar{\boldsymbol{\theta}}_{\pi^1,j},\ldots,\bar{\boldsymbol{\theta}}_{\pi^k,j}\right]}_{E_1} .$$

Since $V$ is strictly non-positive, $-V(\bar{\boldsymbol{\theta}}_{\pi^t,j} - \alpha z_t'^*) \geq 0$. Denote $M := \max_{\pi \in \Pi} -V(\boldsymbol{\theta}_{\pi,j})$ and $D := \max_{\pi \in \Pi} \|\nabla_{\boldsymbol{\theta}} V(\boldsymbol{\theta}_{\pi,j})\|_2^2$. We have

$$E_1 \leq \mathbb{E}\left[\left(\sum_{t=1}^k -p(\pi^t)V(\bar{\boldsymbol{\theta}}_{\pi^t,j}) + p(\pi^t)\nabla_{\boldsymbol{\theta}}V(\bar{\boldsymbol{\theta}}_{\pi^t,j})^\top \alpha z_t'^* + \frac{p(\pi^t)L}{2}\|\alpha z_t'^*\|_2^2\right)^2 \Bigg| \bar{\boldsymbol{\theta}}_{\pi^1,j}, \ldots, \bar{\boldsymbol{\theta}}_{\pi^k,j}\right]$$

$$= \left(\sum_{t=1}^k -p(\pi^t)V(\bar{\boldsymbol{\theta}}_{\pi^t,j})\right)^2 + \mathbb{E}\left[\left(\sum_{t=1}^k p(\pi^t)\nabla_{\boldsymbol{\theta}}V(\bar{\boldsymbol{\theta}}_{\pi^t,j})^\top \alpha z_t'^*\right)^2 \Bigg| \bar{\boldsymbol{\theta}}_{\pi^1,j}, \ldots, \bar{\boldsymbol{\theta}}_{\pi^k,j}\right]$$

$$+ \mathbb{E}\left[\left(\frac{p(\pi^t)L}{2}\sum_{t=1}^k \|\alpha z_t'^*\|_2^2\right)^2\right] - kL\sum_{t=1}^k p(\pi^t)^2 V(\bar{\boldsymbol{\theta}}_{\pi^t,j})\mathbb{E}[\|\alpha z_t'^*\|_2^2]$$

$$\leq \bar{p}^2\left(\sum_{t=1}^k -V(\bar{\boldsymbol{\theta}}_{\pi^t,j})\right)^2 + \bar{p}^2\sum_{t=1}^k \mathbb{E}[\|\nabla_{\boldsymbol{\theta}}V(\bar{\boldsymbol{\theta}}_{\pi^t,j})\|_2^2\|\alpha z_t'^*\|_2^2|\bar{\boldsymbol{\theta}}_{\pi^1,j}, \ldots, \bar{\boldsymbol{\theta}}_{\pi^k,j}]$$

$$+ \frac{\bar{p}^2L^2}{4}\mathbb{E}\left[\left(\sum_{t=1}^k \|\alpha z_t'^*\|_2^2\right)^2\right] - \bar{p}^2 kL\sum_{t=1}^k V(\bar{\boldsymbol{\theta}}_{\pi^t,j})\mathbb{E}[\|\alpha z_t'^*\|_2^2]$$

$$\leq \bar{p}^2\left(\sum_{t=1}^k -V(\bar{\boldsymbol{\theta}}_{\pi^t,j})\right)^2 + \bar{p}^2 D\sum_{t=1}^k \mathbb{E}[\|\alpha z_t'^*\|_2^2] + \frac{\bar{p}^2L^2}{4}\mathbb{E}\left[\left(\sum_{t=1}^k \|\alpha z_t'^*\|_2^2\right)^2\right] - \bar{p}^2 kL\sum_{t=1}^k V(\bar{\boldsymbol{\theta}}_{\pi^t,j})\mathbb{E}[\|\alpha z_t'^*\|_2^2]$$

$$\leq \bar{p}^2 k^2 M^2 + \bar{p}^2(D + kLM)\sum_{t=1}^k \mathbb{E}[\|\alpha z_t'^*\|_2^2] + \frac{\bar{p}^2L^2}{4}\mathbb{E}[(\sum_{t=1}^k \|\alpha z_t'^*\|_2^2)^2] .$$

Let $N = \sum_{t=1}^k \mathbb{E}[\|z_t^*\|_2^2] \geq \sum_{t=1}^k \mathbb{E}[\|z_t'^*\|_2^2], P = \sum_{t=1}^k \mathbb{E}[\|z_t^*\|_2^4] \geq \sum_{t=1}^k \mathbb{E}[\|z_t'^*\|_2^4], Q = \sum_{t=1}^k \sqrt{\mathbb{E}[\|z_t^*\|_2^4]} \geq \sqrt{3/16}\sum_{t=1}^k \sqrt{\mathbb{E}[\|z_t'^*\|_2^4]}$. Additionally, by Lemma C.8, we have $\mathbb{E}[(\sum_{t=1}^k \|\alpha z_t'^*\|_2^2)^2] \leq \frac{16}{3}\mathbb{E}[(\sum_{t=1}^k \|\alpha z_t^*\|_2^2)^2] \leq \frac{16}{3}(P + N^2 + Q^2)$. By Lemma C.6, we have the matrix satisfying $\forall t \in [k], \forall l \in [t-1], \boldsymbol{X}_{t,l} = t^{-1}(k(C\sigma)^2 + \sigma_g^2)^{-1}k(C\sigma)^2$ and $\forall t \in [k], \boldsymbol{X}_{t,t} = t^{-1}(k(C\sigma)^2 + \sigma_g^2)^{-1}(k(C\sigma)^2 + t\sigma_g^2)$ produces the upper bound of $N, P, Q$ with

$$N = \mathcal{O}(k\sigma_g^2 + k\log k),$$
$$P = \mathcal{O}(k^2 + k\sigma_g^4 + \sigma_g^2 k\log k),$$
$$Q = \mathcal{O}(k\log k + k\sigma_g^2).$$

As such, we can further simply $E_1$ as

$$E_1 = \mathcal{O}(k^2 + kN + P + N^2 + Q^2) = \mathcal{O}(k^2\sigma_g^4 + k^2\log^2 k).$$

Therefore, we can bound the variance as

$$\text{Var}\left[\frac{\sum_{t=1}^k p(\pi^t)V(\boldsymbol{\theta}_{\pi^t,j}^*)}{k}\Bigg| \bar{\boldsymbol{\theta}}_{\pi^1,j}, \ldots, \bar{\boldsymbol{\theta}}_{\pi^k,j}\right] \leq \frac{1}{k^2}E_1$$

$$= \frac{1}{k^2}\mathcal{O}(k^2\sigma_g^4 + k^2\log^2 k) = \mathcal{O}(\log^2 k + \sigma_g^4)$$

by the multiplicative property of big-$\mathcal{O}$ notation. As such, for the semivalue estimator, we have

$$\text{Var}[\psi_j^*|\boldsymbol{\theta}_{\pi^1,j}^p, \boldsymbol{\theta}_{\pi^2,j}^p, \ldots, \boldsymbol{\theta}_{\pi^k,j}^p] = \text{Var}\left[\frac{\sum_{t=1}^k p(\pi^t)[V(\boldsymbol{\theta}_{\pi^t,j}^*) - V(\boldsymbol{\theta}_{\pi^t,j}^p)]}{k}\Bigg| \boldsymbol{\theta}_{\pi^1,j}^p, \boldsymbol{\theta}_{\pi^2,j}^p, \ldots, \boldsymbol{\theta}_{\pi^k,j}^p\right]$$

$$= \text{Var}\left[\frac{\sum_{t=1}^k p(\pi^t)V(\boldsymbol{\theta}_{\pi^t,j}^*)}{k}\Bigg| \bar{\boldsymbol{\theta}}_{\pi^1,j}, \bar{\boldsymbol{\theta}}_{\pi^2,j}, \ldots, \bar{\boldsymbol{\theta}}_{\pi^k,j}\right]$$

$$= \mathcal{O}(\log^2 k + \sigma_g^4)$$

where the change in conditioning in the 2nd step is because $\bar{\boldsymbol{\theta}}_{\pi^t,j}$ is deterministic conditional on $\boldsymbol{\theta}^p_{\pi^t,j}$, explained in more detail in the proof for Prop. 5.1. As the operations are all applied to $\tilde{g}$, by the post-processing immunity of DP, the same $(\epsilon, \delta)$-DP guarantee holds. With the given $\boldsymbol{X}$, it is easy to see that as $k \to \infty$, $\boldsymbol{X} \to \boldsymbol{X}^*$.

$\square$

### C.7 PROOF OF PROP. 5.4.

**Proposition C.11.** (Reproduced from Prop. 5.4, *formal*) Let $\tilde{g}_{\pi^l}, l \in [t]$ be perturbed gradients using the Gaussian mechanism that satisfies $(\epsilon, \delta)$-DP. $\forall t \in \{kq + 1, \ldots, k\}$, denote $\boldsymbol{\theta}^*_{\pi^t} \coloneqq \boldsymbol{\theta}^p_{\pi^t} - \alpha \sum_{l=1}^t \boldsymbol{Y}_{t-kq,l} \tilde{g}_{\pi^l}$. Denote $m^*(\pi^t) \coloneqq V(\boldsymbol{\theta}^*_{\pi^t}) - V(\boldsymbol{\theta}^p_{\pi^t})$ and $\psi^* \coloneqq (k - kq)^{-1} \sum_{t=kq+1}^k p(\pi^t) m^*(\pi^t)$ for $q \in (0, 1)$. Assume that $\forall t \in [k], \hat{g}_{\pi^t} - \mathbb{E}[\hat{g}_{\pi^t}]$ i.i.d. follow an diagonal multivariate sub-Gaussian distribution with covariance $\Sigma \in \mathbb{R}^{(d \times d)}$ and let $\sigma_g^2 \coloneqq 1/d \sum_{r=1}^d \Sigma_{r,r}$. Then the matrix satisfying $\forall t \in [k], \forall l \in [t-1], \boldsymbol{X}_{t,l} = t^{-1}(k(C\sigma)^2 + \sigma_g^2)^{-1} k(C\sigma)^2$ and $\forall t \in [k], \boldsymbol{X}_{t,t} = t^{-1}(k(C\sigma)^2 + \sigma_g^2)^{-1}(k(C\sigma)^2 + t\sigma_g^2)$ produces an estimation uncertainty $\text{Var}[\psi^* | \boldsymbol{\theta}^p_{\pi^1}, \boldsymbol{\theta}^p_{\pi^2}, \ldots, \boldsymbol{\theta}^p_{\pi^k}] = \mathcal{O}\left((1-q)^{-2} \log^2(1/q) + \sigma_g^4\right)$ and $\mathbb{E}[\psi^* - \psi] = \mathcal{O}((1-q)\log 1/q + \sigma_g^2)$ while satisfying $(\epsilon, \delta)$-DP.

*Proof.* The proof largely follows the proof for Prop. 5.3. Denote $\bar{\boldsymbol{\theta}}_{\pi^t,j} \coloneqq \boldsymbol{\theta}^p_{\pi^t,j} - \alpha \hat{g}_{\pi^t,j}$. Note that as compared to the case of Prop. C.9, $\hat{g}_{\pi^t,j}$ is now a random variable (coming from an diagonal sub-Gaussian distribution). We may denote the mean of the distribution as $\mu_g$. Then $\hat{g}_{\pi^t,j} = \mu_g + \zeta_t'$ where $\zeta_t' \overset{\text{i.i.d.}}{\sim} \text{sub-Gaussian}(\boldsymbol{0}, \Sigma)$ with $\Sigma$ a diagonal covariance matrix with $\sigma_g^2 = \max_{r \in [d]}(\Sigma)_{r,r}$. As such, $\forall t \in [k]$, denote

$$z_t'^* \coloneqq (\bar{\boldsymbol{\theta}}_{\pi^t,j} - \boldsymbol{\theta}^*_{\pi^t,j})/\alpha = -\hat{g}_{\pi^t,j} + \sum_{l=1}^t \boldsymbol{X}_{t,l} \tilde{g}_{t,l} = -\zeta_t' + \sum_{l=1}^t \boldsymbol{X}_{t,l}(z_l + \zeta_l').$$

We use the Gaussian distribution to bound the moments of $z_t'^*$. To do this, first define a Gaussian counterpart of $\zeta_t'$, $\zeta_t \sim \mathcal{N}(\boldsymbol{0}, \Sigma)$. Then denote

$$z_t^* \coloneqq -\zeta_t + \sum_{l=1}^t \boldsymbol{X}_{t,l}(z_l + \zeta_l).$$

By properties of sub-Gaussian distribution, we have that $z_t'^*$ follows a sub-Gaussian with the same mean and variance as $z_t^*$. Hence, $\mathbb{E}[\|z_t'^*\|_2^2] = \mathbb{E}[\|z_t^*\|_2^2]$ and $\mathbb{E}[\|z_t'^*\|_2^4] \leq \frac{16}{3}\mathbb{E}[\|z_t^*\|_2^4]$. The second result is a direct consequence of (Rigollet, 2015, Lemma 1.4) and (Balakrishnan, 2016, Section 7.2). Specifically, let the variance of $z_t'^*$ and $z_t^*$ be $\sigma_z^2$. (Rigollet, 2015, Lemma 1.4) states that $\mathbb{E}[\|z_t'^*\|_2^4] \leq 16\sigma_z^4$ and (Balakrishnan, 2016, Section 7.2) states that $\mathbb{E}[\|z_t^*\|_2^4] = 3\sigma_z^4$.

Further denote $\bar{p} \coloneqq \max_{\pi \in \Pi} p(\pi)$. Consider that

$$\text{Var}\left[\frac{\sum_{t=kq+1}^k p(\pi^t) V(\boldsymbol{\theta}^*_{\pi^t,j})}{k - kq} \bigg| \bar{\boldsymbol{\theta}}_{\pi^{kq+1},j}, \ldots, \bar{\boldsymbol{\theta}}_{\pi^k,j}\right] = \frac{1}{(k-kq)^2} \text{Var}\left[\sum_{t=kq+1}^k p(\pi^t) V(\boldsymbol{\theta}^*_{\pi^t,j}) \bigg| \bar{\boldsymbol{\theta}}_{\pi^{kq+1},j}, \ldots, \bar{\boldsymbol{\theta}}_{\pi^k,j}\right]$$

$$= \frac{1}{(k-kq)^2} \text{Var}\left[\sum_{t=kq+1}^k p(\pi^t) V(\boldsymbol{\theta}^*_{\pi^t,j}) \bigg| \bar{\boldsymbol{\theta}}_{\pi^{kq+1},j}, \ldots, \bar{\boldsymbol{\theta}}_{\pi^k,j}\right]$$

$$= \frac{1}{(k-kq)^2} \text{Var}\left[\sum_{t=kq+1}^k p(\pi^t) V(\bar{\boldsymbol{\theta}}_{\pi^t,j} - \alpha z_t'^*) \bigg| \bar{\boldsymbol{\theta}}_{\pi^{kq+1},j}, \ldots, \bar{\boldsymbol{\theta}}_{\pi^k,j}\right].$$

Since $-V$ is $L$-smooth, following the proof for Prop. C.9, we have

$$0 \leq -V(\bar{\boldsymbol{\theta}}_{\pi^t,j} - \alpha z_t'^*) \leq -V(\bar{\boldsymbol{\theta}}_{\pi^t,j}) + \nabla_{\boldsymbol{\theta}} V(\bar{\boldsymbol{\theta}}_{\pi^t,j})^\top \alpha z_t'^* + \frac{L}{2}\|\alpha z_t'^*\|_2^2.$$

As such, we have, for the bias, we have

$$
\begin{aligned}
\mathbb{E}[\psi^* - \psi] &= \mathbb{E}\left[\mathbb{E}\left[\frac{\sum_{t=kq+1}^{k} p(\pi^t)[V(\boldsymbol{\theta}^*_{\pi^t,j}) - V(\bar{\boldsymbol{\theta}}_{\pi^t,j})]}{k - kq}\middle| \bar{\boldsymbol{\theta}}_{\pi^1,j}, \ldots, \bar{\boldsymbol{\theta}}_{\pi^k,j}\right]\right] \\
&\leq \mathbb{E}\left[\mathbb{E}\left[\frac{\sum_{t=kq+1}^{k} \nabla_{\boldsymbol{\theta}} V(\bar{\boldsymbol{\theta}}_{\pi^t,j})^\top \alpha z_t'^* + \frac{L}{2}\|\alpha z_t'^*\|_2^2}{k - kq}\middle| \bar{\boldsymbol{\theta}}_{\pi^1,j}, \ldots, \bar{\boldsymbol{\theta}}_{\pi^k,j}\right]\right] \\
&= \mathbb{E}\left[\sum_{t=kq+1}^{k} \frac{\frac{L}{2}\|\alpha z_t'^*\|_2^2}{k - kq}\right].
\end{aligned}
$$

By Corollary C.7, we have

$$
\mathbb{E}[\sum_{t=kq+1}^{k} \|\alpha z_t'^*\|_2^2] = N = \mathcal{O}(k \log 1/q + k(1-q)\sigma_g^2).
$$

Therefore, we have the bound on the bias as $\mathbb{E}[\psi^* - \psi] = \mathbb{E}[\sum_{t=kq+1}^{k} \|\alpha z_t'^*\|_2^2]/(k - kq) = \mathcal{O}((1-q)\log 1/q + \sigma_g^2)$.

Consider that the variance is upper-bounded as

$$
\begin{aligned}
&\mathrm{Var}\left[\sum_{t=kq+1}^{k} p(\pi^t) V(\bar{\boldsymbol{\theta}}_{\pi^t,j} - \alpha z_t'^*)\middle| \bar{\boldsymbol{\theta}}_{\pi^{kq+1},j}, \ldots, \bar{\boldsymbol{\theta}}_{\pi^k,j}\right] \\
&= \mathbb{E}\left[\left(\sum_{t=kq+1}^{k} p(\pi^t) V(\bar{\boldsymbol{\theta}}_{\pi^t,j} - \alpha z_t'^*)\right)^2\middle| \bar{\boldsymbol{\theta}}_{\pi^{kq+1},j}, \ldots, \bar{\boldsymbol{\theta}}_{\pi^k,j}\right] - \mathbb{E}\left[\sum_{t=kq+1}^{k} p(\pi^t) V(\bar{\boldsymbol{\theta}}_{\pi^t,j} - \alpha z_t'^*)\middle| \bar{\boldsymbol{\theta}}_{\pi^{kq+1},j}, \ldots, \bar{\boldsymbol{\theta}}_{\pi^k,j}\right]^2 \\
&\leq \underbrace{\mathbb{E}\left[\left(\sum_{t=kq+1}^{k} p(\pi^t) V(\bar{\boldsymbol{\theta}}_{\pi^t,j} - \alpha z_t'^*)\right)^2\middle| \bar{\boldsymbol{\theta}}_{\pi^{kq+1},j}, \ldots, \bar{\boldsymbol{\theta}}_{\pi^k,j}\right]}_{E_1}.
\end{aligned}
$$

Since $V$ is strictly non-positive, $-V(\bar{\boldsymbol{\theta}}_{\pi^t,j} - \alpha z_t'^*) \geq 0$. Denote $M := \max_{\pi \in \Pi} -V(\boldsymbol{\theta}_{\pi,j})$ and $D := \max_{\pi \in \Pi} \|\nabla_{\boldsymbol{\theta}} V(\boldsymbol{\theta}_{\pi,j})\|_2^2$. We have

$$
E_1 \leq \mathbb{E}\left[\left(\sum_{t=kq+1}^{k} -p(\pi^t)V(\bar{\boldsymbol{\theta}}_{\pi^t,j}) + p(\pi^t)\nabla_{\boldsymbol{\theta}} V(\bar{\boldsymbol{\theta}}_{\pi^t,j})^{\top}\alpha z_t'^* + \frac{p(\pi^t)L}{2}\|\alpha z_t'^*\|_2^2\right)^2 \Bigg| \bar{\boldsymbol{\theta}}_{\pi^{kq+1},j},\ldots,\bar{\boldsymbol{\theta}}_{\pi^k,j}\right]
$$

$$
= (\sum_{t=kq+1}^{k} -p(\pi^t)V(\bar{\boldsymbol{\theta}}_{\pi^t,j}))^2 + \mathbb{E}\left[\left(\sum_{t=kq+1}^{k} p(\pi^t)\nabla_{\boldsymbol{\theta}} V(\bar{\boldsymbol{\theta}}_{\pi^t,j})^{\top}\alpha z_t'^*\right)^2 \Bigg| \bar{\boldsymbol{\theta}}_{\pi^{kq+1},j},\ldots,\bar{\boldsymbol{\theta}}_{\pi^k,j}\right]
$$

$$
+ \mathbb{E}\left[\left(\frac{p(\pi^t)L}{2}\sum_{t=kq+1}^{k}\|\alpha z_t'^*\|_2^2\right)^2\right] - kL\sum_{t=kq+1}^{k} p(\pi^t)^2 V(\bar{\boldsymbol{\theta}}_{\pi^t,j})\mathbb{E}[\|\alpha z_t'^*\|_2^2]
$$

$$
\leq \bar{p}^2(\sum_{t=kq+1}^{k} -V(\bar{\boldsymbol{\theta}}_{\pi^t,j}))^2 + \bar{p}^2\sum_{t=kq+1}^{k} \mathbb{E}[\|\nabla_{\boldsymbol{\theta}} V(\bar{\boldsymbol{\theta}}_{\pi^t,j})\|_2^2\|\alpha z_t'^*\|_2^2|\bar{\boldsymbol{\theta}}_{\pi^{kq+1},j},\ldots,\bar{\boldsymbol{\theta}}_{\pi^k,j}]
$$

$$
+ \frac{\bar{p}^2 L^2}{4}\mathbb{E}\left[\left(\sum_{t=kq+1}^{k}\|\alpha z_t'^*\|_2^2\right)^2\right] - \bar{p}^2 kL\sum_{t=kq+1}^{k} V(\bar{\boldsymbol{\theta}}_{\pi^t,j})\mathbb{E}[\|\alpha z_t'^*\|_2^2]
$$

$$
\leq \bar{p}^2(\sum_{t=kq+1}^{k} -V(\bar{\boldsymbol{\theta}}_{\pi^t,j}))^2 + \bar{p}^2 D\sum_{t=kq+1}^{k} \mathbb{E}[\|\alpha z_t'^*\|_2^2] + \frac{\bar{p}^2 L^2}{4}\mathbb{E}\left[\left(\sum_{t=kq+1}^{k}\|\alpha z_t'^*\|_2^2\right)^2\right]
$$

$$
- \bar{p}^2 kL\sum_{t=kq+1}^{k} V(\bar{\boldsymbol{\theta}}_{\pi^t,j})\mathbb{E}[\|\alpha z_t'^*\|_2^2]
$$

$$
\leq \bar{p}^2(k-kq)^2 M^2 + \bar{p}^2(D + (k-kq)LM)\sum_{t=kq+1}^{k} \mathbb{E}[\|\alpha z_t'^*\|_2^2] + \frac{\bar{p}^2 L^2}{4}\mathbb{E}[(\sum_{t=kq+1}^{k}\|\alpha z_t'^*\|_2^2)^2].
$$

Let $N = \sum_{t=kq+1}^{k} \mathbb{E}[\|z_t^*\|_2^2] \geq \sum_{t=kq+1}^{k} \mathbb{E}[\|z_t'^*\|_2^2], P = \sum_{t=kq+1}^{k} \mathbb{E}[\|z_t^*\|_2^4] \geq \sum_{t=kq+1}^{k} \mathbb{E}[\|z_t'^*\|_2^4], Q = \sum_{t=kq+1}^{k} \sqrt{\mathbb{E}[\|z_t^*\|_2^4]} \geq \sqrt{3/16}\sum_{t=kq+1}^{k} \sqrt{\mathbb{E}[\|z_t'^*\|_2^4]}$. Additionally, by Lemma C.8, we have $\mathbb{E}[(\sum_{t=kq+1}^{k}\|\alpha z_t'^*\|_2^2)^2] \leq \frac{16}{3}\mathbb{E}[(\sum_{t=kq+1}^{k}\|\alpha z_t^*\|_2^2)^2] \leq \frac{16}{3}(P + N^2 + Q^2)$. By Corollary C.7, we have the matrix satisfying $\forall t \in [k], \forall l \in [t-1], \boldsymbol{X}_{t,l} = t^{-1}(k(C\sigma)^2 + \sigma_g^2)^{-1}k(C\sigma)^2$ and $\forall t \in [k], \boldsymbol{X}_{t,t} = t^{-1}(k(C\sigma)^2 + \sigma_g^2)^{-1}(k(C\sigma)^2 + t\sigma_g^2)$ produces an upper bound of $N, P, Q$ with

$$
N = \mathcal{O}(k\log\frac{1}{q} + k(1-q)\sigma_g^2)
$$

$$
P = \mathcal{O}(k^2 + (1-q)k\sigma_g^4 + \sigma_g^2 k\log\frac{1}{q})
$$

$$
Q = \mathcal{O}(k\log\frac{1}{q} + k(1-q)\sigma_g^2).
$$

With these, we can derive the big-$\mathcal{O}$ bound for $E_1$ as

$$
E_1 = \mathcal{O}(k^2(1-q)^2 + k(1-q)N + P + N^2 + Q^2) = \mathcal{O}(k^2\log^2\frac{1}{q} + k^2(1-q)^2\sigma_g^4).
$$

Therefore, we can bound the variance as

$$
\mathrm{Var}\left[\frac{\sum_{t=kq+1}^{k} p(\pi^t)V(\boldsymbol{\theta}_{\pi^t,j}^*)}{k-kq}\Bigg| \bar{\boldsymbol{\theta}}_{\pi^1,j},\ldots,\bar{\boldsymbol{\theta}}_{\pi^k,j}\right] \leq \frac{1}{(k-kq)^2}E_1 = \mathcal{O}\left(\frac{\log^2\frac{1}{q}}{(1-q)^2} + \sigma_g^4\right).
$$

As such, for the semivalue estimator, we have

$$\text{Var}[\psi_j^* | \boldsymbol{\theta}_{\pi^{kq+1},j}^p, \boldsymbol{\theta}_{\pi^2,j}^p, \ldots, \boldsymbol{\theta}_{\pi^k,j}^p] = \text{Var}\left[\frac{\sum_{t=kq+1}^{k} p(\pi^t)[V(\boldsymbol{\theta}_{\pi^t,j}^*) - V(\boldsymbol{\theta}_{\pi^t,j}^p)]}{k - kq} \middle| \boldsymbol{\theta}_{\pi^{kq+1},j}^p, \boldsymbol{\theta}_{\pi^2,j}^p, \ldots, \boldsymbol{\theta}_{\pi^k,j}^p\right]$$

$$= \text{Var}\left[\frac{\sum_{t=kq+1}^{k} p(\pi^t) V(\boldsymbol{\theta}_{\pi^t,j}^*)}{k - kq} \middle| \bar{\boldsymbol{\theta}}_{\pi^{kq+1},j}, \bar{\boldsymbol{\theta}}_{\pi^2,j}, \ldots, \bar{\boldsymbol{\theta}}_{\pi^k,j}\right]$$

$$= \mathcal{O}\left(\frac{\log^2 \frac{1}{q}}{(1-q)^2} + \sigma_g^4\right)$$

where the change in conditioning random variable in the 2nd step is because $\bar{\boldsymbol{\theta}}_{\pi^t,j}$ is deterministic conditional on $\boldsymbol{\theta}_{\pi^t,j}^p$, explained in more detail in the proofs for the above propositions. As all operations are applied to $\tilde{g}$, by the post-processing immunity of DP, the same $(\epsilon, \delta)$-DP guarantee holds. □

# D  EXPERIMENTS

**Hardware and software details.**    We perform all our experiments on Nvidia L40 GPUs using the PyTorch (Paszke et al., 2019) deep learning framework with the Opacus (Opacus) implementation of private random variables (PRV) (Gopi et al., 2021) for privacy accounting. Source codes are included in supplementary materials.

**Hyperparameter settings.**    For experiments using Algorithm 1, we follow (Ghorbani & Zou, 2019) and apply hyper-parameter search to find a suitable $\alpha$ which produces the best model performance with one pass of training examples. For a given number of evaluations $k$ and $\epsilon$, Opacus automatically adjusts the noise multiplier $\sigma$ to achieve the $(\epsilon, 5 \times 10^{-5})$-DP guarantee. Hence, we focus on the analysis of the interaction between DP requirements and data valuation by studying $\epsilon$ and $k$. In all experiments, we choose the value of $\epsilon$ such that we can observe a degradation of the performance of the estimate with i. i. d. noise as compared to the no-DP estimate given the limited budget. This approach of setting $\epsilon$ allows us to highlight the advantage of using correlated noise.

**Model specification.**    We specify the parameterization of the models used in our experiments. We adopt the following notations: ReLU denotes a rectified linear unit activation function; Linear$(x, y)$ denotes a linear layer (i.e. a matrix of dimension $x$ by $y$); Sigmoid denotes a Sigmoid activation function; Softmax denotes a Softmax activate function; Conv$(x, y, z)$ denotes a convolutional layer with input size $x$, output size $y$, and kernel size $z$ (with stride 1 and padding 0); Pool$(z, w)$ denotes a pooling layer with kernel size $z$ and stride $w$ (with padding 0). The neural network models used in our experiments are parameterized as follows:

$$\text{Logistic Regression}(\boldsymbol{x}) \coloneqq \text{Sigmoid} \circ \text{Linear}(\text{no. features}, \text{no. classes})(\boldsymbol{x}) ;$$

$$\text{CNN}(\boldsymbol{x}) \coloneqq \text{Softmax} \circ \text{Linear}(\cdot, \text{no. classes}) \circ \text{Pool}(2, 2) \circ \text{ReLU} \circ \text{Conv}(1, 16, 3)(\boldsymbol{x}) ;$$

ResNet18 and ResNet34 are standard. We follow the implementation of these two networks in PyTorch's torchvision library.

## D.1  MIA ATTACK

We follow the MIA attack proposed in (Wang et al., 2023), which constructs a likelihood ratio test based on the estimated data values. As a setup, we select 25 data points as the members and 25 data points as non-members. We further select 200 data points to subsample "shadow dataset". We choose $k = 200$ for breast cancer and diabetes datasets and $k = 500$ for Covertype dataset. As shown in Table 5, our method offers privacy protection against MIA by reducing the AUROC of the attack to around 0.5 (same level as random guess).

## D.2  GENERALIZING TO OTHER USE CASES

**Notes on dataset valuation.**    For dataset valuation on ResNet18 and ResNet34, we apply some engineering tricks to encourage faster convergence so that we obtain meaningful marginal contributions

Table 5: Mean (std. errors) of AUC on Covertype, breast cancer, and diabetes datasets trained with LR. $\epsilon = 1.0$.

|       | Covertype       | breast cancer   | diabetes        |
| ----- | --------------- | --------------- | --------------- |
| no DP | 0.554 (1.61e-02) | 0.583 (2.57e-02) | 0.567 (1.59e-02) |
| Ours  | 0.490 (3.76e-02) | 0.477 (2.62e-02) | 0.513 (3.68e-02) |

in each iteration using the G-Shapley framework. These include 1) freezing part of the network and only fine-tuning the last residual block and the fully connected layer; 2) reinitializing the model with the pre-trained weight after each iteration.

**Federated learning setup.** (Wang et al., 2020) proposed using an average of Shapley value in multiple rounds of FL as the data value metric

$$\psi_j := k^{-1} \sum_{t=1}^{k} \nu_j^t ,$$

where $\nu_j^t$ refers to the data Shapley of party $j$ at round $t$ using its gradients released at round $t$ (which we estimate with 100 samples of permutations in each participation round). Note that this formulation is analogous to the general semivalue definition of Eq. (1). As such, we may treat each $\nu_j^t$ as a marginal contribution of party $j$ which is averaged over all participation rounds. Note that in a total of $k$ rounds, each party still needs to release the gradients $k$ times, thus incurring the problem of linearly scaling variance of the Gaussian noise which can be alleviated with our method. However, the setting of FL is not identical to the conventional data valuation setting as the global model keeps updating which causes the gradients to change drastically, especially in the first few rounds. To address this challenge, we use a weighted sum that puts more emphasis on the more current gradients in the first few rounds: $X_{t,t} = 0.75 - 0.7 \times t/k$. Note that this choice is consistent with Prop. 5.3 which suggests setting a larger $X_{t,t}$ when $\sigma_g^2$ is large. This can be implemented algorithmically by updating $\tilde{g}_{\pi^t}$ with $\tilde{g}_{\pi^t}^* = (0.25 + 0.7 \times t/k)\tilde{g}_{\pi^{t-1}}^* + (0.75 - 0.7 \times t/k)\tilde{g}_{\pi^t}$. Moreover, as the global model converges with more collaboration rounds, more information about the data values is revealed in the first few rounds. Therefore, setting $q$ too large causes $\psi_j$ to miss important information about the data value, leading to inaccurate data value estimates, even though the estimation uncertainty is lowered. As such, we set a moderately small burn-in ratio $q = 0.2$.

### D.3 ADDITIONAL EXPERIMENTAL RESULTS FOR SEC. 6.1

**Additional results for $s^2$ and $\mu$.** We additionally plot a counterpart version of Fig. 2 with the $\psi$'s evaluated with no DP added in Fig. 5. It can be observed that the variance of $\psi$'s computed with no DP noise is almost the same as that computed with correlated noise. The means $\mu$'s are also similar.

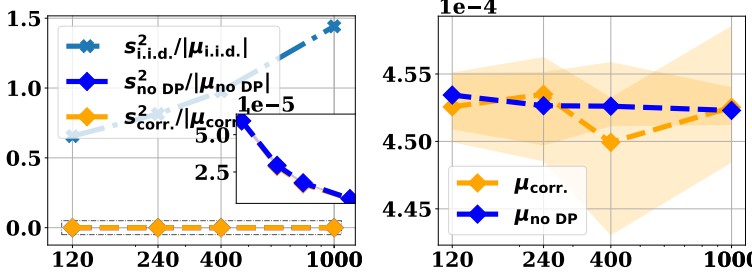

Figure 5: (Left) $n^{-1} \sum_{j \in [n]} s_j^2/|\mu_j|$ and (right) $\mu_j$ vs. $k$ using i.i.d. noise, correlated noise, and no DP noise.

**Additional results for data selections.** We supplement the data selection experiment with results for adding/removing data with high/low $\psi$'s shown in Fig. 6. For i.i.d. noise, as $k$ increases, the test accuracy approaches that of random selection, suggesting that the data value estimates are less reflective of the true worth of data. On the other hand, $\psi$'s computed with correlated noise exhibit a test accuracy curve close to $\psi$'s computed without DP, implying that the data value estimates computed with correlated noise are reflective of the true worth of data.

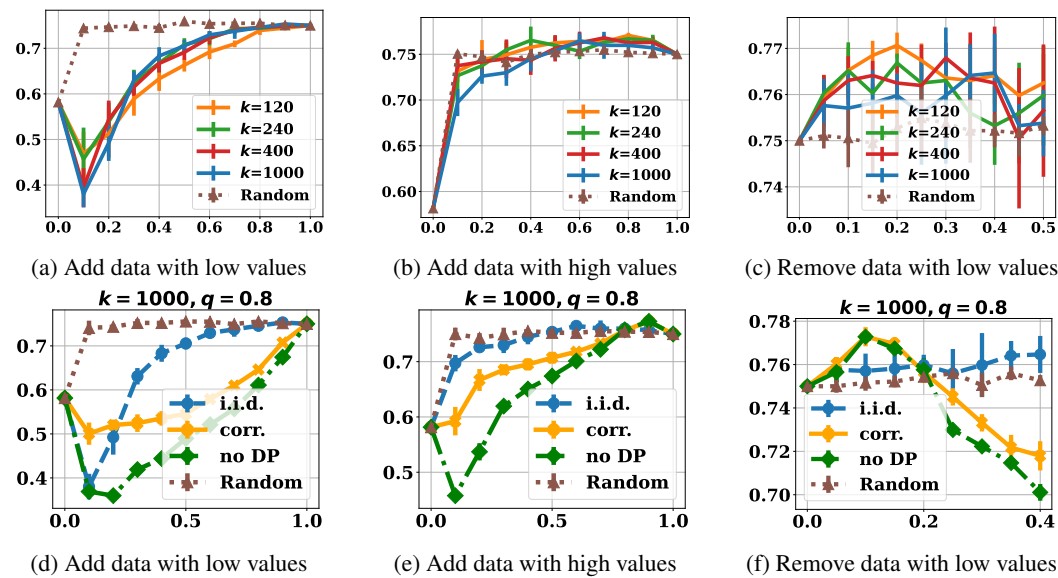

Figure 6: Data selection task where data with high/low $\psi$'s are added/removed from the training dataset with $\psi$'s computed using (top) i.i.d. noise with different $k$ and using (bottom) i.i.d. noise, correlated noise, and no DP noise with $k = 1000$. $y$-axis represents test accuracy and $x$-axis represents percentage of data added/removed.

**Results with regression tasks.** We notice that the noise due to DP has a much less significant impact on the data selection task performance for regression models. This may be attributable to the lack of a critical decision boundary which reduces the importance of individual data points. We conduct an experiment with the wine quality dataset (Cortez et al., 2009) where we randomly select 400 training examples and 1000 test examples trained with a linear regression using the average negated mean squared error as $V$. The results are shown in Fig. 7. It can be seen that the mean absolute errors are almost the same for different $k$ and with different methods.

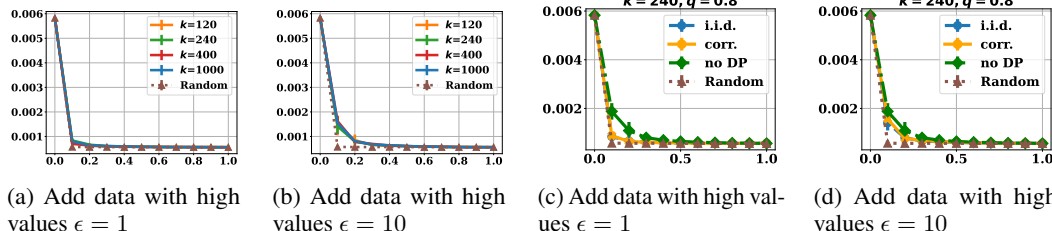

Figure 7: Data selection task where data with high/low $\psi$'s are added/removed from the training dataset with $\psi$'s computed using (a)(b) i.i.d. noise with different $k$ and (c)(d) i.i.d. noise, correlated noise, and no DP noise with $k = 240$. $y$-axis represents mean absolute error and $x$-axis represents percentage of data added/removed.

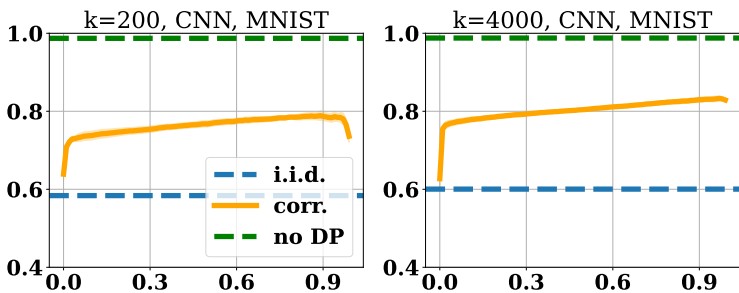

Figure 8: AUC v.s. $q \in [0, 1)$ using negated loss as $V$. Lines (shades) represent mean (std).

## D.4 ADDITIONAL EXPERIMENTAL RESULTS FOR SEC. 6.2

We include in Fig. 8 a counterpart of AUC plots for the noisy label detection task to that shown in Fig. 3. A similar trend can be observed as in the main text.

We also include the AUC plots for the noisy label detection task with other semivalues in Fig. 9. Similar trends as discussed in the main text can be observed in the figures. Moreover, we provide a result for Leave-one-out (LOO) (Cook, 1977) which is not a regular semivalue but also enjoys the improvement offered by our approach, as demonstrated in Table 6. However, the performance is worse than regular semivalues demonstrated in the main content in Table 2, which is expected since LOO does not take into account the marginal contribution over different subsets, as explained in (Ghorbani & Zou, 2019).

Table 6: Average (standard errors) of AUC on Covertype trained with logistic regression (top) and MNIST trained with CNN (bottom). The best score (except for "no DP" which is the baseline to be approximated) is highlighted. Higher is better.

| no DP | i.i.d. noise | $\boldsymbol{X}^*$ | $\boldsymbol{Y}^*$ ($q = 0.3$) | $\boldsymbol{Y}^*$ ($q = 0.5$) | $\boldsymbol{Y}^*$ ($q = 0.9$) |
|---|---|---|---|---|---|
| 0.832 (6.00-e03) | 0.511 (0.00e+00) | 0.659 (2.10e-02) | **0.720 (8.00e-03)** | 0.561 (8.00e-03) | 0.311 (3.00e-03) |
| 0.945 (3.00e-03) | 0.487 (2.00e-03) | 0.552 (2.00e-02) | **0.653 (1.50e-02)** | 0.528 (8.00e-03) | 0.305 (5.00e-03) |

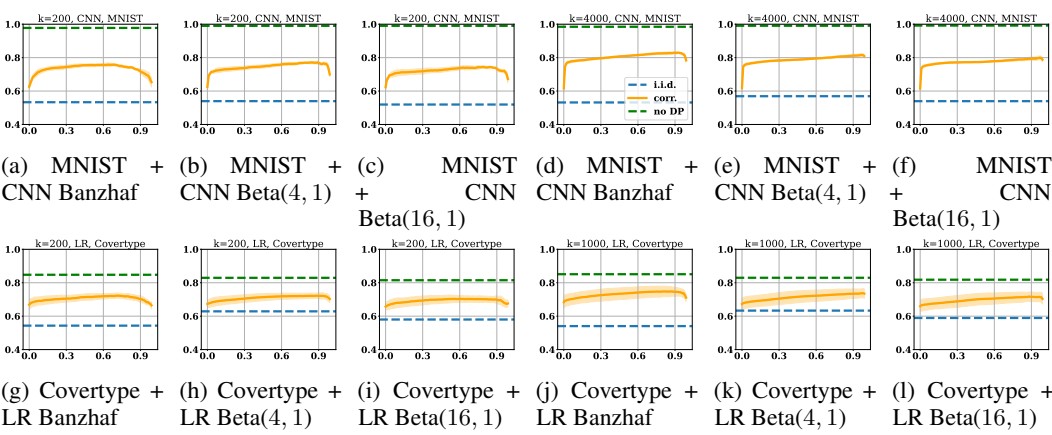

(a) MNIST + CNN Banzhaf (b) MNIST + CNN Beta$(4, 1)$ (c) MNIST + CNN Beta$(16, 1)$ (d) MNIST + CNN Banzhaf (e) MNIST + CNN Beta$(4, 1)$ (f) MNIST + CNN Beta$(16, 1)$

(g) Covertype + LR Banzhaf (h) Covertype + LR Beta$(4, 1)$ (i) Covertype + LR Beta$(16, 1)$ (j) Covertype + LR Banzhaf (k) Covertype + LR Beta$(4, 1)$ (l) Covertype + LR Beta$(16, 1)$

Figure 9: AUC plots for data Banzhaf, Beta$(4, 1)$ and Beta$(16, 1)$ with $k = 200$ and $k = 1000/4000$.

## D.5 ADDITIONAL EXPERIMENTAL RESULTS FOR DIFFERENT VALUES OF $\epsilon$.

We follow the discussion in App. B and show experimental results for the noisy label detection task with different $\epsilon$ values: 0.1 and 10 ($\epsilon = 1$ is provided in the main text). The results are shown in App. D.5. It can be observed that for $\epsilon = 10$, using i.i.d. noise can perform on par with using no

DP, e.g., on Covertype dataset with logistic regression using data Shapley and Beta$(4, 1)$. On the other hand, when $\epsilon = 0.1$, both i.i.d. noise and correlated noise have performance close to random selection on all ML tasks and semivalues.

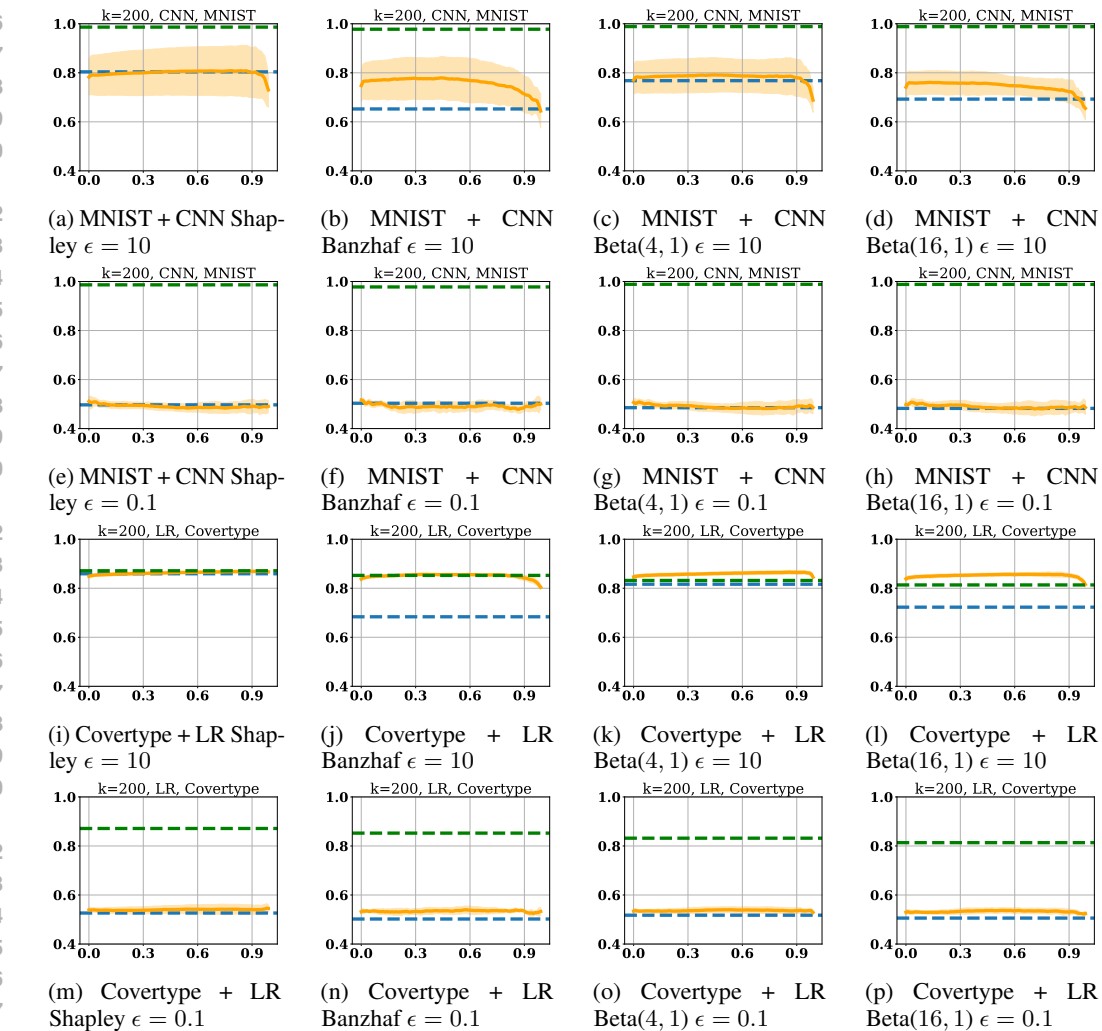

(a) MNIST + CNN Shapley $\epsilon = 10$

(b) MNIST + CNN Banzhaf $\epsilon = 10$

(c) MNIST + CNN Beta$(4, 1)$ $\epsilon = 10$

(d) MNIST + CNN Beta$(16, 1)$ $\epsilon = 10$

(e) MNIST + CNN Shapley $\epsilon = 0.1$

(f) MNIST + CNN Banzhaf $\epsilon = 0.1$

(g) MNIST + CNN Beta$(4, 1)$ $\epsilon = 0.1$

(h) MNIST + CNN Beta$(16, 1)$ $\epsilon = 0.1$

(i) Covertype + LR Shapley $\epsilon = 10$

(j) Covertype + LR Banzhaf $\epsilon = 10$

(k) Covertype + LR Beta$(4, 1)$ $\epsilon = 10$

(l) Covertype + LR Beta$(16, 1)$ $\epsilon = 10$

(m) Covertype + LR Shapley $\epsilon = 0.1$

(n) Covertype + LR Banzhaf $\epsilon = 0.1$

(o) Covertype + LR Beta$(4, 1)$ $\epsilon = 0.1$

(p) Covertype + LR Beta$(16, 1)$ $\epsilon = 0.1$

Figure 10: Noisy label detection task with $\epsilon = 10$ and $\epsilon = 0.1$.

## D.6 ADDITIONAL EXPERIMENTAL RESULTS ON RUNTIME AND MEMORY

We assess with experimental results the memory overhead of saving and computing correlated gradients $\tilde{g}^*_{\pi,i}$ for party $i$. Theoretically, the memory overhead is $\mathcal{O}(n)$ for a valuation with $n$ parties as each party needs to store its rolling gradient. The runtime overhead is $\mathcal{O}(1)$ as each party only needs to update the rolling gradient each time. The memory overhead results tabulated in Table 7 show a minor (GPU) memory overhead w.r.t. the overall memory usage and that the overhead is *linear* in $n$. The runtime overhead results tabulated in Table 8 demonstrate *no obvious difference* in the computational time with or without correlated noise.

Table 7: GPU peak memory usage with and without (shown in bracket) correlated noise in megabytes. Results are obtained on the diabetes and MNIST datasets with logistic regression (LR) and CNN w.r.t. various number of parties $n$.

| $n$ | diabetes+LR | MNIST+CNN |
|-----|-------------|-----------|
| 100 | 18.231 (18.129) | 120.788 (109.729) |
| 200 | 18.329 (18.124) | 131.847 (109.729) |
| 300 | 18.427 (18.120) | 142.906 (109.729) |

Table 8: Program runtime with and without (shown in bracket) correlated noise in seconds. Results are obtained on the diabetes (top) and MNIST (bottom) datasets with logistic regression (top) and CNN (bottom) w.r.t. various number of parties $n$ and evaluation budget $k$.

| $k$ / $n$ | 120 | 240 | 360 |
|-----------|-----|-----|-----|
| 100 | 78.466 (78.896) | 154.954 (159.989) | 262.166 (257.588) |
| 200 | 113.612 (125.255) | 256.431 (252.465) | 407.126 (408.107) |
| 300 | 158.565 (154.425) | 344.286 (365.348) | 530.579 (519.320) |

| $k$ / $n$ | 120 | 240 | 360 |
|-----------|-----|-----|-----|
| 100 | 99.992 (100.140) | 199.471 (195.137) | 328.204 (323.646) |
| 200 | 158.511 (151.589) | 347.525 (329.267) | 527.296 (531.594) |
| 300 | 223.393 (213.637) | 456.763 (467.780) | 703.843 (709.514) |

