# OpenReview forum: "Data Value Estimation on Private Gradients"
_ICLR.cc/2025/Conference — Submitted to ICLR 2025_

### Official Review · Reviewer_hHZY · 2024-10-24

**Soundness:** 3
**Presentation:** 3
**Contribution:** 3
**Rating:** 6
**Confidence:** 4

**Summary:**

This work proposes a novel method for estimation of data value under differentially private training based on the notion of estimation uncertainty which is introduced by the addition of DP noise and propose a new way to mitigate this by injecting correlated noise instead.

**Strengths:**

The work is well-motivated and addresses an important problem (the privacy-utility trade-off, where utility is a value function over the dataset the model could have access to). The structure is clear, the statements are supported by both the empirical and the theoretical results. I also appreciate the openness about some of the limitations i.e. that the fundamental scientific problem of privacy-utility with respect to the theoretical information loss is still unsolved, but this work proposes a method which can alleviate it to some extent by maintaining the ranking of the values.

While a lot of what is discussed in this paper is well-known, this is a very nice attempt to formalise these notions through uncertainty.

**Weaknesses:**

While the results are certainly interesting, they seem to rely on a number of assumptions and limitations. The empirical results only feature a handful of datapoints on 2 relatively simple benchmark datasets (ML evaluation), V being deterministic and model parameters fixed etc. I would like to see more a) utility functions covered, b) larger dataset sizes and c) range of epsilon values. The last one being particularly important because (as discussed in the works linked below), DP diminishes value not just because of noise, but clipping. And it is important to consider the trade-off between noise and clipping (and not just correlated vs independent noise) in more detail when it comes to methods for valuation. Some of the models used in evaluation (the resnet family) is additionally well-protected against the effects of gradient clipping, meaning that this effect is not observed in full detail in the evaluations. You may consider adding another baseline model to discuss the comparative utility loss of clipping vs noise.

The presentation of the figures and algorithms is rather confusing. I found most of the figures (e.g. 1 and 2) to be unreadable even on larger screens and algorithm 1 is difficult for me to digest: are all 3 colors integrated into it already or are they mutually exclusive drop-ins which we need to evaluate separately? Figure 2 in particular needs a rework, there is too much happening at the same time with too little explanation.

I personally think its a bit of an odd choice to move some of the most important results (i.e. how the method 'actually' protects privacy under MIAs) to the appendix. Moreover, there seems to be no comparison to traditional DP and the only one you are comparing against is the uncalibrated non-private method which already has pretty poor performance to begin with. I am not convinced how 'private' this really is given that your budget is 1.0 and with a budget of infinity the attack is better by 5% on average and 10% at most (which could have been a convincing results, but I am yet to see how different budgets or a standard DP mechanism e.g. DP-SGD would perform in comparison).

There are some recent works which show that there are valuation methods which are usable even under DP (and in particular for FL use cases), which are not mentioned in this work [1,2]. Additionally there are also works which discuss the use of privatised statistics under DP training in order to obtain better privacy-utility trade-offs [3,4]. While these rely on different values (and different issues of DP when it comes to utility loss), these should also be part of the prior works (and ideally a set of baselines). [4] actually does something really similar to this work, but under DP-FTRL rather than DP-SGD.

Currently there are a lot of limitations which I would like the authors to address before I can recommend acceptance, but should the responses be satisfactory, I am open to raising my score.

[1] - Sim, Rachael, et al. "Incentives in private collaborative machine learning." Advances in Neural Information Processing Systems 36 (2024).

[2] - Usynin, Dmitrii, Daniel Rueckert, and Georgios Kaissis. "Incentivising the federation: gradient-based metrics for data selection and valuation in private decentralised training." European Interdisciplinary Cybersecurity Conference. 2024.

[3] - Bani-Harouni, David, et al. "Gradient Self-alignment in Private Deep Learning." International Conference on Medical Image Computing and Computer-Assisted Intervention. Cham: Springer Nature Switzerland, 2023.

[4] - Koloskova, Anastasiia, et al. "Gradient descent with linearly correlated noise: Theory and applications to differential privacy." Advances in Neural Information Processing Systems 36 (2023).

**Questions:**

Minor:

Which neighbouring definition is used for DP here? Add/remove or replace one?

Please elaborate on algorithm 1, it is not clear how to interpret the different options.

---

> ### Author Response · Authors · 2024-11-20
>
> ## Response Part (1/2)
>
> Dear reviewer hHZY,
>
> We sincerely thank you for taking time to provide your review and for acknowledging the motivation of our work and its contribution to formalizing differentially private data valuation through uncertainty.
>
> We would like to address your concerns in the following.
>
> ---
>
> **1. Empirical Results**
>
> > The empirical results only feature a handful of datapoints on 2 relatively simple benchmark datasets (ML evaluation), V being deterministic and model parameters fixed etc. I would like to see more a) utility functions covered, b) larger dataset sizes and c) range of epsilon values.
>
> Thank you for your suggestions. We wish to clarify that our experimental results include several datasets, including Covertype, Brease cancer, Diabetes, CIFAR-10, and MNIST. These datasets are commonly used in related works, e.g. [1,2,3,4] in your reference. We evaluated both continuous utility functions (negative loss) and discrete ones (accuracy) across various models, including logistic regression, CNNs, and ResNet18, as detailed in Figure 3. For models, we consider a variety of model families including logistic regression, CNN, and ResNet18, which are also considered in [1,2,3,4].
>
> Regarding a broader range of $\epsilon$ values, we included additional results in Appendix D.5, showing that for $\epsilon \geq 1$, our method achieves significant improvements in data value estimates, while for $\epsilon=0.1$, performance deteriorates as expected in the high-privacy regime.
>
> Following your suggestion, we have conducted an additional experiment using a multi-layered perceptron (MLP), which we present in the following subsection.
>
> ---
>
> **2. Comparison with different Clipping Norms**
>
> > You may consider adding another baseline model to discuss the comparative utility loss of clipping vs noise.
>
> Thank you for suggesting the additional experimental setting. We have conducted an experiment using a custom 2-layered MLP with ReLU activation to test the performance as we gradually increase the clipping norm $C$ while fixing $\epsilon=1$. For other parameters, we use $k=400$, $n=100$, $q=0.5$.
>
> | C | Method | Covertype | Breast Cancer | Diabetes |
> |---|--------|-----------|---------------|----------|
> | 0.1 | DP-SGD | 0.545 (6.2e-02) | 0.640 (3.9e-02) | 0.572 (7.8e-02) |
> | 1.0 | DP-SGD | 0.530 (6.2e-02) | 0.607 (3.5e-02) | 0.541 (7.5e-02) |
> | 5.0 | DP-SGD | 0.529 (6.2e-02) | 0.606 (3.6e-02) | 0.541 (7.3e-02) |
> ||
> | 0.1 | Ours | 0.632 (4.1e-02) | 0.624 (2.2e-02) | 0.651 (3.4e-02) |
> | 1.0 | Ours | 0.588 (3.4e-02) | 0.608 (3.0e-02) | 0.557 (2.2e-02) |
> | 5.0 | Ours | 0.576 (3.4e-02) | 0.596 (3.7e-02) | 0.569 (2.6e-02) |
>
>
> From the experimental results, we observe that for both naive DP-SGD and our method, the data value estimates become less accurate as $C$ increases. This is expected as greater clipping norm results in greater Gaussian noise per-step. Though, for $C > 1$, the performance drop is relatively small.
>
>
>
> ---
>
> **3. Comparison with Baseline DP Method in MIA**
>
> > Moreover, there seems to be no comparison to traditional DP and the only one you are comparing against is the uncalibrated non-private method which already has pretty poor performance to begin with.
>
> Thank you for pointing out the comparison with traditional DP (DP-SGD). To clarify, the i.i.d. method in our experiments refers to data valuation using the plain DP-SGD algorithm. For the MIA experiment, we provide the following comparative results:
>
> |Method| Covertype | Breast cancer | Diabetes |
> |------|-----------|---------------|----------|
> |no DP| 0.554 (1.61e-02)| 0.583 (2.57e-02) | 0.567 (1.59e-02) |
> | Ours | 0.490 (3.7e-02) | 0.477 (2.62e-02) | 0.513 (3.68e-02) |
> |DP-SGD| 0.500 (0.00) | 0.499 (3.03e-02) | 0.486 (1.31e-02) |
>
> It can be observed that traditional DP-SGD achieves  reasonable MIA protection, as expected. However, the key challenge in using DP-SGD for data valuation lies in controlling the scaling of estimation uncertainty, which our work specifically addresses. Due to space constraints, we focused on data value estimation as the primary contribution of our work.
>
> ---
>
> **4. Presentation of Algorithm 1**
>
> > algorithm 1 is difficult for me to digest: are all 3 colors integrated into it already or are they mutually exclusive drop-ins which we need to evaluate separately?
>
> Thank you for raising this conrern. We apologize for any confusion. All 3 colors are integrated, as explained in lines 353-354. To improve clarity, we have labeled each section with explicit "if-else" clauses in the revised version.

---

> > ### Author Response · Authors · 2024-11-20
> >
> > ## Response Part (2/2)
> >
> > **5. Presentation of Figures**
> >
> > > I found most of the figures (e.g. 1 and 2) to be unreadable even on larger screens...Figure 2 in particular needs a rework, there is too much happening at the same time with too little explanation.
> >
> > Thank you for raising the concern. We apologize for the inconvenience caused by unclear figures. In the revised version, we have enhanced all figure labels and captions to provide better context and improve readability.
> >
> >
> > ---
> >
> > **6. Recent Related Works**
> >
> > > There are some recent works which show that there are valuation methods which are usable even under DP (and in particular for FL use cases), which are not mentioned in this work [1,2]. Additionally there are also works which discuss the use of privatised statistics under DP training in order to obtain better privacy-utility trade-offs [3,4].
> >
> > Thank you for pointing us to these recent works. We already include [1] in our related work section and have added discussions on [2,3,4] in the revised version. Specifically:
> >
> > - [2] proposes using VoG and PLSS for DP training. However, as noted by the authors, these metrics are not privatized, introducing privacy leaks during selection.
> > - [3] introduces cosine similarity for gradient selection, primarily to improve DP-SGD performance. While this improves training, it does not ensure accurate data value estimation, as shown in our new experiment below.
> > - [4] analyzes correlated noise in a Follow-The-Regularized-Leader (FTRL) framework, distinct from our data valuation focus.
> >
> > In particular, [3] introduced an intersting technique of using cosine similarity to decide whether to include a gradient for training. While their goal was primarily improving the performance of DP-SGD, we provide an experiment analyzing the effectiveness of their method in noisy label detection. For a fair comparison, we use a batch size of $1$, the same as our setting. We use un-privatized cosine similarity with threshold $\psi=0$. For other parameters, we choose $k=200$, $n=100$, $C=1$ and $\epsilon=1$. The results are tabulated below.
> >
> > | Method | Dataset | q=0 | q=0.5 |
> > |--------|---------|-----|-------|
> > | DP-SGD w. CS | Covertype | 0.459 (4.5e-02) | NA |
> > | DP-SGD w. CS | Breast Cancer | 0.366 (4.2e-02) | NA |
> > | DP-SGD w. CS | Diabetes | 0.479 (5.0e-02) | NA |
> > ||
> > | Ours w. CS | Covertype | 0.489 (2.7e-02) | 0.465 (2.7e-02) |
> > | Ours w. CS | Breast Cancer | 0.695 (5.5e-02) | 0.725 (4.2e-02) |
> > | Ours w. CS | Diabetes | 0.556 (5.0e-02) | 0.535 (3.5e-02) |
> > ||
> > | Ours w/o CS | Covertype | 0.559 (7.7e-02) | 0.581 (6.5e-02) |
> > | Ours w/o CS | Breast Cancer | 0.694 (6.4e-02) | 0.757 (3.1e-02) |
> > | Ours w/o CS | Diabetes | 0.622 (5.8e-02) | 0.634 (3.7e-02) |
> >
> > It can be observed that including CS check in naive DP-SGD does not lead to useful data value estimates. Incorporating CS in our method does not improve the data value estimates either. We believe this is expected: While filtering out gradients may improve model training, it results in the true value of data points not being accurated reflected, which is an aspect not considered in [3].
> >
> > ---
> >
> > **7. DP Definition**
> >
> > > Which neighbouring definition is used for DP here? Add/remove or replace one?
> >
> > We follow the add/remove definition from Adadi et. al. (2016). Two datasets are considered neighbouring if an entry is present in one set but absent in the other.
> >
> > ---
> >
> > We hope our response has improved your opinion of our work. We are happy to clarify any further questions that you might have.

---

> > > ### Comment · Reviewer_hHZY · 2024-11-20
> > > **Response to the rebuttal**
> > >
> > > I would like to thank the authors for their comprehensive response!
> > >
> > > It has addressed the majority of my concerns. Some additional questions/comments I still have:
> > >
> > > 1 - [2] actually does propose the release of these metrics in DP form, so this does make it a comparable method (albeit with a slightly different end purpose, so this might not be a 'baseline' as such, but a prior work nonetheless).
> > >
> > > 2 - MIA 'should be' at worst a random guess (0.5), your results fluctuate below this threshold, could you elaborate on how/why and if this is a numerical phenomenon or if you believe your approach 'tricks' the attacker into accepting false information when performing the attack?
> > >
> > > Other than that I am largely happy with the response

---

> ### Author Response · Authors · 2024-11-21
>
> Thank you for your response and for raising additional questions. We are delighted to hear that our previous response was largely satisfactory.
>
> We address your additional concerns as follows.
>
> ---
>
> **1. DP Algorithm of [2]**
>
> Thank you for pointing this out. To clarify, we observed that while the authors mention the DP version of VoG and PLSS in Section 4.7 of [2], they do not provide experimental results to demonstrate the effectiveness of these proposed DP version. We acknowledge that our original phrasing was imprecise, and we have updated the revision to better reflect this observation.
>
> ---
>
> **2. AUC Results for MIA**
>
> Thank you for making this interesting observation. The results sometimes fall below $0.5$ due to numerical issues. Specifically, the ROC AUC is computed using a trapezium summation method over the TPR against FPR as the threshold changes. In cases where attack scores are closely clustered, small changes in the threshold can result in significant differences in TPR and FPR. This can cause the computed AUC to be smaller than the true AUC, as the trapezium summation aggregates a discrete number of intervals. A similar phenomenon can be found in Table 3 of [5] where some reported AUROC values are below $0.5$.
>
>
> [5] Wang et. al., Threshold KNN-Shapley: A Linear-Time and Privacy-Friendly Approach to Data Valuation, NeurIPS 2023.
>
> ---
>
> We hope the above has addressed your concern and improved your opinion of our work. We are happy to discuss with you for any further questions.

---

> > ### Comment · Reviewer_hHZY · 2024-11-21
> > **Response to the comment**
> >
> > I would like to thank the authors for their prompt response. I have raised the score slightly since my concerns have been addressed.

---

> > > ### Author Response · Authors · 2024-11-22
> > >
> > > Thank you for your increase in score. We are pleased that our responses have adequately addressed your concerns.

---

### Official Review · Reviewer_nG37 · 2024-11-03

**Soundness:** 4
**Presentation:** 4
**Contribution:** 3
**Rating:** 6
**Confidence:** 4

**Summary:**

This paper addresses the data value estimation problem using privately released gradients. To
compare the performance of various data value estimators, the authors introduce the concept of
estimation uncertainty and demonstrate that it grows linearly with the number of evaluations in
the naive method that injects independent Gaussian noise. To reduce the uncertainty, the authors
devise an algorithm that injects correlated noise into the gradients and demonstrates that it eliminates
the estimation uncertainty’s linear dependence on the number of evaluations.

**Strengths:**

- It considers an interesting problem of data value attribution that has wide-ranging applications
in many real-world scenarios.
- The authors introduce the concept of estimation uncertainty and, using this metric, devise two
algorithms for data value estimation.
- The theoretical analysis of the estimation uncertainty of proposed algorithms shows that their
method provably reduces the dependence on the number of evaluations from linear to log-squared
k, where k is the number of evaluations.
- The authors further reduce the dependence of the algorithm’s estimation uncertainty on the number of evaluations to a constant.

**Weaknesses:**

- The proposed method is somewhat simple, as it essentially computes a simple weighted average
of previously released private gradients. While the simplicity of the proposed method does not
imply a lack of novelty, the existence of prior work that also carefully generates the correlated
noise to reduce the variance of released statistics suggests that there might be room for further
improvement in the proposed approach.
- In essence, the introduced estimation uncertainty is the variance of released gradients. The
variance of released statistics has long been used as a utility metric in differential privacy
literature. However, an important distinction is that most works employing the variance as
a utility metric consider unbiased statistics. In the paper, the private gradients are biased, and
I am curious to know how the proposed approach affects the bias of these gradients. Especially,
as the proposed algorithm takes the average of gradients evaluated in the preceding locations
in the optimization trajectory, the proposed approach may increase the bias. It
will be interesting if the authors can show that this is not the case.
- The graphs presented in the paper are not properly labeled. While the captions provide some
explanation, it’s still not easy to read and interpret the graphs. The figures should be self-contained and interpretable on its own.

**Questions:**

- How is the proposed solution $X^∗$ derived? Is it a solution (or an approximation solution) of
constrained optimization problem that minimizes the estimation uncertainty?
- In Figure 3, the performance of proposed method with burn-in seems to drastically decrease
when q is set too high. Is there a rule of thumb for setting this hyperparameter?

---

> ### Author Response · Authors · 2024-11-20
>
> Dear reviewer nG37,
>
> We sincerely thank you for taking your time to provide the review and for recognizing the interesting problem addressed in our work, as well as our theoretical contributions.
>
> We would like to address your concerns in the following.
>
> ---
>
> **1. Prior Work with Correlated Noise**
>
> > the existence of prior work that also carefully generates the correlated noise to reduce the variance of released statistics suggests that there might be room for further improvement in the proposed approach.
>
> Thank you for highlighting this point. We acknowledge that several prior works have explored adding correlated noise in the context of counting problems or online learning, as noted in our related work section. A key contribution of our work lies in the theoretical characterization of the use of correlated noise specifically within the data valuation setting under DP-SGD, a problem not directly addressed by prior research.
>
> We agree with you that there can be potential further improvement in our algorithm and we hope to further explore it in our future research.
>
> ---
>
> **2. Bias of the Estimator**
>
> > I am curious to know how the proposed approach affects the bias of these gradients. Especially, as the proposed algorithm takes the average of gradients evaluated in the preceding locations in the optimization trajectory, the proposed approach may increase the bias. It will be interesting if the authors can show that this is not the case.
>
> Thank you for raising this interesting question on the bias of our estimator! Indeed, our estimator is biased. We wish to also highlight that the naive DP-SGD estimator is also biased, since utility $V$ is often not a linear function of $\tilde{g}$.
>
> To provide a detailed comparison, we have improved our theoretical analysis and included a bias bound for our method in Propositions 5.2 and 5.3, highlighted in red. For Lipschitz-smooth $V$, we can bound the bias of the naive DP-SGD by $O(\log k)$. For our method, we have improved our theoretical analysis by incorporating a result on the bound of bias in Proposition 5.2 and Proposition 5.3, in red. We show that the bias of our method is $O(\log k + \sigma_g^2)$ without burn-in and $O((1-q)\log (1/q) + \sigma_g^2)$ with burn-in.
>
> While our bias has an additional $\sigma_g^2$ term than the naive DP-SGD, the term is a constant and does not grow with $k$. As such, controlling the linear scaling of estimation uncertainty is more crucial for improving the quality of estimates.
>
> ---
>
> **3. Presentation of Figures**
>
> > The graphs presented in the paper are not properly labeled. While the captions provide some explanation, it's still not easy to read and interpret the graphs. The figures should be self-contained and interpretable on its own.
>
> Thank you for pointing this out. We apologize the inconvenience in reading the figures. We have added the labels to all axes and improved the captions of all figures in the revised version. This should make the figures self-contained and more interpretable.
>
> ---
>
> **4. Derivation of $X^*$**
>
> > How is the proposed solution $X^*$ derived? Is it a solution (or an approximation solution) of constrained optimization problem that minimizes the estimation uncertainty?
>
> Thank you for your question. $X^*$ is an approximate solution of Proposition 5.2. In the formal version of Proposition 5.2, (Proposition C.10), an exact form of $X$ is provided. As $k \to \infty$, the exact solution tends to $X^*$.
>
> ---
>
> **5. Choice of Hyperparameter $q$**
>
> > In Figure 3, the performance of proposed method with burn-in seems to drastically decrease when q is set too high. Is there a rule of thumb for setting this hyperparameter?
>
> Thank you for your question. The performance drastically drops when $q$ is too high because insufficient samples remain (after burn-in) for an accurate estimation. Drawing on standard practices in burn-in techniques (e.g., in MCMC), a common practice is to discard the first 20%-50% of the samples, i.e, choosing $q \in [0.2,0.5]$.
>
> ---
>
> We hope our response has been adequate in addressing your concerns. We are happy to further discuss with you for any other questions that you might have.

---

### Official Review · Reviewer_eWxj · 2024-11-04

**Soundness:** 3
**Presentation:** 3
**Contribution:** 2
**Rating:** 6
**Confidence:** 3

**Summary:**

This paper addresses the differential privacy (DP) data valuation problem with perturbed gradients. The authors demonstrate that when i.i.d. noise is added to gradients, the estimation uncertainty does not decrease as the number of draws $k$ increases, because the variance of both the information and noise parts grows linearly with $k$. Assuming the data distribution is i.i.d., they propose an adaptive noise-adding mechanism with $O(\log k)$ variance, which bypasses the linear scaling of variance due to the added DP noise.

**Strengths:**

1. Both the motivation and problem setting are stated clearly, and the proposed adaptive mechanism provably beats the i.i.d. mechanism

2. comprehensive experiments are conducted to illstruate the proposed theory.

**Weaknesses:**

1. The statement of assumptions is unclear. In Proposition 5.2, the authors state that the isotropic sub-gaussian assumption is made for the distribution but then introduce the covariance matrix $\Sigma$. Does isotropic mean that the covariance is the identity matrix?(e.g. Definition 3.2.1 in https://www.math.uci.edu/~rvershyn/papers/HDP-book/HDP-book.pdf )

2. While I don't think the isotropic assumption is restrictive, I wonder whether the previously proposed binary counting mechanism( https://dl.acm.org/doi/10.1145/2043621.2043626 ) can also achieve the same performance bound as the proposed adaptive mechanism under this assumption, since it seems that the main task in this setting is to continually release a sequence of averaged gradient to reduce the variance.

**Questions:**

My main concern is whether the previously proposed method can achieve similar performance to those proposed in this paper, as mentioned in point 2 of the weaknesses. I am willing to raise my score if the authors can provide more explanation on this point.

---

> ### Author Response · Authors · 2024-11-20
>
> Dear reviewer eWxj,
>
> We sincerely thank you for taking time to provide your review and for recognizing the clarity of our motivation and problem setting, as well as the comprehensiveness of our experiments.
>
> We would like to address your concerns in the following.
>
> ---
>
> **1. Clarification on the Sub-Gaussian Assumption**
>
> > The statement of assumptions is unclear. In Proposition 5.2, the authors state that the isotropic sub-gaussian assumption is made for the distribution but then introduce the covariance matrix $\Sigma$. Does isotropic mean that the covariance is the identity matrix?
>
> Thank you for raising this point. We apologize for the confusion. In our revised version, We have corrected the name from "isotropic" to "diagonal". Our assumption only requires the gradients to be independent, which implies that all off-diagonal entries in the covariance matrix are
> $0$. We appreciate your bringing this to our attention and hope the revised terminology resolves any confusion.
>
> ---
>
> **2. Comparison with the Binary Counting Mechanism**
>
> > I wonder whether the previously proposed binary counting mechanism can also achieve the same performance bound as the proposed adaptive mechanism under this assumption, since it seems that the main task in this setting is to continually release a sequence of averaged gradient to reduce the variance.
>
> Thank you for raising this interesting question! While the binary counting mechanism shares some similarities with our problem setting as it addresses a continuous release problem, there are key distinctions that differentiate it from our approach:
>
> - Trusted central server: The binary counting mechanism requires a trusted central server since raw gradients must be shared internally before being privatized. In contrast, our method does not require a trusted central server, as gradients are privatized before being released.
>
> - Integration with DP-SGD: The binary counting mechanism aggregates raw data, which inherently differs from the DP-SGD framework. DP-SGD does not involve aggregating raw gradients in the same manner, making it not naturally compatible with the binary counting mechanism.
>
> However, assuming the presence of a trusted central server and that the model is updated using the aggregated gradient in one batch, we believe that the binary counting mechanism can achieve sub-linear estimated uncertainty as well. Here is a rough sketch of the proof: For each set $S$ and an individual data point $x$, to find the marginal contribution $V(S \cup \{x\}) - V(S)$, we can build a p-sum tree and use that to find the noisy aggregate
>
> $\tilde{g}_{S \cup \{x\}}$
>
> and $\tilde{g}_{S}$, each of them having a variance of size $O(\log{k})$ instead of $O(k)$ with naive DP-SGD. Since any polynomial of logarithmic of $k$ grows slower than linear $k$, for most utility functions $V$ including logistic regression and linear regression, we have $\text{Var}(V) = o(k)$. One can then derive that the estimation uncertainty is sublinear.
>
> ---
>
> We hope our response has adequately addressed your concerns. We are delighted to discuss with you any further questions that you may have.

---

> > ### Author Response · Authors · 2024-12-01
> >
> > Dear reviewer eWxj,
> >
> > We wish to kindly remind you that the deadline for the author-reviewer discussion period is approaching. Thank you for your thoughtful comments on our paper. We’ve addressed the points raised in our response and would greatly appreciate it if you could review the response at your convenience. Please let us know if there’s anything further we can clarify.

---

### Official Review · Reviewer_wKAq · 2024-11-04

**Soundness:** 2
**Presentation:** 2
**Contribution:** 2
**Rating:** 3
**Confidence:** 3

**Summary:**

This paper considers the question of data valuation under differential privacy. For each data point, we aim to produce a number describing the "value" of that example. This could be used later, for example in an auction. The paper focuses on adapting existing nonprivate techniques for data value estimation based on stochastic gradient descent to the private gradient descent setting.

I will note that this is a rather unusual goal within the study of differentially private algorithms, as we aim to infer details about each example, rather than statistics about the underlying dataset or distribution. At least one of the prior works cited, Wang et al. 2023, operates in a different privacy model: the $i$-th value estimates is released only to the $i$-th individual.

**Strengths:**

This is an interesting topic, clearly connected to prior work. There is a lot of interest in estimating data values.

The techniques here are non-trivial. It seems plausible to me that the experiments demonstrate a privacy-utility tradeoff that is acceptable for some uses.

**Weaknesses:**

I see three main issues with the submission. I look forward to discussion with the authors and other reviewers.

**First,** I did not understand why this notion of estimation uncertainty (Eq 3) is meaningful. An algorithm that always returns $\psi_j=0$ is perfectly private and has no variance. So I don't know how to interpret Proposition 5.3, which says that we can bound the variance by a constant. Is that good? Perhaps the correlated noise technique is simply bringing us closer to the $\psi_j=0$ example?

**Second,** I feel the paper is missing a basic discussion of what is possible here. Consider a setting where each example is blatantly either "valuable" or "not valuable." This is reasonable: maybe some examples are random noise. Here, the best private estimate of the value us randomized response, where each individual returns their true value with probability $e^\epsilon/(e^\epsilon+1)$. If, analogous to the experiment in Figure 3, 30% of examples are not valuable, then by Bayes theorem with $\epsilon=1$ we expect roughly 14% of examples that returned "valuable" to not be so.

Some passages may give the reader the wrong idea. Both the Remark on p4 and Appendix B.4 suggest one can accurately preserve the ranks under differential privacy (so that valuable data receives higher estimates). But, as the above example shows, privacy also constrains our ability to do this.

**Third,** Propositions 5.2 and 5.3 assume the following: for a given point $j$, its $k$ gradient estimates are iid and sub-Gaussian about a fixed mean. I'm confused about the assumption on a technical level (the distribution is isotropic, so isn't $\Sigma$ the identity?), but more importantly I don't understand how we could expect something like this to hold, even approximately.

I look forward to discussing this assumption with the authors and other reviewers, but even if it is reasonable I disagree with the informal discussion around it. For example, the abstract says you "provably remove the linear scaling of estimation uncertainty," but what is actually proven seems much weaker.

**Questions:**

Are there any settings where the iid sub-Gaussian gradient assumption is satisfied?

---

> ### Author Response · Authors · 2024-11-20
>
> ## Response Part (1/2)
>
> Dear reviewer wKAq,
>
> We sincerely thank you for taking time to provide the review and for recognizing the interesting aspects of our work, its clear connection to prior research, and that our non-trivial techniques present useful applications.
>
> We would like to address your concerns in the following.
>
> ---
>
> **1. Interpretation of Estimation Uncertainty (Eq. 3) and Proposition 5.3**
>
> > I did not understand why this notion of estimation uncertainty (Eq 3) is meaningful. An algorithm that always returns $\psi_j=0$ is perfectly private and has no variance. So I don't know how to interpret Proposition 5.3, which says that we can bound the variance by a constant. Is that good? Perhaps the correlated noise technique is simply bringing us closer to the $\psi_j=0$ example?
>
> Thank you for raising this insightful question. While setting $\psi_j=0$ always results in zero variance, it has substantial bias. To complement our focus on minimizing estimation uncertainty, we have enhanced our theoretical analysis to include bounds on the bias of our method (see our revised Propositions 5.2 and 5.3, highlighted in red). Specifically, with burn-in, our method achieves an estimation bias of $E[\psi_j^* - \psi_j] = O((1-q)\log (1/q) + \sigma_g^2)$. Note that this bound does not grow with $k$.
>
> As a side note, the naive DP-SGD method also introduces bias, as most utility functions $V$ are not linear in $\tilde{g}$. With Lipschitz-smooth $V$, the naive DP-SGD algorithm achieves a bias bound $E[\psi_j^* - \psi_j] = O(\log k)$.
>
> The core problem we address in our work is that even when $\sigma_g$ is moderate, current naive estimation method fails to produce accurate estimates because of linear scaling in estimation uncertainty. Since the estimation bias is already bounded by a constant, our work naturally focuses on minimizing estimation uncertainty.
>
> ---
>
> **2. Rank Preservation Under Differential Privacy**
>
> > Data value estimation of under a simple binary model...Both the Remark on p4 and Appendix B.4 suggest one can accurately preserve the ranks under differential privacy (so that valuable data receives higher estimates). But, as the above example shows, privacy also constrains our ability to do this.
>
> We appreciate your comment. We wish to clarify that the claim that our method "accurately preserve" the ranks is a comparative statement relative to the naive MC approach. As demonstrated in our experiments and supported by our constant-bound estimation uncertainty, our method achieves significantly better rank preservation. We do not claim "perfect" rank preservation and we agree with you that it remains a challenge due to the inherent constraint of DP.
>
> To address your feedback, we have revised the relevant passages (see lines 214-216) to explicitly clarify the limit of our claim and avoid potential misunderstandings.
>
> ---
>
> **3. Assumptions on Sub-Gaussian Distribution**
>
> > Propositions 5.2 and 5.3 assume the following: for a given point
> , its gradient estimates are iid and sub-Gaussian about a fixed mean...the distribution is isotropic, so isn't $\Sigma$ the identity?
>
> Thank you for pointing this out. To clarify, we have revised the text to explicitly state that $\Sigma$ is "diagonal," not the identity matrix. The diagonal sub-Gaussian assumption involes two components: (1) the gradients are independent, and (2) each gradient follows a sub-Gaussian distribution.
>
> To demonstrate that the assumption can hold in practice, we provide some empirical results. For independence, we compute the 50 percentile of absolute correlation coefficient between gradients. For sub-Gaussianess, we conduct a concentration test where we compare the area under curve of $P(|g - \bar{g}|) > t$ v.s. $t$ and that of a Gaussian (i.e., $e^{-t^2/(2σ_g^2)}$ v.s. $t$). For sub-Gaussian distribution, the former should be smaller than the latter. For RestNet18, we randomly select $10000$ parameters to compute the gradient statistics due to computational constraint.
>
> |Dataset | abs. corr. coef. @ 50% | mean AUC Gradients | mean AUC Gaussian |
> |--------|-------------------------|--------------------|-------------------|
> |MNIST + CNN | 0.0671                  | 8.29e-03           | 3.10e-02          |
> | Covertype + LR | 0.287 | 1.21e-02 | 3.35e-02 |
> | CIFAR10 + Resnet18 |0.218|5.07e-03|1.02e-02|
>
> It can be observed that while there is correlation between gradients, they are not significant. The gradients also exhibit a sub-Gaussian according to the tail-decay test.

---

> > ### Author Response · Authors · 2024-11-20
> >
> > ## Response Part (2/2)
> >
> > **4. Abstract Phrasing on "Provable" Results**
> >
> > > the abstract says you "provably remove the linear scaling of estimation uncertainty," but what is actually proven seems much weaker.
> >
> > Thank you for this suggestion. While many theoretical results are conditioned on assumptions, we recognize the importance of accurately reflecting the strength of our claims. Following your advice, we have revised the abstract to adopt a more precise and moderate tone.
> >
> > ---
> >
> > We hope that our clarifications and additional empirical results have addressed your concerns and improved your opinion of our work. We welcome further discussions or questions you may have.

---

> > > ### Comment · Reviewer_wKAq · 2024-11-26
> > >
> > > I have read the other reviews and your responses. After giving it some thought, I retain my opinion of the paper.
> > >
> > > I feel there is a mismatch between (i) the theoretical results and (ii) their purpose and discussion. The quantity called "estimation uncertainty" I still find rather un-interpretable. For example, it does not reflect the analyst's uncertainty about the value of point $j$, nor does it capture uncertainty about value estimation in the absence of privacy noise (since the noise changes the trajectory). If I want to minimize the "estimation uncertainty," why don't I simply iid noise with $k=1$? Informally, we know the answer relates to the quality of optimization, but this type of question seems to be missing from the paper. It's not clear what questions this conditional variance term addresses. For similar reasons, I don't understand the "bias" analysis that shows up in Props 5.2 and 5.3, either.
> > >
> > > Looking back, I believe I would have been more receptive to this paper if it was introduced as fundamentally practical in nature, with the math presented later as a way to give confidence to the empirical results.
> > >
> > > I have three further comments on the sub-Gaussian assumption.
> > > 1. Any future versions of this work should include definition of multivariate sub-Gaussian distributions.
> > > 1. Props 5.2 and 5.3 assume that the gradients are sub-Gaussian with identity covariance, which is a weaker assumption than the independent coordinates you discuss in another comment. I'm not sure about the proof where this assumption shows up, on p33: I don't see why the fourth moment of $z_t^{'*}$ is strictly bounded by that of $z_t^*$.
> > > 2. I'm not sure I was able to correctly parse the description of your sub-Gaussian test experiments. However, I do not feel this is an important point.

---

> ### Author Response · Authors · 2024-11-27
>
> ## Response Part(1/2)
>
> Thank you for your thoughtful feedback. We would like to address your questions and clarify our responses as follows:
>
> ---
>
> > I feel there is a mismatch between (i) the theoretical results and (ii) their purpose and discussion. The quantity called "estimation uncertainty" I still find rather un-interpretable. For example, it does not reflect the analyst's uncertainty about the value of point
> , nor does it capture uncertainty about value estimation in the absence of privacy noise (since the noise changes the trajectory).
>
> Thank you for raising this concern. We would like to clarify that the *estimation uncertainty* captures the uncertainty about the value of an estimate through its definition as conditional variance. Importantly, we **do not** define estimation uncertainty to capture uncertainty in the absence of privacy noise; rather, it is specifically intended to reflect the uncertainty induced by privacy (see lines 203-207).
>
> By defining estimation uncertainty in this way, a high level of estimation uncertainty directly corresponds to substantial uncertainty in the estimation of data value due to differential privacy (DP). Consequently, reducing estimation uncertainty is necessary to improve data value estimates (see lines 208-210), although the converse may not hold.
>
> We would greatly appreciate it if you could indicate specific instances in the paper where you perceive a "mismatch".
>
> ---
>
> > If I want to minimize the "estimation uncertainty," why don't I simply iid noise with $k=1$? Informally, we know the answer relates to the quality of optimization, but this type of question seems to be missing from the paper. It's not clear what questions this conditional variance term addresses.
>
> Thank you for your question. We wish to re-emphasize that we **do not** claim that minimizing estimation uncertainty necessarily yields the best data value estimates (see lines 212-217). To provide an explicit statement of what estimation uncertainty entails, we have the following:
>
> *Under certain utility functions, low estimation uncertainty is a necessary but not sufficient condition for obtaining high-quality data value estimates.*
>
> This statement follows directly from Proposition 5.1.
>
> While alternative methods (e.g., using $k=1$ in your example) could potentially achieve lower estimation uncertainty, our work specifically addresses the naive Monte Carlo (MC) approach. A thorough comparison with methods outside the MC framework would require additional analysis, which is beyond the scope of this work.
>
> ---
>
> > I don't understand the "bias" analysis that shows up in Props 5.2 and 5.3, either.
>
> Thank you for raising this point. The bias analysis in Propositions 5.2 and 5.3 serves as a complementary result, highlighting the bias of our estimator. In our earlier response, we assumed that your query was related to the bias of our proposed estimator, given your example of setting $\psi=0$, which results in zero estimation uncertainty but introduces substantial bias.
>
> Based on your response, we now understand that you may be questioning whether our definition of estimation uncertainty appropriately facilitates comparisons between our method and alternatives like setting $\psi=0$. We reiterate that while controlling estimation uncertainty is necessary, it alone **does not** guarantee accurate data value estimates. However, failing to control estimation uncertainty invariably leads to poor estimates. A thorough comparison with methods outside of the MC framework is beyond the scope of this work.
>
> ---
>
> > Looking back, I believe I would have been more receptive to this paper if it was introduced as fundamentally practical in nature, with the math presented later as a way to give confidence to the empirical results.
>
> Thank you for recognizing the empirical results presented in our work. We would like to clarify that our empirical results are grounded in the theoretical framework we establish. This framework specifically addresses the problem of linearly scaling estimation uncertainty in the naive MC approach, which causes the noise induced by DP to escalate. We emphasize that while we solve a practical problem, our method **is not** practical in nature, but based on properly defined theoretical results (Proposition 5.2 and Proposition 5.3). We would like to emphasize the significance of the theoretical results, as highlighted in other parts of this response.
>
> ---
>
> > Any future versions of this work should include definition of multivariate sub-Gaussian distributions.
>
> Thank you for this suggestion. We have included a definition of multivariate sub-Gaussian distributions at their first mention in our revised version (see lines 330-332).

---

> ### Author Response · Authors · 2024-11-27
>
> ## Response Part (2/2)
>
> ---
>
> > Props 5.2 and 5.3 assume that the gradients are sub-Gaussian with identity covariance, which is a weaker assumption than the independent coordinates you discuss in another comment.
>
> Thank you for this observation. We wish to clarify that we have updated the assumption in Propositions 5.2 and 5.3 to specify a "diagonal" sub-Gaussian distribution (see our earlier response). The term "isotropic" in the original version was a mis-spelling. Our proofs have consistently relied on the assumption that gradients are diagonal sub-Gaussian random variables with varying variances and $0$ covariance.
>
> ---
>
> > I'm not sure about the proof where this assumption shows up, on p33: I don't see why the fourth moment of $z_t'^4$ is strictly bounded by that of $z_t^4$.
>
> Thank you for your question. The fourth moment of $z_t'^4$ is strictly bounded by that of $z_t^4$ due to the sub-Gaussian assumption, as shown in Lemma 1.4 of [1]. Intuitively, sub-Gaussian distributions exhibit smaller higher moments than Gaussian distributions due to their stronger tail-decay properties.
>
> In our proof, we acknowledge a slight oversight in omitting the constant in the inequality, as we focused on the asymptotic bound. We have now amended the proof and included a more detailed derivation. Specifically, we should have $\mathbb{E}[z_t'^4] \leq \frac{16}{3} \mathbb{E}[z_t^4]$.
>
> ---
>
> > I'm not sure I was able to correctly parse the description of your sub-Gaussian test experiments. However, I do not feel this is an important point.
>
> Thank you for this feedback. The sub-Gaussian test experiment demonstrates that the diagonal sub-Gaussian distribution approximately holds in practice. Specifically, we report the correlation coefficient and tail-decay statistics. The small correlation coefficients suggest that gradients exhibit near-independence, while the mean AUC of gradients being smaller than that of a corresponding Gaussian distribution supports strong tail-decay and the sub-Gaussian property.
>
> ---
>
>
> **Summary of Key Clarification**
>
> We summarize our key clarifications to your concerns:
> 1. Our definition of estimation uncertainty captures the uncertainty of data value estimates caused by DP noise. Our central claim, supported by Proposition 5.1, is that controlling estimation uncertainty is a **necessary but not sufficient** condition for achieving high-quality data value estimates. While alternative methods (e.g., setting $\psi=0$ or $k=1$) may achieve low estimation uncertainty, they do not necessarily result in better data value estimates. This distinction is crucial, as establishing necessary conditions is as important as identifying sufficient ones.
>
> 2. Our proofs rely on the assumption that gradients follow a **diagonal sub-Gaussian distribution**, which enables bounding noise moments via Gaussian random variable bounds. Our experimental results further support this assumption, showing that gradients are approximately uncorrelated and exhibit sub-Gaussian behavior.
>
> ---
>
> We hope these clarifications address your concerns and improve your understanding of our work. We welcome any further questions or feedback and would be happy to discuss these points in more detail.
>
>
>
> [1] Chapter 1, Lecture Notes on Sub-Gaussian Random Variables. MIT OpenCourseWare. Accessible at: https://ocw.mit.edu/courses/18-s997-high-dimensional-statistics-spring-2015/a69e2f53bb2eeb9464520f3027fc61e6_MIT18_S997S15_Chapter1.pdf

---

> > ### Comment · Reviewer_wKAq · 2024-11-27
> >
> > Thank you for your detailed response.
> >
> > I am close to satisfied on the way in which you apply the sub-Gaussian assumption. I don't think your constants are quite right in your application of the lemma from Rigollet: the coordinates of the gradient may be dependent (even though they are uncorrelated), so one cannot apply the lemma coordinate-by-coordinate. But I agree that there is at most a constant-factor difference. (While looking at the proof, I noticed an unrelated inconsistency in the definition of $\sigma_g^2$.)
> >
> > I will try to restate where I see the mismatch between the theorems and the rest of the paper. The experiments strongly suggest that correlated noise is better for data value estimation. The paper interprets the theorems as supporting this claim. I disagree with this interpretation.
> > - We want our private estimate $\tilde{\psi}_i$ to approximate the semivalue $\phi_i$. Since $\phi_i$ seems difficult to compute even non-privately, we might also ask how well $\tilde{\psi}_i$ approximates $\psi_i$, the non-private MC estimate. I do not view the results in this paper as addressing these questions.
> > - We agree that the algorithms using correlated noise satisfy a necessary condition for $\tilde{\psi}_i$ to be a good approximation. But many worthless algorithms also satisfy this condition, such as setting $\tilde{\psi}_i=0$, or a low-variance random walk in $\theta$-space. So I cannot interpret this theorem as evidence that the correlated noise algorithms are good.
> > - We see that iid noise causes the conditional variance to scale with $\Omega(k)$. However, perhaps the best private algorithm for producing $\tilde{\psi}_i \approx \psi_i$ has this behavior and requires tuning $k$ to trade off between bias and variance? So I cannot interpret this theorem as evidence that the iid noise algorithm is bad.
> > - Proposition 5.1 in particular has another problem. The lower bound is not of the form "$\ge ck$ for an universal constant $c$." It scales with $\underline{p}^2$, which I think is exponentially small in $n$ (see below). The other terms that appear are a little opaque to me, but none appear exponentially large. So to make this lower bound a small constant, we need $k$ to be exponentially large in $n$. This doesn't seem like a reasonable regime to me.
> >
> > **Calculation:** $\underline{p} := \min_\pi n \cdot w(|P_j^{\pi} \cup \lbrace j\rbrace|)$, where $P_j^{\pi}$ denotes the predecessors of $j$ in $\pi$ and $w:[n]\to R$ is a weighting function. (It seems $j$ is implicit in the $\underline{p}$ notation.) So $\underline{p}= n \cdot \min_i w(i)$. Section 3.1 says $\sum_{i=1}^n \binom{n-1}{i-1} w(i) \le n$. Since $\sum_{i=1}^n \binom{n-1}{i-1} = 2^{n-1}$, we have $\min_i w(i) \le 2n 2^{-n}$.

---

> > > ### Author Response · Authors · 2024-11-28
> > >
> > > ## Response Part (1/2)
> > >
> > > Thank you for your response! We are delighted that you find our sub-Gaussian assumption largely satisfactory. We would like to further address your remaining concerns below:
> > >
> > > ---
> > >
> > > > I don't think your constants are quite right in your application of the lemma from Rigollet: the coordinates of the gradient may be dependent (even though they are uncorrelated), so one cannot apply the lemma coordinate-by-coordinate.
> > >
> > > Thank you for your comment. We are applying this lemma coordinate-by-coordinate by leveraging the linearity property of expectation. Specifically, $\mathbb{E}[z_t^4]$ can be expanded as the sum of the expectations of individual random variables and the products of different random variables, which are evaluated to
> > > $0$ due to their uncorrelatedness. Each individual random variable follows a sub-Gaussian distribution, which admits applying Lemma 1.4 mentioned in our previous response.
> > >
> > > ---
> > >
> > > > The experiments strongly suggest that correlated noise is better for data value estimation. The paper interprets the theorems as supporting this claim. I disagree with this interpretation.
> > >
> > > Thank you for pointing out your interpretation. We realize that there might be a slight mis-alignment between your interpretation and our actual claims. To clarify our stance and facilitate mutual understanding, we provide a comparison below between your interpretation and our claims:
> > >
> > > **Reviewer Interpretation**
> > >
> > > 1. *Implication*: An algorithm that achieves low estimation uncertainty should inherently be better for data value estimation.
> > > 2. *Consequence*: There is no evidence supporting that an algorithm with low estimation uncertainty translates to better data value estimation. Therefore, one cannot conclude that the authors' proposed algorithm is good.
> > > 3. *Best algorithm*: The best algorithm might not be related to estimation uncertainty as "perhaps the best private algorithm for producing a good estimate has this behavior and requires tuning $k$ to trade-off between bias and variance".
> > >
> > > **Author Actual Claim**
> > > 1. *Implication*: An algorithm with high estimation uncertainty is detrimental for data value estimation. The converse may not hold.
> > > 2. *Consequence*: The current naive MC method exhibits linearly scaling estimation uncertainty. Our proposed method controls the escalating estimation uncertainty. Therefore, our method is "good" in the sense that it is better than naive MC in data value estimation. We acknowledge that some "worthless algorithms" as you have mentioned are also "better" than MC (although they are equally bad empirically as our theory only offers a necessary condition).
> > > 3. *Best algorithm*: Our theory suggests that the best algorithm must have controlled estimation uncertainty. In fact, your observation that "perhaps the best private algorithm for producing a good estimate has this behavior and requires tuning $k$ to trade-off between bias and variance" is a direct implication of our theoretical results.
> > >
> > > We wish to re-emphasize that we **do not claim or interpret** that minimizing estimation uncertainty leads to better data values. In our abstract (lines 18-23), we specifically state that our theory addresses the linear scaling problem of estimation uncertainty. Our empirical results show that our methods provide better data value estimates than MC. The link between our theory and improved performance lies in the necessary condition established by our definition of estimation uncertainty, as developed in the main text.
> > >
> > > ---
> > >
> > > > Proposition 5.1 in particular has another problem. The lower bound is not of the form "$\geq ck$ for a universal constant...So to make this lower bound a small constant, we need $k$ to be exponentially large in $n$. This doesn't seem like a reasonable regime to me.
> > >
> > > Thank you for raising this concern. We note that this issue stems from a typo in our definition of $p_j(\pi)$. The correct definition should be
> > >
> > > $p_j(\pi) = 2^{n-1}n^{-1}w(|P^\pi_j \cup \{j\}|)$
> > >
> > > This ensures that $p_j(\pi)$ is normalized to have an average $1$ for expectation. We have updated this in our revision (see line 132). To provide further intuition, Sthe hapley value should result in $p_j(\pi) = 1$ for all $\pi$. We sincerely apologize for the oversight. With this correction, $p$ does not scale exponentially small with $n$, which addresses your concern.
> > >
> > > ---

---

> ### Author Response · Authors · 2024-11-28
>
> ## Response Part (2/2)
>
> **Further Comment on the Merit of Our Theory**
>
> While we acknowledge that demonstrating the sufficiency of a specific objective for achieving better data value estimation is an exciting direction for future research, we note that significant contributions in the machine learning literature often **begin with necessary conditions as their theoretical foundation**. For instance, loss minimization is generally a necessary, but not sufficient, condition for good model performance. Similarly, in our work, the focus on necessary conditions provides a robust starting point for theoretical exploration and practical implementation, even as we recognize that future progress will likely depend on establishing stronger sufficiency results to fully address the problem.
>
>
> We hope these clarifications resolve your concerns. Please let us know if further elaboration is required.

---

> > ### Comment · Reviewer_wKAq · 2024-12-02
> >
> > On the question of formal claims and implications, your reply seems like an apt summary of our disagreement. As you say in "Author Actual Claim > Consequence", you prove two theorems about conditional variance/estimation uncertainty $\mathrm{Var}[\psi_j \mid \lbrace \theta_t\rbrace]$: 1) iid noise/naive MC scales with $\Omega(k)$, and 2) correlated noise has constant variance. From these you infer that "our method is better than naive MC in data value estimation." Because of the gap between (i) bounds on the conditional variance and (ii) the goal of performing accurate data value estimation, I disagree with this conclusion. It seems we are unlikely to change each other's minds.
> >
> > Thank you for fixing the issue around $\underline{p}$. You should address $\sigma_g^2$, as it is defined inconsistently across the statement and proof of Proposition 5.3/C.10.
> >
> > On sub-Gaussianity, I have two more points. They are not critical to my evaluation of the paper and you do not need to respond to them.
> > 1. I wrote "I am close to satisfied on the way in which you apply the sub-Gaussian assumption." Your reply included "We are delighted that you find our sub-Gaussian assumption largely satisfactory." I want to be clear: I view the theorems' reliance on such an assumption as a major weakness. I only meant to convey that I (mostly) understand how those assumptions are applied.
> > 2. Your reply mentions $\mathbb{E}[z_t^4]$; I assume you are still referring to $\mathbb{E}[ \lVert z_t' \rVert^4]$ around line 1819, where $z_t' \in \mathbb{R}^d$ is sub-Gaussian with zero mean and (let's assume) identity covariance. But then I am confused by your reply, perhaps I misunderstand something? Let $z := z_t'$ for simplicity. We know $z\in\mathbb{R}^d$ is sub-Gaussian and we want to bound $\mathbb{E}[\lVert z\rVert^4]$. We write out the norm and expand: $\mathbb{E}(\sum_i z_i^2)^2 = \mathbb{E}\sum_i z_i^4 + 2\mathbb{E}\sum_{i<j} z_i^2 z_j^2$. The cross-terms are $\mathbb{E}[ z_i^2 z_j^2]$ and these do not evaluate to zero even for Gaussians. Moreover, and this is my whole point here, in general we have $\mathbb{E}[ z_i^2 z_j^2]\neq \mathbb{E}[z_i^2]\mathbb{E}[z_j^2]$ because the vector's entries may be dependent, even if $\mathbb{E}[ z_i z_j ]= 0$ by the assumption of uncorrelatedness. The proof is easily fixed, but I'm confused that it's not already cleared up.

---

> > > ### Author Response · Authors · 2024-12-04
> > >
> > > Thank you for your thoughtful feedbacks and constructive suggestions.
> > >
> > > We greatly appreciate the time and effort you have dedicated to reviewing our work. From your response, it seems there may be a difference in perspective regarding the conclusions of our theorem and our assumption of a diagonal sub-Gaussian distribution. Specifically, the statement that "our method is better than naive MC in data value estimation" is a **logical conclusion** from our Proposition 5.1-5.3. Our assumption on the diagonal sub-Gaussian distribution is supported by **empirical evidence** to largely hold in practice, as reflected in our provided experiments during rebuttal.
> > >
> > > We fully acknowledge your valuable suggestions for improvement, including bridging "the gap between (i) bounds on the conditional variance and (ii) accurate data value estimation", as well as relaxing the diagonal sub-Gaussian assumption. While we agree these are important directions for future work, we respectfully believe that their absence should not be viewed as a "major weakness" of the current submission. Our work offers meaningful theoretical insights and practical results within the scope of our assumptions, and we hope these contributions are recognized as significant.
> > >
> > > We are grateful for your pointing out the technical oversights related to $\sigma_g$ and the proof regarding $\mathbb{E}[\Vert z_t' \Vert^4]$. While we are unable to update the PDF during this stage of the review process, we will address these issues in a revised version of our paper.
> > >
> > > Thank you once again for your detailed feedback, which has provided us with valuable directions for further strengthening our work.

---

### Meta-Review · Area_Chair_FbSZ · 2024-12-20

**Metareview:**

The paper proposes using correlated noise for data value estimation under differential privacy (DP) and provides both empirical and theoretical analysis (subject to additional assumptions) to support this claim.

**Strengths:**
* Relevant problem
* Strong empirical performance of the method

**Weaknesses:**
* Theoretical analysis relies on poorly supported assumptions and focuses on secondary metrics

Overall, while the method appears to perform well empirically, I agree with Reviewer wKAq that the misaligned and poorly justified theoretical analysis that is made a central point of the paper fundamentally reduces the value of the contribution and drops the paper below the ICLR bar for acceptance.

The controversial assumptions are those of uncorrelatedness and sub-Gaussianity in Propositions 5.3 and 5.4. The authors provided some empirical evidence to support these claims during the discussion, but this does not appear to be included in the paper itself and it is not clear why the observed correlations are insignificant as noted by the authors.

**Additional Comments On Reviewer Discussion:**

The authors addressed many but not all of the reviewers' concerns. All reviewers contributed to the discussion and updated their evaluation in light of the authors' response. In the end, Reviewer wKAq remained unconvinced by aspects of the theoretical analysis of the paper and I agree with the reviewer's evaluation.

---

### Decision · Program_Chairs · 2025-01-22

Reject